# An Antarctic ecosystem value index to quantify ecological value across trophic levels and over time

Alice K. DuVivier [1] ✉, Kristen M. Krumhardt [1], Laura L. Landrum [1],
Zephyr Sylvester [2], Bilgecan Şen [3,4], Sara Labrousse [5],
Christian Che-Castaldo [6,7], Alice Eparvier [3,8], Marika M. Holland [1],
Michelle A. LaRue[9], Cara Nissen [10,11], Michael N. Levy [1],
Stephanie Jenouvrier [5,12] & Cassandra Brooks [2,10,12]

The Southern Ocean around Antarctica is one of the fastest changing regions on the planet and an emerging resource frontier for fisheries. Here, we present the Antarctic Ecosystem Value Index created by merging ecosystem information across food web trophic levels, from phytoplankton to fish and penguins, to quantify the ecological value of marine areas around the Antarctic continent. We find that coastal polynyas - areas of reduced sea-ice - have Index values 31–72% higher than surrounding areas, suggesting that these areas are biologically valuable hot spots for a number of ice-dependent Antarctic Species. Using output from an Earth system model to generate future projections of the Index, we find that high-value locations, often within polynyas, are likely to continue to be valuable throughout the 21st century despite environmental changes. The Antarctic Ecosystem Value Index indicates that penguins lose importance as their habitat becomes increasingly unsuitable, so protecting high-value habitat areas may be critical for these species. This study also shows that while many high-value Index areas are within existing or proposed Marine Protected Areas, there are several opportunities for adopting additional protection, particularly in East Antarctica and the Amundsen Sea.

The Southern Ocean around Antarctica plays a critical role in the global climate system and biogeochemical cycles[1] and provides habitat for unique marine biodiversity[2]. Antarctic species have adapted to unique environmental conditions in the Antarctic, including the presence of strong winds, cold temperatures, months of polar night, and sea-ice. Coastal polynyas are regions along the Antarctic coast with reduced sea-ice. They form repeatedly in particular locations due to strong katabatic winds from the continent blowing ice away from the coast.

[1]U.S. National Science Foundation National Center for Atmospheric Research, 1850 Table Mesa Drive, Boulder, CO, USA. [2]Environmental Studies Department, University of Colorado at Boulder, Boulder, CO, USA. [3]Woods Hole Oceanographic Institution, Woods Hole, MA, USA. [4]Appalachian Laboratory, University of Maryland Center for Environmental Science, Frostburg, MD, USA. [5]Laboratoire d'Océanographie et du Climat: Expérimentations et approches numériques (LOCEAN), UMR 7159 Sorbonne-Université, CNRS, MNHN, IRD,IPSL, Paris, France. [6]U.S. Geological Survey, Wisconsin Cooperative Wildlife Research Unit, Madison, WI, USA. [7]Department of Forest and Wildlife Ecology, University of Wisconsin-Madison, Madison, Wisconsin, USA. [8]Master de Biologie, École Normale Supérieure de Lyon, Université Claude Bernard Lyon 1, Université de Lyon, Lyon, France. [9]School of Earth and Environment, Te Whare Wnanga o Waitaha - University of Canterbury, Christchurch, New Zealand. [10]Institute of Arctic and Alpine Research, University of Colorado at Boulder, Boulder, CO, USA. [11]Department of Freshwater and Marine Ecology, Institute for Biodiversity and Ecosystem Dynamics, University of Amsterdam, Amsterdam, The Netherlands. [12]These authors contributed equally: Stephanie Jenouvrier, Cassandra Brooks. ✉e-mail: duvivier@ucar.edu

Polynyas have specific oceanographic conditions that drive high primary productivity from massive phytoplankton blooms[3,4]. They are also areas predators, such as seals and penguins, use to access pelagic resources[5]. The Antarctic ecosystem is facing critical threats from climate change, commercial fishing, and other stressors (e.g., pollution)[6–8]. Amidst these threats, an international effort is underway towards developing a network of Southern Ocean Marine Protected Areas (MPAs) that is representative of marine biodiversity and ecosystems[9]. To guide this process, areas of high biodiversity and ecological importance need to be identified for potential inclusion in this network of Southern Ocean MPAs.

Management of the Southern Ocean is the responsibility of the multi-national Commission for the Conservation of Antarctic Marine Living Resources (CCAMLR), and CCAMLR's mandate stipulates an ecosystem and science-based approach[10]. CCAMLR has shown an increasing emphasis on managing for climate change[11], thus necessitating methods that include future changes relevant for regional-scale management. Previous studies of the Southern Ocean have sought to identify regions that are the highest priority for biodiversity. Some have focused on aggregating animal-borne satellite tracking data from bird and mammal predator species to assess habitat importance, high-use areas, and where species are most abundant[12–14]. These previous studies provide important present-day information for upper trophic levels in the Antarctic ecosystem, but they do not provide information for the future or direct inclusion of lower trophic levels. Other studies have focused on how $CO_2$ emission scenarios influence future climate risk for marine life, though these often focus on one species (e.g., Emperor penguins, *Aptenodytes forsteri*)[15,16], fewer trophic levels[17,18], or are global in scope and provide relatively little coverage of the Southern Ocean[19]. Additionally, local-scale assessments of valuable ocean habitat have been completed for policy purposes for a particular planning region[20]. However, a comprehensive evaluation spanning trophic levels, applicable around the Antarctic continent, and incorporating climate change effects in the Antarctic has yet to be conducted.

Here, we present the Antarctic Ecosystem Value (AEV) Index based on several key trophic levels. We focus on areas covered by seasonal sea-ice in five important MPA planning regions - the Weddell Sea, East Antarctic, Ross Sea, Amundsen Sea, and Antarctic Peninsula. These regions include both proposed and existing MPAs. The AEV Index is based on input layers of ecosystem metrics that span trophic levels: net primary productivity, Antarctic krill growth potential, demersal fish biomass potential, and Emperor penguin and Adélie penguin (*Pygoscelis adeliae*) populations. We use data from an Earth system model (ESM) that includes marine biogeochemistry[21,22] for both a historical ice-ocean reconstruction[23] and a fully-coupled, free-running model simulation that includes both a historical period and a future projection with upper-middle $CO_2$ emissions (scenario SSP3-7.0)[24]. The model data are combined with an empirical krill model[25], a fish model[26], and penguin demographic models[15,27–29]. The AEV Index, computed by adding normalized values around Antarctica for each input layer and scaling these by the regional maxima, serves as an indicator of the significance of various areas in a management region across trophic levels within the seasonal sea-ice zone.

The AEV Index is useful for three reasons. First, it seeks to comprise the broader ecosystem value by focusing on Antarctic-specific species and integrates across trophic levels (see[30]). Second, it focuses on management-relevant scales and areas for ecological conservation. And third, it provides future projections of these regions to assess how climate change is likely to affect ecologically valuable regions. When we use the term "valuable" to describe a location, we mean that the location is important for the ecosystem itself, not for any other purpose (e.g., ecosystem services, commercial fishing). When locations of high density or productivity overlap in the same location for multiple AEV Index input layers, it implies species connectivity and richness in

this particular location and that this location is likely important for ecosystem function. We demonstrate that areas most ecologically valuable across these trophic levels are also frequently associated with coastal polynyas. We also show that areas valuable in the present-day are projected to stay valuable until the 2090s, suggesting that they may be optimal for inclusion within existing and proposed MPAs.

## Results
### Present-day AEV Index
We compute a historical reconstruction of the AEV Index that approximates observed conditions from ∼ 2000-2020. We calculated two versions of the AEV Index by scaling either by the largest summed value in the entire hemisphere (Fig. 1a) or within a regional domain (Fig. 1b). Acknowledging that the magnitude of the AEV Index is not directly comparable between the two panels due to the differences in scaling, we find that the Ross Sea and tip of the Antarctic Peninsula stand out as particularly valuable and that many coastal regions have elevated AEV Index values compared to their surroundings in both formulations. It is important to note that the AEV Index values are not directly comparable between the hemispheric and regional indices, nor between separate regions, because each grid point has been scaled differently. To directly compare the most valuable locations, we use three bins - exceptional (top 5th percentile), very high (top 10th percentile), and high (top 25th percentile) value - to describe points based on the distribution of AEV Index values within a particular region. All other points are classified as other (lowest 75th percentile). Grid points classified within the exceptional, very high, or high bins are the most valuable within a particular region (and we often refer to the combination of all three bins as highly valuable areas), and since they are determined from consistent percentiles from regional distributions of the AEV Index they are directly comparable between regions.

In both the hemispheric and regional AEV Indices (Fig. 1c and d) the most valuable areas occur along the Antarctic coasts. There are also some points classified as valuable in the northern parts of the domain in each region, and these are dominated by the Krill Growth Potential (KGP) input layer to the AEV Index (Supplementary Fig. 2). The hemispheric scaling leads to more points with higher value in the Ross Sea and Antarctic Peninsula (Supplementary Fig. 1c), indicating that on a hemispheric basis they have large numbers of penguins and high primary and secondary productivity. However, because so many points in these two regions are classified within the exceptional, very high, or high range using hemispheric scaling it can be difficult to identify and evaluate valuable areas within the CCAMLR MPA planning domains that may be locally critical for the ecosystem. With regional scaling the other regions gain importance as the scalings become regional instead of hemispheric, and local hotspots for biology are emphasized more in the AEV Index bins (Supplementary Fig. 13c). For the rest of this paper we focus on regional AEV Index values to draw attention to regional areas that are biologically valuable and policy relevant.

The most valuable areas occur along the coast (Fig. 1d) and are generally places where there are high values across trophic level inputs to the AEV Index (Supplementary Fig. 2). Specifically, they have high net primary productivity (Fig. 2a) and krill growth potential (Fig. 2b). They also show high demersal fish biomass potential (Fig. 2c). Demersal fish spend part of their life cycle at the sea floor, and present-day Antarctic species include commercially-important Antarctic toothfish. The calculation of demersal fish biomass potential depends on particulate organic carbon (POC) export to the benthos, so the highest concentration of demersal fish biomass potential occurs on the continental shelves below areas with high NPP. Emperor and Adélie penguin colonies are often located in close proximity to the locations with high productivity at lower trophic levels (Fig. 2d, e). The alignment of higher trophic level predators with lower trophic level productivity suggests that the predator colonies may be located so that

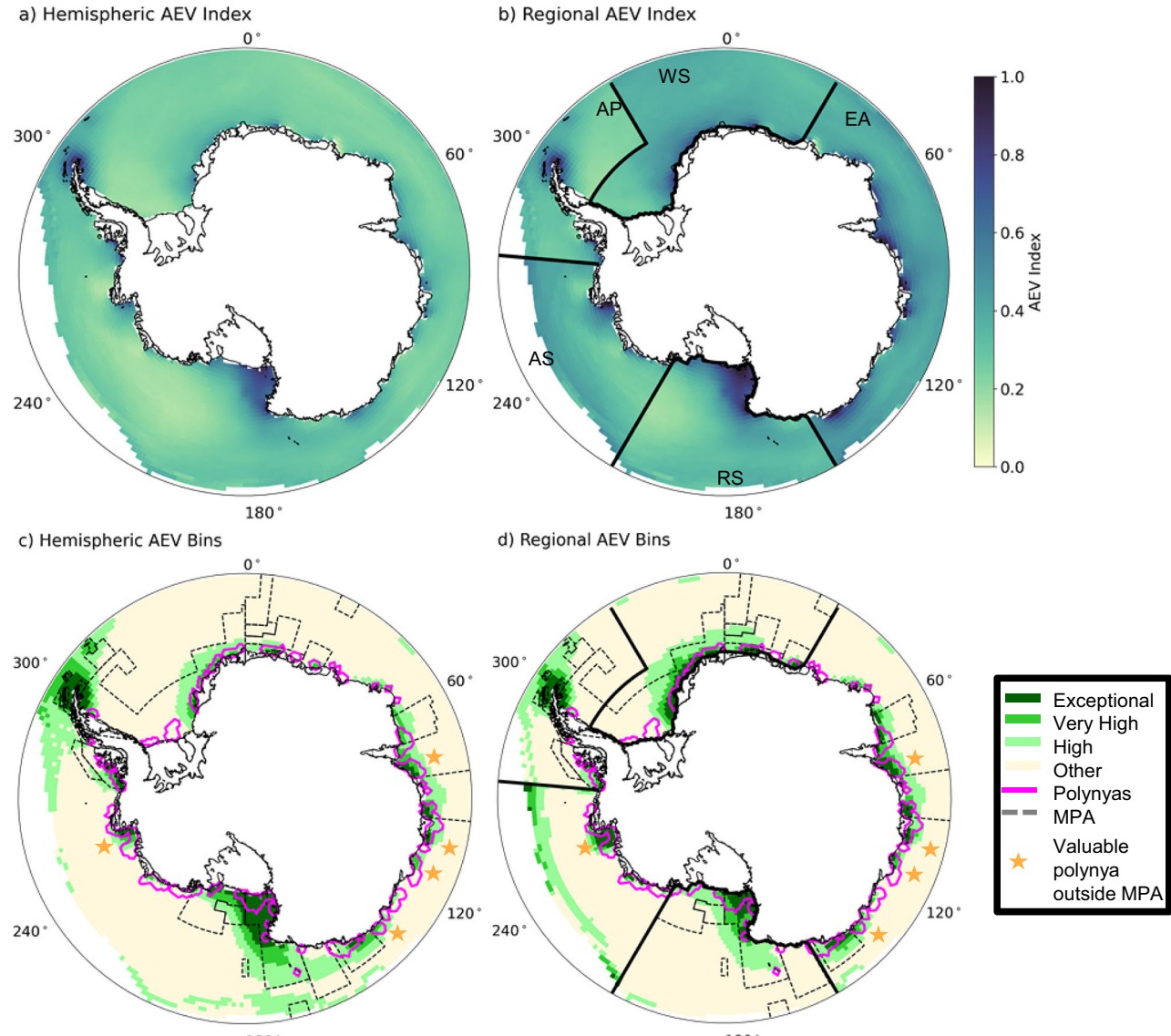

**Fig. 1 | Antarctic Ecosystem Value Index and relationship to polynyas and Marine Protected Areas.** Antarctic Ecosystem Value (AEV) Index values derived from historical reconstruction ( ∿ 2000–2020) for (**a**) hemispheric, and (**b**) the five management-relevant regions (WS Weddell Sea; EA East Antarctica; RS Ross Sea; AS Amundsen Sea; AP Antarctic Peninsula) weightings. In **c**, **d** the locations of exceptional (dark green, top 5th percentile), very high (green, top 10th percentile), and high (light green, top 25th percentile) value within each region compared to other (light yellow, bottom 75th percentile). **c**, **d** include locations of satellite observed typical polynyas (magenta), adopted and proposed Marine Protected Areas (MPAs; gray dashed lines), and gold stars indicate valuable polynyas that are not within boundaries of adopted or proposed MPAs. Coastlines were provided by the US National Ice Center[98]; adopted MPA boundaries were provided by Commission for the Conservation of Antarctic Marine Living Resources (CCAMLR)[87] and proposed MPA boundaries were provided by the Australian Antarctic Division (East Antarctic), Instituto Antártico Argentino (Domain 1), Alfred Wegner Institute (Domain 3), and Norwegian Polar Institute (Domain 4).

they can exploit the sea-ice substrate to access essential pelagic food resources.

The AEV Index also shows there is co-location of coastal polynyas and valuable areas across the ecosystem and that the AEV Index values within polynyas are on average 30–70% higher than outside polynyas, though there is higher variability within polynyas as well (Fig. 3a; and Table 1). While polynyas occupy less than 7% of the total area in any region, the total exceptional value area also within polynyas ranges from 14–59% (Table 1). While polynyas occupy a small spatial footprint, they are frequently the most valuable areas within each region. At any given longitude, the total area occupied by the three high value AEV Index bins typically makes up less than 40% of the total area (Fig. 3b), but there are several exceptions where valuable regions comprise over 50% of the total area. These occur in the D'Urville region, the Ross Sea, the Amundsen Sea, and the northern tip of the Antarctic Peninsula.

Other than at the tip of the Antarctic Peninsula, these exceptionally valuable longitudes are co-located with polynyas (Fig. 2c and d). While not all polynyas are identified as highly valuable, many highly valuable areas correspond to locations where polynyas typically form. Many valuable polynyas are within proposed or adopted MPA boundaries (Figs. 1 and 3; Table 1). However, there are four valuable polynya areas that are not currently in adopted or proposed MPA boundaries (Figs. 1 and 3, gold stars); three of these areas are in the East Antarctica management region and one is in the Amundsen Sea management region.

**Future AEV index**

To obtain future projections of the AEV Index, we use data from a fully-coupled, free-running model simulation instead of data from an ice-ocean model simulation for the historical reconstruction in the

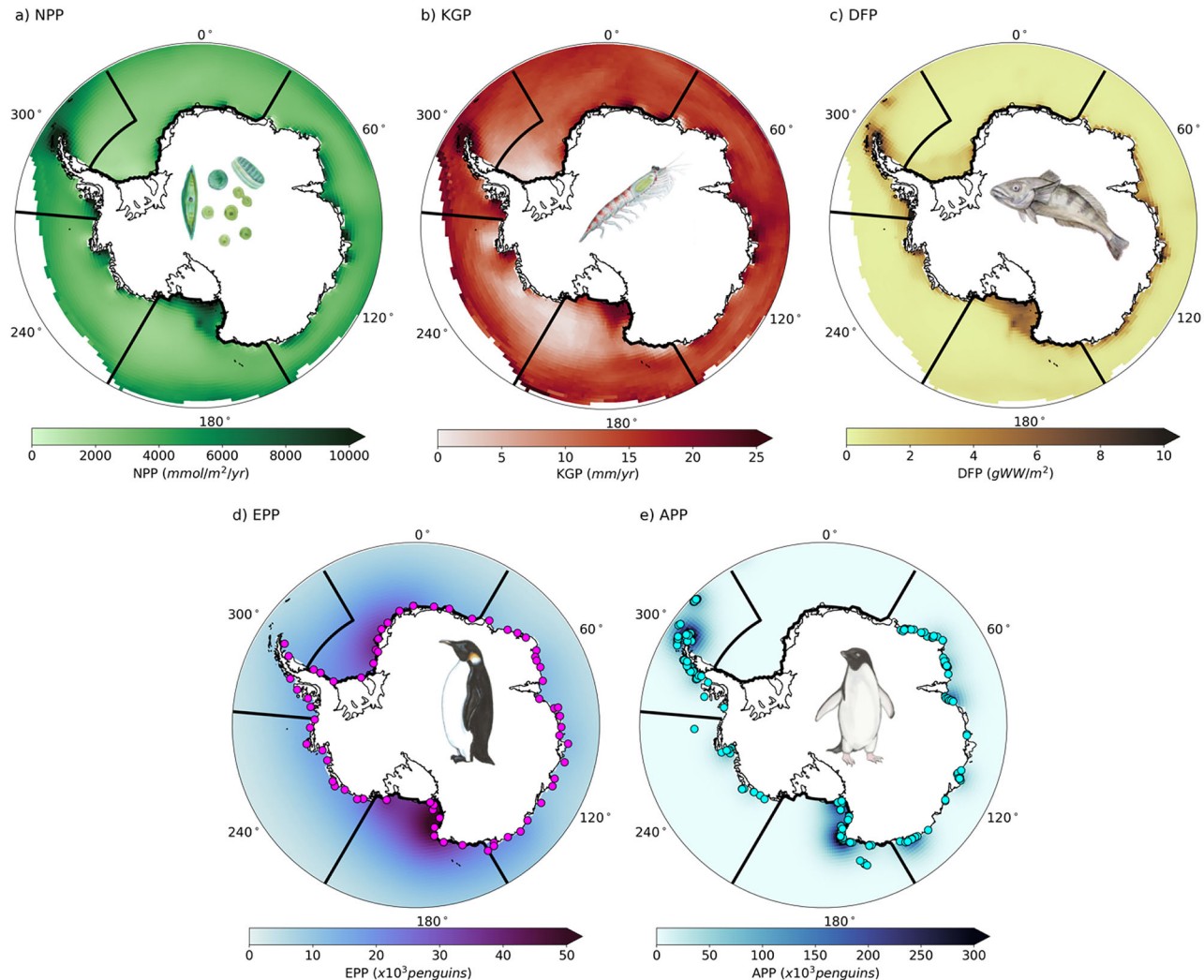

**Fig. 2 | Antarctic Ecosystem Value Index Input Layers.** Historical reconstruction ( ∽ 2000–2020) input layers before scaling: **a** net primary productivity (NPP), **b** krill growth potential (KGP), **c** demersal fish biomass potential (DFP), **d** Emperor penguin (*Aptenodytes forsteri*) population (EPP), and **e** Adélie penguin (*Pygoscelis adeliae*) population (APP). Locations of satellite observed penguin colonies are shown by circles on panels d and e. The five management-relevant regions (see Fig. 1b for labels) are shown on each panel for reference, but input layers have not been regionally scaled. Coastlines were provided by the US National Ice Center[98]. Species illustrations courtesy of K. Krumhardt.

previous section (see Section 10 for details). Since both methods identify the same large valuable areas in all regions for the present day (compare Figs. 4a and 1d), it is reasonable to use the free-running, model-based AEV Index to assess potential future changes. Future projections show that the locations with the highest AEV Index in the seasonal sea ice zone generally remain consistent in the same coastal locations from 2000 to 2090 (Supplementary Figs. 3 and 4). Over the 21st century, the highly valuable coastal points generally increase in value, with notable exceptions in the Weddell Sea and Ross Sea (Fig. 4e–g). The average AEV Index in polynyas remains consistently higher than outside polynyas in both the 2000s and 2090s, and the fraction of the highly valuable regions within polynyas remains high throughout the 21st century (Table 2). In the Ross Sea, the AEV Index decreases very slightly though still remains highly valuable in most areas. The most dramatic decline in high value regions occurs in the Antarctic Peninsula and Weddell Sea as the sea ice zone retreats (Fig. 4). In particular, sections of the Antarctic Peninsula and East Antarctic regions become ice free year round and are no longer optimal habitat for Antarctic species that rely on sea-ice.

Investigation of the contribution of individual inputs to the AEV Index reveals that the importance of penguins to the index is projected to decrease while lower trophic levels become more pronounced (Fig. 5 and Supplementary Fig. 4; Table 3). In the 2000s, the Emperor penguin layer is particularly prominent in the Weddell Sea and Ross Sea, while the Adélie penguin layer is prominent in the Antarctic Peninsula. However, by the 2090s the penguin dominant regions have shrunk dramatically. The shift in which input layer is the primary contribution to the AEV Index reflects large changes occurring throughout the food web and highlights potential climate-change impacts on whole-ecosystem dynamics. Below, we describe changes in each AEV Index input layer both hemispherically and in regional averages.

Net primary productivity is elevated in polynyas, and total polynya area does not change substantially over the 21st century (Fig. 6a–c). In all regions total net primary productivity increases over time (Fig. 7a and Supplementary Fig. 5), likely because ice loss reduces phytoplankton light limitation and extends the growing season. There may also be increased nutrient availability in polynya regions, likely

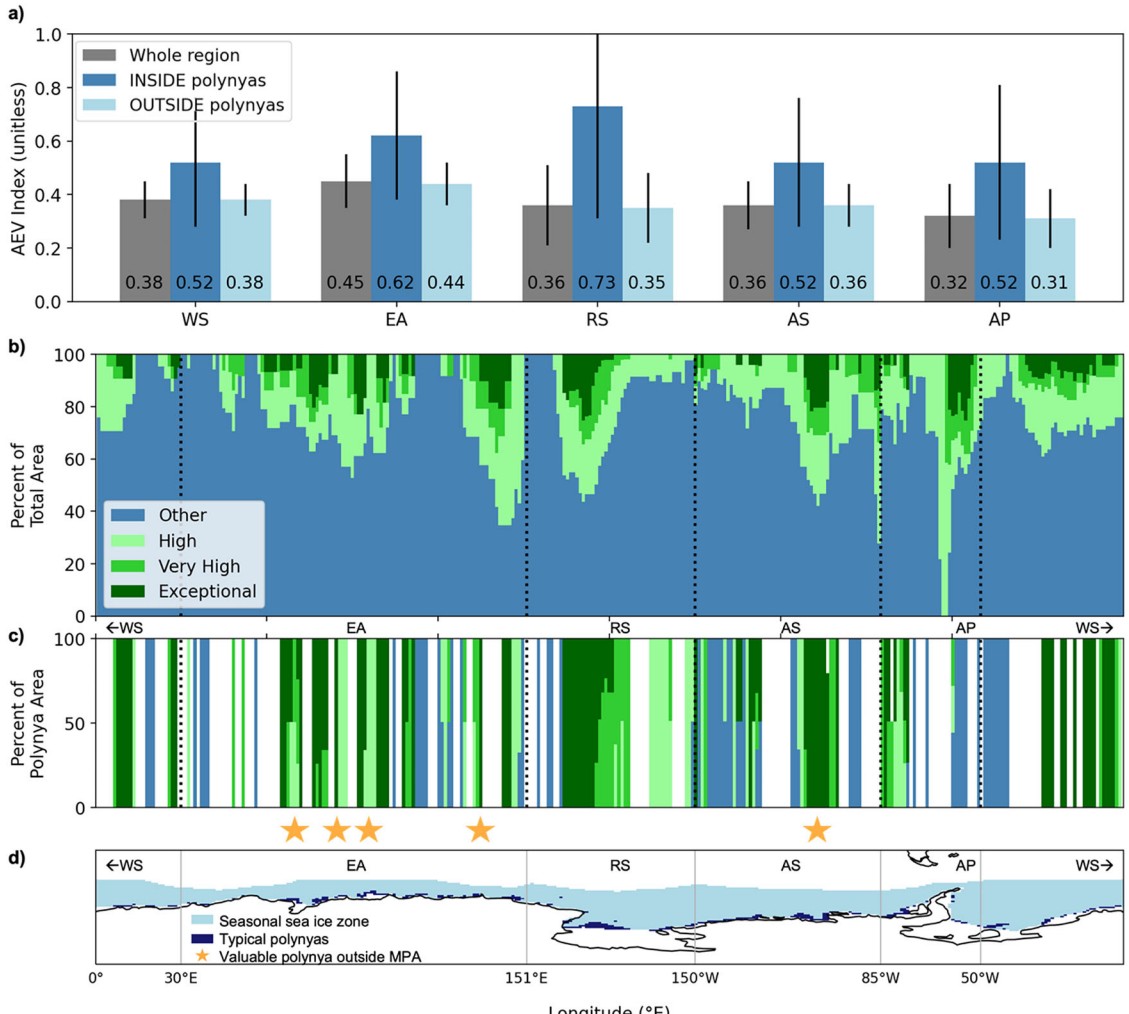

**Fig. 3 | Antarctic Ecosystem Value Index and relationship to polynyas and Marine Protected Areas within management regions. a** Regional area weighted average Antarctic Ecosystem Value (AEV) Index values (bars) and standard deviation (black lines) for the five management-relevant regions (WS Weddell Sea; EA East Antarctica; RS Ross Sea; AS Amundsen Sea; AP Antarctic Peninsula) for the whole region (gray), inside polynyas (dark blue), and outside polynyas (light blue). Bar plot height shows the mean AEV regional value and the whisker spread indicates plus or minus one standard deviation. The mean, standard deviation and number of points, respectively, is as follows for each region: WS whole region - 0.38/0.07/1431; WS polynya - 0.52/0.24/55; WS non-polynya - 0.38/0.06/1376; EA whole region - 0.45/0.10/1300; EA polynya - 0.62/0.24/97; EA non-polynya - 0.44/0.08/1203; RS whole region - 0.36/0.15/1368; RS polynya - 0.73/0.42/74; RS non-polynya - 0.35/0.13/1312; AS whole region - 0.36/0.09/1068; AS polynya - 0.52/0.24/66; AS non-polynya - 0.36/0.08/1002; WAP whole region - 0.32/0.12/791; AP polynya - 0.52/0.29/24; AP non-polynya - 0.31/0.11/767. **b** Percent of total area by longitude and (**c**) percent of polynya area by longitude for the low, medium, high, and exceptional AEV Index bins with regional weighting (see Fig. 1d). **d** Locations of the present-day seasonal sea-ice zone (purple) and typical polynyas (gold) in the model. Blue stars correspond to valuable polynyas that are within boundaries of adopted or proposed MPAs. Gold stars correspond to valuable polynyas that are not within boundaries of adopted or proposed MPAs. Coastlines were provided by the US National Ice Center[98]; adopted MPA boundaries were provided by CCAMLR[87] and proposed MPA boundaries were provided by the Australian Antarctic Division (East Antarctic), Instituto Antártico Argentino (Domain 1), Alfred Wegner Institute (Domain 3), and Norwegian Polar Institute (Domain 4).

due to increased ocean mixing in these areas, which could lead to increases in productivity as well.

Krill growth potential remains high in polynyas into the 2090s as there is ample food and the summer sea surface temperature is not yet outside the krill's habitat range (Fig. 6d–f). However, in most regions, regional average krill growth potential levels off, and in the Antarctic Peninsula and East Antarctic regions it starts to decrease by the mid 21st century (Fig. 7 and Supplementary Fig. 5). Since net primary productivity, and thus chlorophyll, increases in these regions, the decrease in krill growth potential is likely driven by rising sea surface temperatures, which increase in all regions and exceed the optimal thermal growth range for adult krill (−1 to 2 °C) and increase thermal stress on krill as they approach the upper bounds of krill's thermotolerance range (3–5 °C).

Demersal fish biomass potential also remains elevated on the continental shelves, in the vicinity of polynyas and is projected to increase given the likely increases in POC flux to the benthos with warming (Fig. 6g–i). This increase in fish biomass is consistent with other studies across multiple models that show increasing fish biomass around Antarctica[31]. However, present day Antarctic toothfish habitat temperature thresholds (see Section 10) imply that this species would lose habitat since projected bottom temperatures increase from the 2000s to 2090s (Fig. 6g, h, blue contours). As a result, regional projections of Antarctic toothfish potential biomass decline over the 21st century in most regions (Fig. 7e and f and Supplementary Fig. 5) as the ocean bottom temperatures are projected to become increasingly uninhabitable for Antarctic toothfish as they exceed the upper bound of their habitat threshold range of 1 °C[32].

**Table 1 | Historical Reconstruction regional Antarctic Ecosystem Value (AEV) Index (unitless) information related to polynyas (poly.) and marine protected areas (MPAs)**

| Region[a] | WS | EA | RS | AS | AP | Hemi |
|---|---|---|---|---|---|---|
| **AEV Index** | | | | | | |
| avg. *inside* poly. | 0.52 | 0.61 | 0.73 | 0.52 | 0.52 | 0.43 |
| avg. *outside* poly. | 0.38 | 0.43 | 0.34 | 0.36 | 0.31 | 0.28 |
| std. *inside* poly. | 0.24 | 0.24 | 0.42 | 0.24 | 0.29 | 0.23 |
| std. *outside* poly. | 0.06 | 0.08 | 0.13 | 0.08 | 0.11 | 0.08 |
| % difference[b] | +31 | +35 | +72 | +36 | +50 | +42 |
| **% Total Inside Poly.** | | | | | | |
| Regional area | 3.0 | 6.7 | 3.6 | 4.8 | 2.4 | 4.3 |
| Exceptional[c] area | 38.2 | 58.7 | 45.5 | 52.1 | 14.8 | 1.3 |
| Very high[d] area | 6.5 | 24.1 | 25.7 | 7.6 | 11.4 | 0.9 |
| High[e] area | 0.9 | 13.4 | 4.7 | 4.9 | 3.5 | 1.0 |
| **% Total Inside MPAs** | | | | | | |
| Regional area | 43.0 | 22.1 | 38.2 | 0 | 36.3 | 28.8 |
| Exceptional[c] area | 100 | 55.0 | 97.3 | 0 | 93.7 | 3.1 |
| Very high[d] area | 93.9 | 35.5 | 85.0 | 0 | 66.7 | 2.7 |
| High[e] area | 69.5 | 46.4 | 71.3 | 0 | 36.7 | 7.4 |

[a]For each region, the average (avg.) and standard deviation (std.) values are area weighted over the relevant regions. Each of the regions (*Hemi* hemispheric; *WS* Weddell Sea; *EA* East Antarctica; *RS* Ross Sea; *AS* Amundsen Sea; *AP* Antarctic Peninsula) is shown on Fig. 1a, b.
[b]Positive values indicate a larger AEV Index for points inside polynyas. Percent
Difference $= 100 * \frac{INpoly. - OUTpoly.}{0.5*(INpoly. + OUTpoly.)}$.
[c]Exceptional indicates an AEV Index in the top 5th percentile. See labels on Fig. 1.
[d]Very high indicates an AEV Index in the top 6-10th percentile. See labels on Fig. 1.
[e]High indicates an AEV Index in the top 11-25th percentile. See labels on Fig. 1.

Finally, while both the Emperor and Adélie penguin demographic projections include only sea ice concentration as an environmental driver, the projected penguin demographics have very different future trajectories depending on species. By the 2090s most Emperor penguin colonies are projected to decrease everywhere and become quasi-extinct except for in a few places in the Ross Sea and Weddell Sea regions (Figs. 6l, 7, and Supplementary Fig. 5). In contrast, the Adélie penguin populations remain fairly stable in many places and even increase in the Ross Sea and Amundsen Sea (Figs. 6o, 7, and Supplementary Fig. 5). The largest declines in individual Adélie penguin colonies are in the East Antarctic and the Antarctic Peninsula regions, yet the regional population is stable.

## Discussion

Where are the most ecologically valuable marine areas around Antarctica and will these locations change in the future? To answer this question, our study presents the AEV Index, a model-based integrative tool that synthesizes information across trophic levels for several species within the seasonal sea ice zone to investigate the present and future ecological value of broader regions. The ecological input layers we use include some representing ice-dependent species that are core ecosystem components in the Southern Ocean. The AEV is consistent with other similar Antarctic indices based on top predator distributions[12–14] (see Supplementary Fig. 1), and it adds value by accounting for previously unconsidered trophic levels and future climate change. It is important to note that the AEV Index is not a measure of overall ecosystem health, but an indicator of the ecological importance of different areas within a given region. It is a quantitative measure of how important a particular ocean location is to each species, individually, that form the input layers of the AEV Index. Since these inputs span trophic levels of the ecosystem, the AEV Index also provides a general measure of how valuable a location is to the ecosystem as a whole, and possibly for ecosystem function, in terms of likely connections between trophic levels when high value locations

for multiple input layers overlap in the same location. This location is then deemed valuable, meaning it is a hotspot of activity and productivity for the ecosystem itself.

## Importance of polynyas

Within each management region, coastal polynyas have an AEV Index 1.4–2 times larger than areas outside polynyas (Table 1), and our analysis indicates that coastal polynyas are particularly valuable for Antarctic ecosystems across trophic levels. Given the importance of coastal polynyas, many could be prioritized for protection. Importantly, not all polynyas are identified as valuable by the AEV Index in our work, and the existence of polynyas alone is not enough to assume ecosystem health. Polynyas have less and thinner sea-ice, allowing for primary productivity to begin earlier in spring as light becomes available[3,4]. High primary production and potentially warmer surface temperatures from exposure to sunlight are favorable for present-day krill growth. More mesozooplankton biomass and increased POC flux to the benthos would provide more prey resources for demersal fish through the FEISTY mechanistic forcing[26]. Polynyas are thus potentially advantageous for predators as they provide access to pelagic and benthic resources and also allow an easy ability to return to the surface for breathing. There are differences in how higher predators use polynyas; some species regularly use the yearly recurrent polynya openings[33,34] while others use ephemeral polynya openings opportunistically[5]. These differences underscore the importance of better understanding the connection between lower trophic levels and predators within polynyas. Additionally, the sustained high net primary productivity in polynyas into the future, even while Emperor penguin populations are projected to plummet, suggests that the high future AEV Index in some areas may be driven primarily by lower trophic levels rather than predators.

## Protection

The primary outcomes of this work are the identification of highly valuable regions across the marine ecosystem in the present-day and the assessment to what extent they remain valuable into the future. It is important to note that this is an exploratory scale that uses standard ESM spatial and temporal resolution data to identify locations valuable to the marine ecosystem at a large scale, but that more detailed studies for these regions is necessary for specific policy decisions. Still, this initial study identifies some important patterns. In most management regions, the majority (55–100%) of exceptional value areas are within current proposed or existing MPA boundaries; the exception is the Amundsen Sea (discussed below). The existing Ross Sea region MPA protects the majority of exceptional, very high, and high value areas in this region in the present day (Table 1), and is projected to keep protecting the majority of these high value areas (Table 2). While the MPA is set to expire in 2052, this study suggests that keeping the MPA in effect would likely continue to protect most of these valuable regions through 2090.

While we are not able to explicitly assess the impact of fishing pressures that an MPA may mitigate, we do show that there are substantial climate pressures on the Antarctic ecosystem. In management regions with proposed MPAs, CCAMLR's recent lack of progress in adopting these proposed MPAs[35] leaves the species within these areas vulnerable to the combination of both climate change and fishing pressure. This study identifies several important polynyas in the East Antarctic near Prydz Bay that remain outside proposed MPA boundaries (Figs. 1 and 2, gold stars) that are opportunities for protection. Despite the recognized ecological importance of the D'Urville region in East Antarctica, where nearly the whole area is classified as exceptional, very high, or high value (Fig. 1), the area remains without formal protection as the proposed MPA has not been adopted. Additionally, the proposed MPA excludes three nearby valuable polynyas (Fig. 1, gold stars). Finally, and notably, there are not yet any protected areas

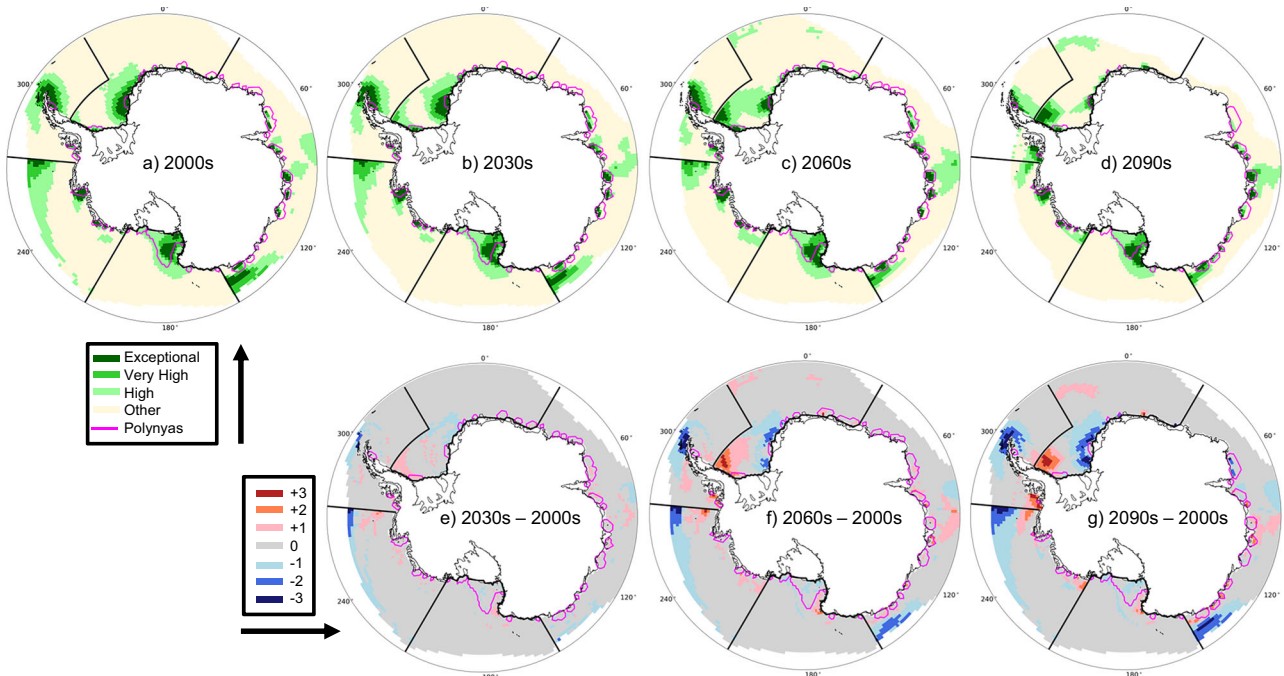

**Fig. 4 | Antarctic Ecosystem Value Index and relationship to polynyas over time.** Antarctic Ecosystem Value (AEV) Index for (**a**) 2000s, **b** 2030s, **c** 2060s, and **d** 2090s. Points with exceptional (dark green, top 5th percentile), very high (green, top 10th percentile), high (light green, top 25th percentile), and other (light yellow, bottom 75th percentile) value are shaded. These values are derived from a fully-coupled, free-running model for five management-relevant regions (labeled on Fig. 1b) and percentiles were determined individually for each region and decade. Changes in value assignment compared to the 2000s are shown for the (**e**) 2030s, **f** 2060s, and (**g**) 2090s where red colors indicate increase in value (e.g., +1 means movement from high to very high), blue colors indicate decrease in value (e.g., -1 means movement from very high to high), and gray indicates no change (e.g., 0 means it was assigned high in both decades). Locations of typical polynyas by decade are overlain in magenta in all panels. Coastlines were provided by the US National Ice Center[98].

**Table 2 | Earth System Model regional Antarctic Ecosystem Value (AEV) Index information related to polynyas (poly.) and marine protected areas (MPAs)**

| Region[a]<br>Decade[b] | WS<br>2000/2090 | EA<br>2000/2090 | RS<br>2000/2090 | AS<br>2000/2090 | AP<br>2000/2090 |
|---|---|---|---|---|---|
| **AEV Index** | | | | | |
| avg. *inside* poly. | 0.55/0.52 | 0.57/0.65 | 0.82/0.77 | 0.66/0.44 | 0.54/0.80 |
| avg. *outside* poly. | 0.38/0.41 | 0.46/0.47 | 0.36/0.43 | 0.28/0.38 | 0.26/0.26 |
| std. *inside* poly. | 0.26/0.06 | 0.19/0.22 | 0.43/0.34 | 0.44/0.34 | 0.39/0.68 |
| std. *outside* poly. | 0.06/0.13 | 0.09/0.11 | 0.14/0.15 | 0.04/0.06 | 0.10/0.08 |
| % difference[c] | +35/+24 | +21/+32 | +78/+57 | +81/+55 | +70/+102 |
| **% Total Inside Poly.** | | | | | |
| Regional area | 1.9/1.9 | 6.9/11.1 | 9.1/8.4 | 1.4/1.9 | 1.0/0.6 |
| Exceptional[d] area | 25.1/23.0 | 53.0/93.9 | 79.9/45.1 | 35.3/40.7 | 9.7/13.0 |
| Very high[e] area | 5.0/7.7 | 17.7/27.6 | 66.5/61.0 | 0/2.8 | 3.9/0 |
| High[f] area | 2.0/1.8 | 5.1/14.9 | 29.7/30.7 | 0/0 | 1.4/0 |
| **% Total Inside MPAs** | | | | | |
| Exceptional[d] area | 100/100 | 54.6/51.7 | 86.8/85.0 | 0/0 | 91.6/86.7 |
| Very high[e] area | 100/86.6 | 48.9/64.7 | 87.2/85.1 | 0/0 | 82.7/80.3 |
| High[f] area | 67.6/44.8 | 42.0/66.2 | 66.3/72.6 | 0/0 | 46.6/83.0 |

[a]For each region, the average (avg.) and standard deviation (std.) values are area weighted over the relevant regions. Each of the regions (*WS* Weddell Sea; *EA* East Antarctica; *RS* Ross Sea; *AS* Amundsen Sea; *AP* Antarctic Peninsula) are shown on Fig. 1b.

[b]Decades are calculated by averaging data from all 50 ensembles in the five years surrounding a given decade (e.g., 2000 averages include data from years 1998-2002). Thus, each statistical calculation includes 250 total samples.

[c]Positive values indicate a larger AEV Index for points inside polynyas. Percent Difference $= 100 * \frac{INpoly. - OUTpoly.}{0.5*(INpoly. + OUTpoly.)}$.

[d]Exceptional indicates an AEV Index in the top 5th percentile. See labels on Fig. 4.

[e]Very high indicates an AEV Index in the top 6–10th percentile. See labels on Fig. 4.

[f]High indicates an AEV Index in the top 11–25th percentile. See labels on Fig. 4.

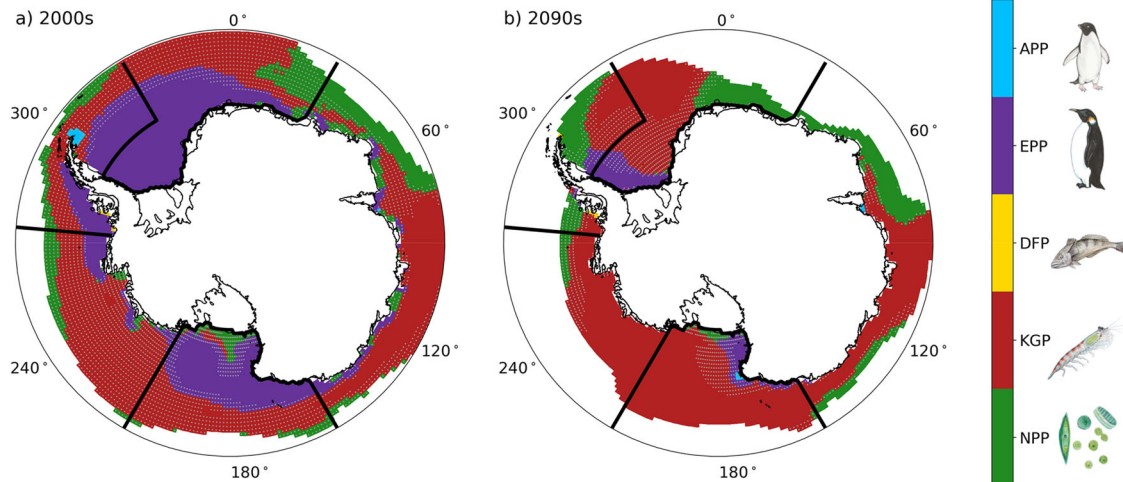

**Fig. 5 | Dominant input to the Antarctic Ecosystem Value Index.** Largest Antarctic Ecosystem Value (AEV) Index input layer for the (**a**) 2000s and **b** 2090s. Coloring indicates the input layer with the highest contribution to the index where light blue is Adélie penguin (*Pygoscelis adeliae*) population (APP), purple is Emperor penguin (*Aptenodytes forsteri*) population (EPP), yellow is demersal fish biomass potential (DFP), red is krill growth potential (KGP), and green is net primary productivity (NPP). Locations with gray stippling indicate where the dominant layer is less than 10% more than the next highest contribution. Coastlines were provided by the US National Ice Center[98]. Species illustrations courtesy of K. Krumhardt.

**Table 3 | Regional average percent (%) contribution of each input layer to the Antarctic Ecosystem Value (AEV) Index**

| Region[a] | WS | EA | RS | AS | AP |
|---|---|---|---|---|---|
| Decade[b] | 2000 → 2090 | 2000 → 2090 | 2000 → 2090 | 2000 → 2090 | 2000 → 2090 |
| **Inside** polynyas | | | | | |
| NPP[c] | 28.5 → 36.8 | 33.7 → 36.0 | 30.6 → 30.1 | 39.8 → 37.3 | 25.8 → 33.4 |
| KGP[d] | 20.1 → 37.2 | 28.8 → 48.1 | 27.9 → 27.6 | 35.9 → 49.1 | 22.2 → 30.2 |
| DFP[e] | 3.1 → 0.0 | 6.2 → 5.1 | 8.1 → 8.3 | 6.7 → 0.0 | 23.3 → 23.3 |
| EPP[f] | 48.4 → 25.9 | 23.4 → 3.3 | 31.2 → 30.9 | 15.1 → 7.5 | 25.4 → 10.8 |
| APP[g] | 0.0 → 0.0 | 7.9 → 7.6 | 2.3 → 3.1 | 2.5 → 6.2 | 3.4 → 2.4 |
| **Outside** polynyas | | | | | |
| NPP[c] | 26.7 → 22.8 | 28.2 → 23.6 | 22.2 → 16.8 | 26.9 → 15.6 | 22.4 → 19.7 |
| KGP[d] | 27.1 → 24.2 | 31.7 → 23.6 | 26.2 → 30.6 | 29.6 → 21.3 | 24.1 → 19.7 |
| DFP[e] | 0.2 → 0.1 | 0.3 → 0.2 | 0.4 → 0.4 | 0.1 → 0.1 | 0.6 → 0.4 |
| EPP[f] | 43.0 → 9.1 | 16.7 → 2.1 | 26.1 → 16.4 | 15.8 → 4.5 | 31.5 → 11.0 |
| APP[g] | 0.0 → 0.0 | 1.7 → 1.4 | 1.2 → 1.8 | 0.3 → 0.6 | 3.3 → 1.1 |
| **Valuable**[h] areas | | | | | |
| NPP[c] | 12.5 → 34.0 | 33.1 → 33.2 | 26.1 → 23.6 | 39.9 → 34.9 | 26.7 → 35.2 |
| KGP[d] | 9.3 → 42.3 | 41.1 → 55.5 | 27.4 → 30.1 | 41.8 → 50.0 | 31.8 → 32.1 |
| DFP[e] | 0.8 → 0.6 | 1.6 → 2.6 | 5.4 → 5.0 | 0.5 → 0.3 | 2.8 → 3.8 |
| EPP[f] | 77.4 → 23.1 | 20.6 → 4.9 | 36.1 → 32.5 | 17.4 → 11.9 | 26.4 → 24.6 |
| APP[g] | 0.0 → 0.1 | 3.2 → 3.9 | 5.0 → 8.7 | 0.4 → 2.8 | 12.2 → 4.3 |

[a]For each region, the average (avg.) and standard deviation (std.) values are area weighted over the relevant regions. Each of the regions (*WS* Weddell Sea; *EA* East Antarctica; *RS* Ross Sea; *AS* Amundsen Sea; *AP* Antarctic Peninsula) are shown on Fig. 1b. Note that at each grid point we have verified the sum of the contribution from each input layer is 100%, but the average percent contribution over a region does not necessarily add up to 100%.

[b]Decades are calculated by averaging data from all 50 ensembles in the five years surrounding a given decade (e.g., 2000 averages include data from years 1998–2002). Thus, each statistical calculation includes 250 total samples.

[c]Net primary productivity (NPP).

[d]Krill growth potential (KGP).

[e]Demersal fish biomass potential (DFP).

[f]Emperor penguin (*Aptenodytes forsteri*) population (EPP).

[g]Adélie penguin (*Pygoscelis adeliae*) population (APP).

[h]Valuable areas correspond to areas classified as exceptional, very high, and high, based on having an AEV Index in the top 25th percentile. See labels on Fig. 4.

proposed in the Amundsen Sea region despite studies that emphasize priority areas in the region[35]. Given the international effort to develop a network of Southern Ocean MPAs, these areas are currently gaps and therefore present opportunities to develop for protection as present-day ecological oases that are likely to remain valuable into the future.

**Gaps in ecosystem representation**

When possible, the AEV Index uses species-specific future projections, but more detailed species-specific modeling is needed, especially for ice-obligate species. We include input layers for primary producers (NPP), primary/secondary consumers (KGP), secondary consumers

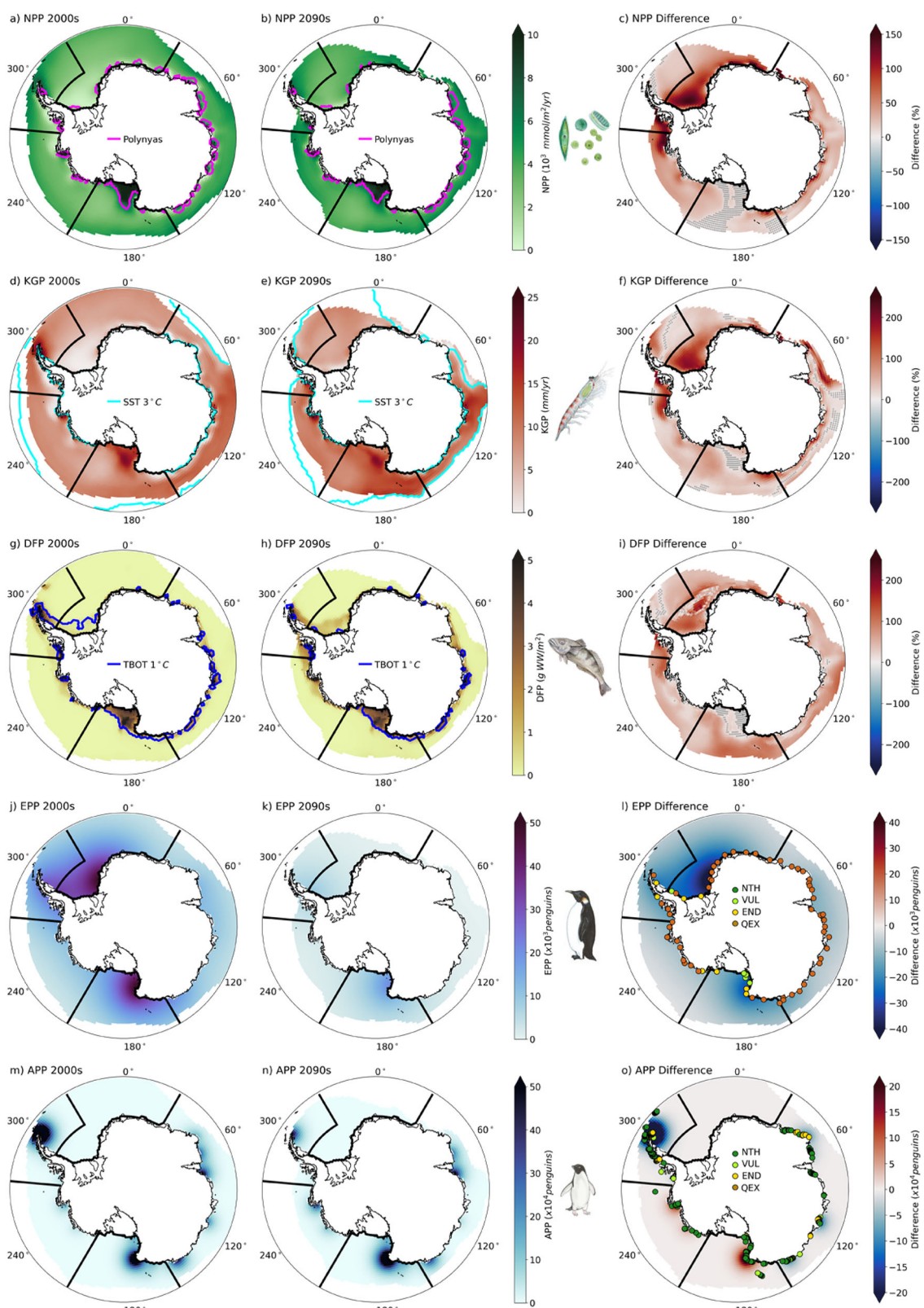

(DFP), and secondary/tertiary consumers (EPP/APP) (see Section 10 for details). However, other key ecosystem species like whales, seals, other penguins and fish, and flighted birds are not included. Some detailed species-specific models already exist describing today's environmental conditions[36–38], but these would require substantial work to couple with a global ESM and to assess potential future changes and if statistical relationships are robust in different climate states. For other

species (e.g., Antarctic silverfish, Pleuragramma antarctica) there are no existing models of which we are aware, so species-specific model development would be necessary. Our study focuses on the Antarctic-specific and ice-dependent species for which future projection data are available from existing biological models and that use the types of data available from existing ESM experiments. However, this means that we do not include other mesozooplankton or pelagic fish that are

**Fig. 6 | Changes in inputs for the Antarctic Ecosystem Value Index over time.** Antarctic Ecosystem Value (AEV) Index input layers before scaling for 2000s (left column), 2090s (middle column), and difference (right column). **a**, **b** show decadal net primary productivity (NPP) overlain with locations of typical polynyas (magenta contours). **d**, **e** show decadal krill growth potential (KGP) overlain with the 3 °C sea surface temperature (SST) contour. **g**, **h** show decadal demersal fish biomass potential (DFP) overlain with the 1 °C bottom temperature (TBOT) contour. **j**, **k** show decadal Emperor penguin (*Pygoscelis adeliae*) population (EPP)

accessibility, and panels **m**) and **n**) show decadal Adélie penguin (*Pygoscelis adeliae*) population (APP) accessibility. Percent differences are shown in (**c**), **f** and **i** where stippling indicates points where the percent difference is less than 10%. Absolute differences are shown in (**l** and **o**), and colored circles show locations of observed penguin colonies and indicators of colony health (as defined in Section 10) where "NTH" means not threatened, "VUL" means vulnerable, "END" means endangered, and "QEX" means quasi-extinct. Coastlines were provided by the US National Ice Center[98]. Species illustrations courtesy of K. Krumhardt.

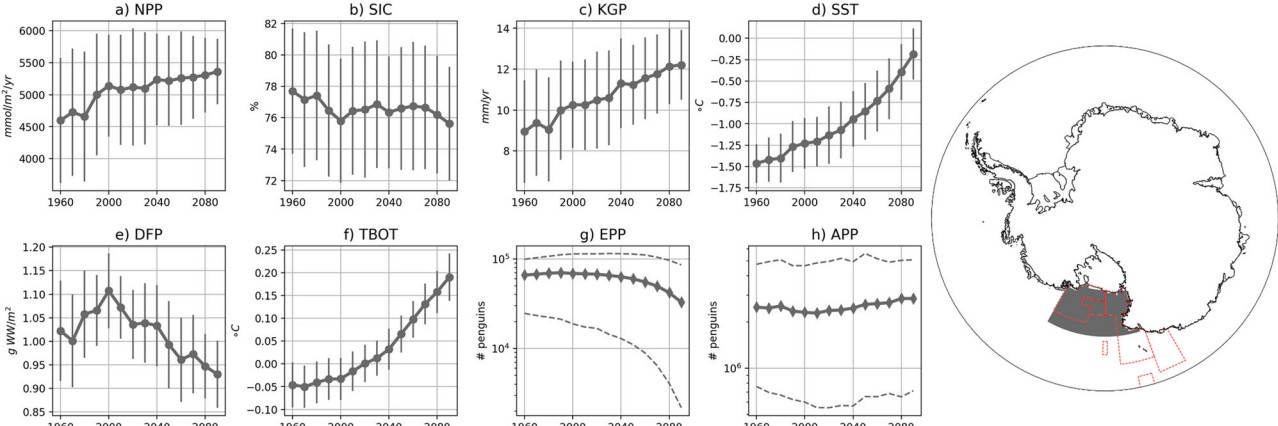

**Fig. 7 | Regional biological change over time for Antarctic Ecosystem Value Index inputs.** Regional biological and environmental variables for the Ross Sea region, as shown on the map, with adopted Marine Protected Areas (MPAs) in red dashed lines. Regional plots include (**a**) summer (months ONDJFM) net primary productivity (NPP), **b** spring (months OND) sea ice concentration (SIC), **c** summer (months ONDJFM) krill growth potential (KGP), **d** end of summer (March) sea surface temperature (SST), **e** annual mean demersal fish biomass potential (DFP), **f** annual mean bottom temperature (TBOT), **g** Emperor penguin (*Pygoscelis adeliae*) total population (EPP), and **h** Adélie penguin (*Pygoscelis adeliae*) total population (APP). Map at right shows area over which timeseries are calculated (gray shading),

Ross Sea Region Marine Protected Area boundaries were provided by Commission for the Conservation of Antarctic Marine Living Resources (CCAMLR)[87], and coastlines were provided by the US National Ice Center[98]. All timeseries plots, except EPP and APP, are area weighted averages over the shaded region and standard deviation is shown with error bars; each of these decadal statistics was calculated from 250 samples (50 ensemble members * 5 years surrounding the decade). EPP and APP are the total population over the relevant region where the solid line is the median projection and the dashed lines are the 95th percent confidence intervals directly from the metapopulation model projections. All panels have linear y axes except for EPP and APP, which have a logarithmic *y* axis.

important for energy pathways from the primary producers to predators[39,40]. Additionally, excluding species means that we may not be capturing the scale needed for protection of all critical ecosystem processes, and we can only assess processes relevant to the species represented as AEV Index inputs. While it would have been ideal to include a larger range of species, we included only those species for which we had readily available future projection information around the entire Antarctic continent. The results presented here show that the AEV Index framework is valuable as a tool to demonstrate the utility of such studies across trophic levels, and future work to include more species would help fill gaps about the Antarctic ecosystem that this study is unable to address. Until then, the AEV Index is associated with larger uncertainty in regions or at times when species not included in our assessment play a vital role in the ecosystem.

The AEV Index considers NPP across all phytoplankton species included in the ESM, but it does not include information about changes in community composition that could impact resources for higher trophic levels[41]. For Antarctic krill, we include summer growth potential estimates based on environmental data available from the ESM. This empirical metric reflects adult krill growth and uses only surface chlorophyll and sea surface temperature (SST) as inputs (see Section 10). Thus, it is an optimal measure because it uses easily available environmental variables from existing ESMs experiments to make future projections[25], but it does not account for other critical life history stages for krill. While more complex Antarctic krill models that incorporate biomass dynamics and full life cycle processes could

provide a more detailed perspective[36], such models were not used in this study because of the necessity to use existing coupled ESM data which does not have temporal or spatial resolution needed. A more sophisticated regional modeling study of krill in some of the large scale regions identified by this study would be valuable.

Nonetheless, we were able to investigate two important environmental variables known to impact Antarctic krill: SST and sea ice availability. By assessing KGP only within the sea ice zone we implicitly consider only locations where sea ice is available during the year, though not the seasonality or direct impact of sea ice on the krill lifecycle. Additionally, krill are stenothermic, meaning they have an optimal thermal range of 0–3 °C, and the empirical KGP metric has a maximum growth rate when $SST = 0.386\,°C$. As SSTs increase, the habitat conditions for Antarctic krill become unfavorable due to thermal stresses at various krill life stages[25,42–47]. Winters characterized by extensive sea ice have been associated with strong post-larval krill recruitment in the subsequent spring, indicating that sea ice is essential for survival of larvae during winter[48–50]. However, the precise mechanisms driving the connection between sea ice and krill recruitment remain unclear[51]. Besides the Antarctic Peninsula, no regions are projected to become ice free in the autumn months, but sea-ice concentrations drop sharply by 2090 in many regions (Fig. 7 and Supplementary Fig. 5). It is likely that there are smaller areas within those zones that remain ice free and become unable to support krill larvae. These localized changes could further constrain the spatial and temporal extent of suitable krill habitats, with cascading effects on the broader ecosystem.

Similarly, the projections of demersal fish biomass potential (DFP) are for a functional type, not specific Antarctic species. The ESM projections show that the habitat for Antarctic toothfish is becoming increasingly unfavorable for Antarctic toothfish as bottom temperatures rise in all regions, which drives the resulting projected DFP declines nearly everywhere over the 21st century (Fig. 7 and Supplementary Fig. 5). This is because in some regions, particularly the Weddell Sea and Amundsen Sea, the bottom temperatures exceed this temperature habitat threshold by 2090 for the upper-high emission scenario (SSP3-7.0; Fig. 6). We use Antarctic toothfish as an example species of regional demersal fish because of its importance as the largest predatory fish in the Southern Ocean, but other Antarctic demersal fish likely have similar habitat constraints. It also remains unclear to what extent southward-expanding subpolar/subtropical species could adapt to high-latitude environmental conditions on decadal time scales, where substantial sea-ice cover is projected to remain present at least for parts of the year.

## Model Uncertainty

There are a number of uncertainties that will impact the AEV Index inputs. Each biological model has their own uncertainty, yet the uncertainty is difficult to quantify, in part because of a dearth of observations in both space and time. Still, it is important to mention that this biological model uncertainty will propagate into the AEV Index calculation. There are also uncertainties in the environmental model conditions that we discuss below.

The model projections used for the AEV Index follow an upper-middle $CO_2$ emissions scenario, but there is also uncertainty in the mean climate state that would depend on the emissions scenario used. In the regional timeseries (Fig. 7 and Supplementary Fig. 5) we include error bars to show the range in possible projections in environmental variables and the resulting impact on biological variables. In all cases, the trends over time are robust for this particular emission scenario. However, depending on the level of emissions and future warming, the Antarctic climate could reach the state projected for the 2090s earlier or later than that decade in the upper-middle scenario. Recent sharp decreases in Antarctic sea-ice extent[52] may indicate a fundamental shift in the Antarctic physical climate and could suggest more rapid change than is projected here. Already, steep declines in sea ice have led to impacts throughout the Antarctic ecosystem, including on Emperor penguins[7,53-55].

Moreover, the Earth system model we use has a coarse ( ~1°), but common, grid size for this type of model. Other forced and fully-coupled models or high resolution ice-ocean models (e.g., ACCESS-OM2[56] or FESOM-REcoM[18]) could elucidate the uncertainty due to, e.g., small-scale ice features, representation of small-scale bathymetric features, or the intensity and locations of warm circumpolar deep water intrusions onto the shelf that could all impact habitat suitability[18,57-59]. Marine heat waves and other short-lived events in the present-day would not be reflected in the decadal climatology presented here, but they could become more frequent or even a permanent feature in some regions[18,58], directly impacting Antarctic species that inhabit the Antarctic shelf.

Finally, the AEV Index takes into account only pressure from climate change, and it does not include other human-induced threats like fishing or pollution. In many places, the Southern Ocean environment is increasingly unsuitable for Antarctic marine species. It is hard to fully understand the impact of habitat change without dynamic coupling, including feedback loops between trophic levels, such that changes impact the whole food web. The overall implications of the changing environment, including sea ice loss, are uncertain throughout the food web. Predators may shift to new prey species, affecting the predators' survival and reproductive success[18,53,60,61], and therefore trophic pathways. The AEV Index helps identify large-scale hot spots across trophic levels. Continuing regionally specific research and monitoring within

these areas may help us better understand how the Antarctic ecosystem may respond to threats, so that we can devise more effective ways to protect it.

## Methods

### Earth system model simulations

For this analysis we use data from the Community Earth System Model version 2 (CESM2)[21]. CESM2 includes explicit atmosphere (Community Atmosphere Model version 6; CAM6), terrestrial (Community Land Model version 5; CLM5), ocean (Parallel Ocean Program version 2; POP2), and sea-ice (CICE version 5.1.2) model components run at ~1° horizontal resolution[21]. Important for this study, CESM2 includes prognostic ocean biogeochemistry (Marine Biogeochemistry Library, or MARBL[22]). The time evolving phytoplankton and zooplankton are represented by functional types and global cycling of oxygen, C, and nutrients (including N, P, Si, and Fe). MARBL also includes specific treatment for light penetration through sea-ice to accurately represent photosynthesis in polar regions[62]. We use data from two CESM2 experiments for this analysis:

1. For the historical reconstruction data, we use the forced ocean-sea-ice (FOSI) simulation documented in ref. 23 and that is freely available. The FOSI version of CESM2 has prognostic ocean, sea-ice, and marine biogeochemistry components. The experiment is integrated over the 1958-2021 period using reanalysis data from the JRA55-do atmospheric state and runoff dataset[63]. Thus, the FOSI run provides an estimate of the observed historical period. We use the average CESM2 FOSI output from 2000-2020 when calculating the AEV Index as an estimate of present-day conditions.

2. For the free-running model data, we use an ensemble of 50 simulations from the CESM2 large ensemble (CESM2-LE;[24]) that are also freely available. Each ensemble member is initialized in 1850 from a pre-industrial control with different initial states. All ensemble members then use identical prescribed forcing for the historical (1850-2014) and future (2015-2100) periods. The historical forcing and SSP3-7.0 (upper-middle) forcing use protocols from the $6^{th}$ Coupled Model Intercomparison Project (CMIP6;[64]). Each member of the CESM2-LE is an equally likely realization of a future climate projection. Thus, the ensemble mean captures the response of the system to external forcing alone, while differences between ensemble members demonstrate the internal climate variability[65]. For each decade shown, we take the average over all 50 ensemble members and the five years surrounding the decade (e.g., 2010 average is taken over 2008–2012) such that there are 250 samples used for decadal statistics. The CESM2-LE is an ideal tool for robustly identifying climate change signals since it has a large number of ensembles from which to sample. By calculating decadal statistics this way, we can focus on the climate change signal and not internal climate variability.

The historical reconstruction AEV Index uses data from the ice-ocean model forced by atmospheric observations to reconstruct observed ocean and sea ice conditions[23]. On the other hand, an individual fully-coupled, free-running model simulation should not exactly replicate observations, since observations represent one possible response of the climate system to given external forcing in the presence of internal climate variability. Instead, to ensure the AEV Index derived from the fully-coupled, free-running model data[24] is robust, we compare the historical reconstruction (average for 2000-2020 conditions) with the fully-coupled simulation data for the 2000s (average over 50 ensemble members from 1998-2002) to ensure they produce similar results in a similar climate state. Indeed, we find that the valuable areas identified by the regional AEV Index historical

reconstruction (Fig. 1d) align well with valuable areas identified by the AEV Index calculation using a fully-coupled, free-running model for the 2000s (Fig. 4a). Both methods identify the same large, valuable areas in all regions, though the free-running model has more valuable grid points further north in the Amundsen Sea and the East Antarctic. A major benefit of the AEV Index derived from the fully-coupled model is that it can be used to identify valuable areas into the future. Because the two AEV Index formulations provide similar information in the present day, it is reasonable to use the free-running, model-based AEV Index in the next section of this manuscript to provide information about potential future changes.

### Sea-ice data

For the sea-ice data we focus on two variables: sea-ice concentration (SIC) and sea-ice thickness (SIT). These variables are directly output from CESM2 at each grid cell in both the FOSI and CESM2-LE datasets. For model evaluation, we use daily 1979-2020 Climate Data Record (CDR) SIC satellite data product available from the National Snow and Ice Data Center[66,67]; there are no long term observational datasets of Antarctic SIT available. Antarctic sea-ice in CESM2 has also been well studied and found to be reasonably represented[68–71]. Supplementary Fig. 6 illustrates that both the CESM2-FOSI and CESM2-LE reasonably represent the mean state and variability of Antarctic sea ice in both winter and summer. Thus, it is appropriate to use the CESM2 model output for analysis of Antarctic climate.

The seasonal sea-ice zone is the maximum extent to which sea-ice expands northward at the end of Antarctic winter (September). We calculate the seasonal sea-ice zone by finding all points where the September mean SIC is 15% or greater; the same method was used for both observational data from 2000–2020 and each decade from the CESM2-LE data.

Polynyas are regions of reduced SIC or SIT surrounded by more extensive or thicker sea-ice and/or coastline. Polynyas form around the Antarctic coasts in specific locations due to strong winds from the continent blowing ice away from the coast. For this study, we assess how the identified ecologically valuable regions correspond to regions where polynyas typically occur but the AEV Index does not directly use environmental data on polynya location or sea-ice state.

We use an algorithm to identify polynyas within gridded data products - either satellite or Earth system models[72]. This algorithm defines polynyas from a threshold value of either SIC or SIT. Polynyas are identified as contiguous regions of grid cells that fall below a given threshold and are surrounded by land or ice-covered regions above the threshold. As discussed in detail by ref. 72, different polynya metrics are needed for modeled and satellite observed sea-ice and we use a threshold of 85% SIC and 0.4 m SIT to identify polynyas in the satellite and Earth system model data, respectively. We then calculate the locations of frequent polynyas by identifying regions where a polynya is identified more than 10% of the year over the relevant time period (Supplementary Fig. 7). Over the satellite record, the locations of frequent polynyas in the CESM2 FOSI correspond well with the satellite data. Similarly, the CESM2-LE polynyas in the 2000s correspond well with the CESM2 FOSI polynyas for both SIC and SIT threshold metrics. When comparing the regions where polynyas typically occur with high value AEV regions we use the model SIT threshold methods for comparison since the typical polynyas match better with the satellite observed typical polynyas.

### Antarctic ecosystem value (AEV) index inputs

The AEV index uses five inputs from different trophic levels in the Antarctic ecosystem to provide a broader metric across trophic levels (illustrated graphically in Fig. 2). This research prioritizes the use of models with reliance on readily available environmental variables produced by standard ESM data, which is available at monthly temporal resolution. While this limits the biological models we can use, it

highlights the models that are capable of linking with an ESM and thus providing a framework for integrating large-scale environmental change to ecological/biological processes. This is a tradeoff that allows the research to simultaneously make broad projections and highlight pathways for additional research to address.

**Net primary productivity (NPP).** This input represents primary producers, the base of the food web. As documented in detail in ref. 22, the Marine Biogeochemistry Library (MARBL) component of CESM2 directly calculates time evolving phytoplankton production for several functional types. The phytoplankton growth rate ($\mu_i$) is a function of the resource-unlimited growth rate ($\mu_{ref}$) at a reference temperature (30 °C), and the temperature ($Lim_T$), nutrient ($Lim_V$), and light availability ($Lim_L$) functions such that:

$$\mu_i = \mu_{ref} \cdot Lim_{(T)} \cdot Lim_{(V)} \cdot Lim_{(L)} \tag{1}$$

The temperature dependence formulation is represented as:

$$Lim_{(T)} = 1.7 \cdot \left( \frac{T - 30°C}{10°C} \right) \tag{2}$$

where $T$ is the temperature.

Light limitation is calculated as a function of irradiance ($I$; W m$^{-2}$), whereby

$$Lim_{(L)} = 1 - exp \frac{-\alpha_i^{Chl} \cdot \theta_i^C I}{\mu_{ref} \cdot Lim_T \cdot Lim_V} \tag{3}$$

where $\alpha_i^{Chl}$ (g C m$^2$ (g Chlorophyll-$a$ W s)$^{-1}$) is the initial slope of the Chlorophyll-$a$- (Chl) specific photosynthesis-irradiance (PI) curve and $\theta_i^C$ is the Chl to C ratio (g Chl:g C), and where the demand for growth is represented as the maximum photosynthetic rate constrained by temperature and nutrient limitation ($\mu_{ref} \cdot Lim_T \cdot Lim_V$).

Nutrient limitation is represented as the minimum limitation term of the N, P, Si, and Fe limitation terms for each phytoplankton functional type. Because phytoplankton can assimilate both nitrate ($NO_3$) and ammonium ($NH_4$), the N limitation term ($Lim_N$) is represented as:

$$Lim_{(N)} = \frac{NO_3/K^{NO_3}}{1 + NO_3/K^{NO_3} + NH_4/K^{NH_4}} + \frac{NH_4/K^{NH_4}}{1 + NO_3/K^{NO_3} + NH_4/K^{NH_4}} \tag{4}$$

where $K^{NO_3}$ and $K^{NH_4}$ are the half-saturation constants for $NO_3$ and $NH_4$, respectively.

While we are aware of the complex dynamics of plankton functional types, we chose to use NPP alone to maintain the metric's broader applicability across different models without requiring detailed assumptions about species-specific responses. NPP is a direct output from CESM2. All configurations of CESM2 calculate phytoplankton growth rates separately for three phytoplankton functional types: diatoms, diazotrophs, and small phytoplankton[22]. The CESM2-FOSI configuration also includes coccolithophores as a functional type[23]. None of these functional types are Antarctic specific, but instead parametrized to broadly capture their global distribution. Sea-ice algae are not included. We use the total NPP summed over all phytoplankton types during the Antarctic growing season (months ONDJFM). As seen in Supplementary Fig. 8, the NPP in both the CESM2-FOSI and CESM2-LE captures the observed spatial patterns of a subpolar NPP maximum that decreases within the seaonal sea ice zone, and then areas of higher productivity along the coast that are aligned with polynyas. We compare the model output to three products: the vertically generalized production model [VGPM;[73]], the carbon-based productivity model [CbPM;[74]], and the carbon, absorption, and fluorescence euphotic-resolving NPP model [CAFE;[75]]. These observational products have a wide range in are estimated NPP magnitude, and the

model estimates are within this range and thus appropriate for this analysis.

**Krill growth potential (KGP).** This input represents primary/secondary consumers. We calculate Antarctic krill (*Euphausia superba*) growth potential over the growing season (months ONDJFM). We use the empirical relationship from[25] that depends only on surface quantities[76]. This KGP metric has been used at a broad scale across the Southern Ocean in several prior studies[43,76–78]. While more complex Antarctic krill models that incorporate biomass dynamics and full life cycle processes could provide a more detailed perspective[36], such models were not used in this study because of the necessity to use existing coupled ESM data which does not have temporal or spatial resolution needed. A more sophisticated regional modeling study of krill in some of the large scale regions identified by this study would be valuable.

This relatively simple relationship includes contributions due to initial length ($KGP_{length}$), food availability ($KGP_{food}$), and sea surface temperature ($KGP_{temp}$):

$$KGP = KGP_{length} + KGP_{food} + KGP_{temp} \quad (5)$$

The length term is calculated as:

$$KGP_{length} = -0.066 + (0.002 * L) + (-0.000061 * L^2) \quad (6)$$

where we use an initial krill length ($L$) of 40mm.

The food term is calculated based on the total surface chlorophyll as follows:

$$KGP_{food} = 0.385 * \frac{Chl_{surf}}{(0.328 * Chl_{surf})} \quad (7)$$

where the total chlorophyll is summed over all phytoplankton functional types. In reality, krill have been observed to be selective feeders and prefer diatoms to other phytoplankton[79]. While the MARBL model has a diatom functional type and it generally dominates productivity in polar regions with CESM2, we continue to use total surface chlorophyll from all phytoplankton in our calculation. We do this for consistency with observations from which the equation was derived since they did not take into account different phytoplankton species. However, this use of total chlorophyll should be considered an upper bound on the food term and could lead to an overestimation of krill growth.

We used the KGP model only in a range of sea surface temperature (SST) of -1 to 5 °C, the temperature range over which the model was derived. Within this range, the temperature term depends SST as follows:

$$KGP_{temp} = (0.0078 * SST) + (-0.0101 * SST^2) \quad (8)$$

and has a peak growth rate when $SST = 0.386 °C$. This reflects that krill are stenothermic, meaning they can live only in a narrow thermal range. Though adult Antarctic krill can survive in waters up to 5 °C, their optimal thermal range lies within 0–3 °C and they experience increasing thermal stress at higher temperatures e.g., ref. 25,42–47.

The empirical relationship for KGP was derived for conditions for adult Antarctic krill in summer and does not account for other seasons or growth phases. While Antarctic krill depend on sea ice concentrations for some life stages (e.g., spawning, larval over wintering[50]), this ice-dependence is not directly taken into account in the KGP modeling. However, because we only use KGP in areas that are within the seasonal sea ice zone, the impact of sea ice availability on other life stages is implicitly included for this input layer. For the AEV Index inputs, CESM2 directly provides the surface chlorphyll and temperature

needed to calculate growth rate. The best estimate of krill habitat of which the authors are aware is that provided by AquaMaps[80]. As seen in Supplementary Fig. 9, KGP calculated from CESM2-FOSI and CESM2-LE data show qualitatively that spatial patterns of maxima in KGP occur in locations where krill are most likely to be found.

**Demersal fish biomass potential (DFP).** This input represents secondary consumers. The Fisheries Size and Functional Type (FEISTY) model is a mechanistic global size- and trait- based model that resolves the structure of fish communities from environmental forcing[26]. The CESM-FEISTY framework has been documented in ref. 23. The FEISTY model used in this analysis was run in an "offline" fashion using CESM2 output, i.e., FEISTY is forced by output from the MARBL biogeochemical ocean model without FEISTY impacting fields in MARBL. Forcing variables include water temperature in the top 100m and at the ocean floor, particulate organic carbon (POC) flux to the ocean floor, and mesozooplankton biomass and loss rates averaged over the top 100m (as with temperature). Since only one generic zooplankton tracer is simulated in CESM2, we derive mesozooplankton biomass and loss rates using the diatom fraction of phytoplankton biomass following[81] and[31].

FEISTY biomass is output as grams of wet weight per square meter ($g\ WW/m^2$) for each functional type: demersal ("bottom-dwelling") fish and pelagic fish (forage fish and large pelagic fish). FEISTY simulates three different size classes of fish, from larval to adult, capturing life cycle transitions from one size class to the next (growth), as well as trophic connections between the fish classes and their benthic and pelagic food resources. Once fish reach the adult size class, they can allocate energy towards reproduction to create more of the larval size class (reproduction).

FEISTY is a global model, and the demersal fish functional type in FEISTY represents all types of demersal fish and not a particular species. In the present-day, Antarctic demersal fish include, but are not limited to, species like Antarctic toothfish (*Dissostichus mawsoni*). We focus on demersal fish biomass potential in the AEV index because Antarctic toothfish are important for fisheries, but FEISTY does not have polar specific modifications to represent this Antarctic species directly. In the absence of considering such Antarctic-specific adaptations, the simulated biomass concentrations in FEISTY should be considered maximum potential biomass. Temperatures above 1 °C become increasingly uninhabitable for Antarctic toothfish[32]. For the DFP used in the AEV Index, we have summed over each size class of the global FEISTY data. Additionally, we only use values where the bottom temperature is ≤1 °C to account for Antarctic toothfish habitat suitability. When we compare the locations of highest DFP from FEISTY with the best estimate of Antarctic toothfish habitat from AquaMaps[80], we see that the spatial patterns of maxima in DFP occur primarily on the continental shelves and continental slopes (Supplementary Fig. 10). These locations are consistent with catch data from East Antarctica of where Antarctic toothfish are likely to be found[82]. The map of FEISTY biomass is also qualitatively similar to statistical predictions of Antarctic toothfish habitat, which shows that Antarctic toothfish can be particularly well predicted by sea ice thickness, ocean temperature, and depth[38].

**Penguin population and accessibility.** We use two input layers representing different species of penguins and their projected population evolution to represent secondary/tertiary consumers. We include the Emperor penguin (*Aptenodytes forsteri*) and Adélie penguin (*Pygoscelis adeliae*), both upper trophic level predators, because these two species experience different future population projection trajectories. In the AEV Index we assign each penguin input layer equal weight as the other layers, though testing shows that using a single penguin layer where each species has a weight of 0.5 results in a similar AEV Index as the results shown here.

The Emperor penguin Population (EPP) accessibility layer is calculated as how many penguins can access each ocean model grid cell by first calculating the distance from each of the 66 Emperor penguin colonies to the center of the model grid cells. Then we calculate how many individuals from a colony can reach each cell by assuming an exponential decay rate of travel from the colony. We determine the decay rate by using an average dispersal distance of 414 km[83] and assume half the individuals from a colony can access that distance. Then we sum how many individuals from all colonies can access any given ocean grid cell. The initial colony population is taken from satellite observations from[84].

The Adélie penguin Population (APP) accessibility layer is calculated as how many penguins can access each ocean model grid cell by first calculating the distance from each of the 287 Adélie penguin colonies to the center of the model grid cells. Then we calculate how many individuals from a colony can reach each cell by assuming an exponential decay rate of travel from the colony. We determine the decay rate by using maximum dispersal distance of 1933 km[85] and assume that 1/1,000,000 of the individuals from a colony can access that distance; we use maximum dispersal distance for Adélie penguins because an average dispersal distance for this species is not known. Then we sum how many individuals from all colonies can access any given ocean grid cell. The initial colony population is taken from satellite observations from[86].

Supplementary Fig. 11 shows the penguin colony locations and sizes estimated from satellite data as well as ocean accessibility, calculated for each species as described above. Detailed information about the Emperor penguin and Adélie penguin metapopulation models are found below. Projected penguin populations for each decade are the average over the five years surrounding each decade. This way we can look at the long term signal in the population rather than possible year-to-year variations. For future colony health, we follow[15] and define penguin colony health as follows: non-threatened' penguin colonies are projected to experience either an increase in population or a decline less than 30%; vulnerable' penguin colonies as those projected to experience a population decline by more than 30%; endangered' is a likely population decline by more than 50%; quasi-extinct' is a population decline by more than 90%. We use the penguin populations from the 2000s as a baseline for the penguin colony health.

### Antarctic ecosystem value (AEV) index calculation

The AEV index identifies the areas of highest value over multiple trophic levels of the Antarctic food web. We calculate AEV Index values for a historical reconstruction that correlates to conditions experienced from 2000–2020 and for a coupled, free running Earth system model that represents changing climate conditions. Supplementary Table 1 lists the sources for each of the five inputs for both the historical reconstruction and the free running AEV Index calculations.

The metric is computed within the seasonal sea-ice zone south of 60°S and separately for the five regions shown in Fig. 1a: Weddell Sea, East Antarctic, Ross Sea, Amundsen Sea, and Antarctic Peninsula. We calculate the AEV Index by region because Antarctic marine protection is considered by individual planning regions, and we aim to identify the most valuable area for protection in each management region. These regions generally correspond to the Marine Protected Area (MPA) planning domains determined by the Commission for the Conservation of Antarctic Marine Living Resources (CCAMLR)[87]: domains 3 and 4 (Weddell Sea), domain 7 (East Antarctic), domain 8 (Ross Sea), domain 9 (Amundsen Sea), and domain 1 (Antarctic Peninsula). For our analysis, we make two important changes to the standard CCAMLR regions. First, we combine planning domains 3 and 4 to form the Weddell Sea region because they both span the Weddell Gyre and the current MPA proposals for the Weddell Sea span both regions. Second, we have shifted the southeastern boundary of the Antarctic Peninsula

domain to include some of the Weddell Sea because there are important Adélie penguin colonies on each side of the CCAMLR domain boundary and we have tried to use regions that are more aligned with the known biological connectivity.

We calculate the AEV Index at each grid point at location $i$ (latitude) and $j$ (longitude) as follows. We scale each of the five inputs by the maximum value over the entire Southern Ocean domain and then sum the scaled inputs:

$$AEVsum_{ij} = \frac{NPP_{ij}}{NPP_{max}} + \frac{KGP_{ij}}{KGP_{max}} + \frac{DFP_{ij}}{DFP_{max}} + \frac{EPP_{ij}}{EPP_{max}} + \frac{APP_{ij}}{APP_{max}} \quad (9)$$

Each input is weighted equally in the sum of inputs. Then, to get the final regional AEV Index, we scale the summed value at each point by the maximum summed value in each of the five analysis regions:

$$AEV_{ij} = \frac{AEV\,sum_{ij}}{AEV\,sum_{max-region}} \quad (10)$$

We also performed several sensitivity studies with the AEV Index calculation. We experimented with scaling the index by the median and mean values, instead of the maximum, and it did not impact the AEV Index results significantly. As a result, we use a maximum value for scaling as it is more directly interpretable. Thus, the AEV Index presented in this analysis has values from 0 to 1, with 1 indicating the grid point(s) with the largest sum and therefore has the highest value over all inputs in that region. It is important to note that an AEV Index value within a particular region is not directly comparable to values outside that region as the calculations and scaling are done separately. To highlight the particularly valuable locations within each region, we have binned the AEV Index into four categories: exceptional (AEV Index in the top 5th percentile for that region), very high (AEV Index in the top 10th percentile for that region), high (AEV Index in the top 25th percentile for that region), and other (all other points). Note that the percentile thresholds for these bins are determined separately for each of the regions and decades (see Supplementary Figs. 12 and 13).

A full comparison of biodiversity metrics around Antarctica is beyond the scope of this study. However, to demonstrate that the historical reconstruction AEV Index - both the hemispheric and regionally scaled versions - provides sensible information about valuable biological areas, we qualitatively compare it to the Areas of Ecological Significance (AES) identified by ref. 12 using in-situ tracking data from 17 marine mammal and bird species to identify regions preferred by these predators. Both the AES and AEV Index metrics identify ecologically valuable regions using multiple species, yet it is important to note that while species overlap between these two metrics, the methodology is entirely different and the metrics do not share any overlapping data. This comparison is done primarily to show that a model based metric to identify valuable biological areas is comparable to a similar metric that uses observational data, but a full analysis comparing multiple Antarctic metrics of biodiversity and biological importance is beyond the scope of this work. We find that both metrics identify similar grid cells as valuable (grid cells in the exceptional, very high, and high value bins) as the AES locations, particularly in the coastal areas of the southwest Ross Sea, parts of the East Antarctic, and in the northern Antarctic Peninsula. However, the regionally weighted AEV Index has more overlap with the AES in sectors beyond the Ross Sea and Antarctic Peninsula, thus bolstering our conclusion that the regional weighting is a better way to identify regionally important high-value areas. In other areas, the results do not align as well, which may be due to differences in data availability. In the Amundsen Sea there is less animal tracking data and it is also an area the AES does not identify as a valuable area while the AEV Index does. In other areas, like the Balleny Islands northeast of the D'Urville station in East Antarctica, the AEV Index does not show these as valuable possibly because we

didn't include species inputs that emphasized this area. Thus, we believe that the AEV Index, which is calculated using data on a coarse (-1°) Earth system model grid, is capable of providing high level information about valuable areas around Antarctica, particularly into the future. It is important to note that additional research and monitoring, including with existing observational datasets such as those used in the AES study, could be conducted to further refine locally valuable regions.

**Penguin metapopulation models**

Two metapopulation models were employed, one for each of the Emperor and Adélie penguins, to forecast the population vector $\mathbf{n}(t)$, where each element represents the total population size at each colony at time $t$. This requires the vector $\mathbf{r}_t$ of intrinsic population growth rates at each colony at time $t$, that is estimated using two approaches: one based solely on Capture-Mark-Recapture (CMR) data and a Structured-non-linear Population model (SPCMR model)[88,89] and the other using nest count data from the Antarctic Penguin Biodiversity Project[90] and a global empirical population model[28,29].

For Emperor penguins, the metapopulation model is also parameterized using new data on dispersal distances, emigration rates, and dispersal behaviors derived from[83,91]. Emperor penguins are most likely to disperse through semi-informed dispersal (informed departure but random settlement), characterized by a relatively small mean dispersal distance of 414 km and low emigration rates[83]. For Adélie the model is parameterized with a mean emigration rate of 0.035 with a semi-informed dispersal assumption.[92] demonstrated that dispersal of breeding Adélie penguins is typically low (<1%), except during years with challenging environmental conditions when movement rates increased, particularly away from the smallest colonies (up to 3.5%). For our analysis, we adopted the highest observed emigration rate (3.5%). Despite incorporating this elevated rate of emigration, the dynamics of the regional and global populations exhibited a trajectory similar to that of a metapopulation without dispersal, suggesting that the higher dispersal rate did not significantly alter overall global and regional population trends for Adélie penguins.

The metapopulation projects the total population size at each colony at time $t$ using:

$$\mathbf{n}(t+1) = \mathbf{D}[t,\mathbf{n}(t)]\mathbf{F}[t,\mathbf{n}(t)]\mathbf{n}(t), \tag{11}$$

to indicate that the projection interval is divided into two main phases of possibly different duration: the motionless reproduction phase ($\mathbf{F}$) followed by the dispersal phase ($\mathbf{D}$). The projection matrices ($\mathbf{F}$) and ($\mathbf{D}$) depend on both the current population density $\mathbf{n}(t)$ and time $t$ because habitat conditions vary among patches and over time. The reproduction matrix $\mathbf{F}$ and dispersal projection matrix $\mathbf{D}$ are detailed in refs. [91,93], but both depend on $\mathbf{r}$, the vector of colony-specific intrinsic population growth rates at each time step. For Adélie penguin $\mathbf{r}$ is calculated using equation B9. For emperor penguin, $\mathbf{r}$ is calculated using

$$r_{i,t} = \log\left(\frac{\sum \mathbf{w}_i(t+1)}{\sum \mathbf{w}_i(t)}\right), \tag{12}$$

where $\mathbf{w}_i(t)$ is a vector of the number of individuals in each stage for colony $i$ at time $t$ calculated from equations B3.

**Emperor penguin model intrinsic growth rates.** We used a nonlinear, two-sex, stochastic, seasonal, and climate-dependent matrix population model developed by refs. [15,88,89] to calculate the intrinsic population growth rate $r_{i,t}$ at each colony $i$ and year $t$. To obtain $r_{i,t}$, we

projected the population vector $\mathbf{w}_i$ as:

$$\mathbf{w}_i(t+1) = \mathbf{A}[E(t),\mathbf{w}_i(t)]\mathbf{w}_i(t), \tag{13}$$

where the projection matrix $\mathbf{A}[E(t),\mathbf{w}_i]$ is determined by vital rates, including survival probabilities, breeding probabilities, breeding success, and mating probabilities, all of which depend on sea ice concentration anomalies ($E(t)$) around each colony. The functional relationships between vital rates and sea ice concentration anomalies were estimated by ref. 88 using the long-term longitudinal data from Pointe Géologie. In addition, $\mathbf{A}[E(t),\mathbf{w}_i]$ is influenced by the current population vector ($\mathbf{w}_i$) due to the mating processes that depend on the proportion of males to females available to mate. The population vector $\mathbf{w}_i$ distinguishes five stages within the population according to breeding status and sex: male and female prebreeders, male and female nonbreeders, and breeding pairs. A prebreeder is an individual who has yet to breed for the first time. A nonbreeder is an individual that has bred at least once before, but is not breeding in the current year. A breeding pair is a female and a male who mate and cooperate over the breeding season to produce offspring. This sea-ice-dependent two-sex stochastic model has been validated to project the emperor penguin population in previous studies[15,16,27,91].

**Adélie penguin model intrinsic growth rates.** To estimate relationships between long-term average colony growth and long-term average sea ice concentration, this study uses the Adélie penguin global population model from[28] and expanded by ref. 29.

To parameterize the model, we used nest count data from the Antarctic Penguin Biodiversity Project[90]. The model fitting uses the 38 colonies that have at least 10 years of counts between 1979-2018, as was done in ref. 29. Sea ice concentration for the same years is from satellite observed passive microwave radiation at a 25 x 25 km resolution[94] and rescaled to a 1-degree resolution. The sea ice concentration for each colony is identified within a 500 km polygon for March to June each year, which corresponds to the 4-month period following the breeding season.

We modeled true logged nest abundance ($lz_{i,t}$) for colony i in year t as:

$$lz_{i,t} \sim N(lz_{i,t-1} + \mu_{r_i}, \sigma_i^2) \tag{14}$$

where $\mu_{ri}$ is the average growth rate for colony i, and $\sigma_i^2$ is process error capturing annual variation in abundance for colony i that is not explained by simple exponential growth. We model $\mu_{r_i}$ as a function of the long-term average sea ice concentration ($\overline{SIC_i}$) surrounding each colony:

$$\mu_{r_i} \sim N(\beta_0 + \beta_1 \times \overline{SIC_i} + \beta_2 \times \overline{SIC_i}^2, \sigma_r^2) \tag{15}$$

where the $\beta$ parameters fit a quadratic relationship between long-term average SIC and average colony growth rate, and $\sigma_r^2$ is process error capturing colony-specific variation in growth rates not explained by its modeled relationship with sea ice. We model $\sigma_i$ hierarchically using a normal distribution truncated at 0 with hyper-parameters $\gamma$ and $\tau$:

$$\sigma_i \sim N^+(\gamma, \tau^2) \tag{16}$$

We model the logged nest count at colony $i$ in year $t$ as:

$$y_{i,t} \sim N(lz_{i,t}, s_{i,t}^2) \tag{17}$$

whose mean is the true logged nest abundance at colony $i$ in year $t$, and where $s_{i,t}^2$ is the observation error associated with the nest count. See ref. 28 for details regarding how the observation error for each nest

count was determined. We chose weakly informative priors for all parameters in the model.

**Projecting long-term populations under climate change.** We used output from the Community Earth System Model version 2 (CESM2)[21] to forecast and hindcast colony growth rates into years 2100 and 1900 (Emperor penguin) or 1939 (Adélie penguin), respectively. CESM2 is a state-of-the-art Earth system model, run at the nominal 1° resolution with ocean biogeochemistry. A thorough description of the ocean ecosystem model in CESM2 is documented in ref. 22.

We used a large ensemble of fully coupled CESM2 simulations (CESM2-LE)[24]. We used 50 members of the CESM2-LE which are all forced by the Coupled Model Intercomparison Phase 6 (CMIP6) forcing following shared societal pathway 3-7.0 (SSP3-7.0) but differ slightly in their initial conditions. The small differences at initialization produce a spread in the timing of natural (internal) climate variability of the model (for further description of initial condition large ensembles see ref. 95).

The coupled model configuration produces a different mean state and variance as compared to observed sea ice data. Thus, to make the future and the past coupled model simulations more comparable to passive microwave observations, we applied a bias correction procedure to each of CESM sea ice concentration values of CESM simulations:

$$\left(\frac{\delta_{obs}}{\delta_{cesm}}\right) \cdot (y_t - \theta_{cesm}) + \theta_{obs} \qquad (18)$$

where, $\delta_{obs}$ and $\delta_{cesm}$ are standard deviations of the observed and CESM sea ice concentration data for the period 1979 to 2018, respectively; $\theta_{obs}$ and $\theta_{cesm}$ are the means of the observed and CESM sea ice concentration data for the period 1979 to 2018, respectively; $y_t$ is the CESM sea ice concentration value of year $t$ for a given member of CESM simulations. We apply this procedure separately for each of the 66 emperor penguin colonies or 287 Adélie penguin colonies that were used for colony growth projections.

For Adélie penguin, because we use 40-year SIC average to build the empirical model explained above, we used 40-year moving window average of CESM data as well. After bias correction we calculated each year SIC data as the average SIC of the preceding 40 years. So, our colony growth projections range between 1939 and 2100 as the earliest CESM projection we used was 1900. We truncated SIC projections that were beyond the data bounds observed between 1979 and 2018 with the maximum or minimum observed SIC. So, none of the projections were extrapolations.

For the Emperor penguin, our metapopulation models SPCMR employed a time-for-space substitution to project population dynamics for other colonies by assuming that the functional relationships between environmental factors and vital rates estimated at Point Géologie are consistent across space, despite potential local adaptive differences or other site-level effects. The model generalizability and predictive performance on new data was conducted by ref. 91 using two approaches: (1) the ability of the model to project year-to-year dynamics at Pointe Géologie; (2) the ability of the model to match the observed global trend and annual fluctuations in emperor penguin abundance between 2009 and 2018.

We compute $\mathbf{A}[\mathbf{w}_i]$ for all colonies and each year by applying the functional relationships between vital rates and sea ice established at Pointe Géologie to the bias-corrected CESM projections $E(t)$ at each colony for each year. From this, we can then calculate $\mathbf{w}_i$ (Equation B3) and $\mathbf{r}$ (Equation B2) for the total population. We performed 100 simulations per colony for each climate projection by 1) uniformly sampling the initial population structure from the $k$-simplex using the broken stick method and 2) sampling the relevant vital rates from their

multivariate normal distribution while incorporating both measurement error and environmental stochasticity through explicit error propagation in the observed and process variance-covariance structure. This resulted in a set of intrinsic growth rate vectors, $\mathbf{r}$, that account for different sources of uncertainty included in our metapopulation model, such as measurement error, parameter uncertainty, natural variability in demographic processes, and uncertainties in initial conditions.

For Adélie penguin, we used the fitted relationship in equation 2 coupled with the bias-corrected CESM projections to project average colony growth rates ($\mu_{r_{i,t}}$) for each colony ($i$) and year ($t$), between 1939 and 2100. Because equation 2 only encompasses spatial gradients in sea ice variability, these projections can be considered as space-for-time substitutions[96]. To employ parameter uncertainty in these projections, we randomly sampled parameters from their respective posterior distributions across 100 iterations. In each iteration, we additionally sampled $\mu_{r_{i,t}}$ from the distribution to employ uncertainty caused by variance in average colony growth unexplained by sea ice. Finally, we projected annual colony growth with spatio-temporal process variance at a given colony and year using:

$$r_{i,t} \sim N(\mu_{r_{i,t}}, \gamma^2) \qquad (19)$$

Where, $r_{i,t}$ is the projected annual growth at colony $i$ and year $t$. This resulted in 100 annual colony growth time series for each colony in each CESM member, a total of 1,445,000 time series.

**Metapopulation projections.** For each metapopulation model, we then projected the annual abundances for all colonies, $\mathbf{n}(t)$, using equation B1 by drawing 1000 simulations randomly from the intrinsic population growth rates projections just discussed to parameterize $\mathbf{F}m$. In addition, for emperor penguins we draw from the posterior distributions of dispersal rates and distances[83] to parameterize $\mathbf{D}m$.

For Emperor penguin projections, we used $N_0$, based on the average population size derived by ref. 84 from satellite data and the stable population distribution computed from average sea ice concentrations during the observation period at each colony[91]. For Adélie penguin projections we used $N_0$, based on the median count for each colony from MAPPD.

### Reporting summary
Further information on research design is available in the Nature Portfolio Reporting Summary linked to this article.

## Data availability
All processed data as well as data used to generate figures and tables is freely available at: (https://doi.org/10.5281/zenodo.14827913). The raw CESM2 Large Ensemble data[97] are freely available at:(https://gdex.ucar.edu/datasets/d651056/). The raw CESM2 FOSI data, originally published by ref.[23]. As specified in that paper, the data are freely available on a Globus Guest Collection: (https://app.globus.org/file-manager?origin_id=6f5e56da-0353-4bd4-bac0-04a104e05d58&origin_path=%2F&two_pane=false). Note that Globus requires users to create a free account for data download.

## Code availability
The data analysis for this manuscript was done using Jupyter Notebooks and Python 3.7.12. All code for data processing and analysis are freely available at: (https://doi.org/10.5281/zenodo.14827913).

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

## Acknowledgements

We acknowledge and appreciate support for this work from the National Aeronautics and Space Administration (NASA), the National Science Foundation (NSF), the U.S. Geological Survey (USGS), and as detailed below. This work uses resources supported by the U.S. National Science Foundation National Center for Atmospheric Research (NSF NCAR) under cooperative agreement 1852977. Computing and data storage resources were provided by the Computational and Information Systems Laboratory (CISL) at NSF NCAR. We thank all the scientists, software engineers, and administrators who contributed to the development of CESM2. We thank Marthe Vienne and Lucie Bourreau, students affiliated with this project. We also acknowledge OnlyOne and John Weller, MAC3 Impact Philanthropies, the Pew Bertarelli Ocean Legacy, and the Antarctic and Southern Ocean Coalition who provided feedback and input. Any opinions, findings, and conclusions or recommendations expressed in this material are those of the authors and do not necessarily reflect the views of NSF or NASA. Any use of trade, firm, or product names is for descriptive purposes only and does not imply endorsement by the U.S. Government. • NASA Award 80NSSC21K1132: AKD, LLL, ZS, CB, SJ • NASA Award 80NSSC20K1289: KMK, LLL, SJ, CCC, BS, MNL • NSF Award 2037531: MMH, LLL • NSF Award 2037561: SJ, AE • US Department of Energy, Grant no. DE-SC0025505: CN • US Geological Survey: CCC • Royal Society Rutherford Discovery Fellowship, Royal Society CSG-UOC2302: MAL • Center national de la recherche scientifique: SL.

## Author contributions

• A.K.D.: Conceptualized the science plan and methodology, performed the analysis and visualizations and wrote and edited the manuscript draft. • K.M.K.: Conceptualized the science plan and methodology, performed the FEISTY simulations and provided biological expertise, provided original biological illustrations, provided feedback throughout the analysis and reviewed and edited the manuscript. • L.L.L.: Conceptualized the science plan and methodology, provided sea ice expertise, provided feedback throughout the analysis and reviewed and edited the manuscript. • Z.S.: Conceptualized the science plan and methodology, provided krill modeling expertise, provided feedback throughout the analysis and reviewed and edited the manuscript. • B.S.: Performed the penguin population projections, provided feedback throughout the analysis and reviewed and edited the manuscript. • SL: Provided feedback on penguin population observations and modeling, provided feedback throughout the analysis and reviewed and edited the manuscript. • C.C.C.: Performed the penguin population projections, provided feedback throughout the analysis and reviewed and edited the manuscript. • A.E.: Performed the penguin population projections, provided feedback throughout the analysis and reviewed and edited the manuscript. • MMH: Conceptualized the science plan and methodology, provided sea ice expertise, provided feedback throughout the analysis and reviewed and edited the manuscript. • M.A.L.: Provided feedback on penguin population observations and modeling, provided feedback throughout the analysis and reviewed and edited the manuscript. • C.N.: Provided feedback on fish and krill modeling, provided feedback throughout the analysis and reviewed and edited the manuscript. • M.N.L.: Performed the FEISTY simulations, provided feedback throughout the analysis and reviewed and edited the manuscript. • S.J.: Conceptualized the science plan and methodology, performed the penguin population projections, and provided feedback throughout the analysis and reviewed and edited the manuscript. • C.B.: Conceptualized the science plan and methodology, provided feedback throughout the analysis, and reviewed and edited the manuscript.

## Competing interests

The authors declare no competing interests.
