## [Transparent Peer Review file · Nature Communications]

An Antarctic ecosystem value index to quantify ecological value across trophic levels and over time

Corresponding Author: Dr Alice DuVivier

Version 0:

Reviewer comments:

Reviewer #1

(Remarks to the Author)

General Comments

The manuscript proposes a novel way to identify ecologically valuable areas in the Southern Ocean, applies this to current and future climate scenarios (up to 2090), and links their geospatial pattern to the occurrence of polynyas. It is novel in that it is the first index to consider multiple trophic levels, and the first to explicitly link them to polynya occurrence. Results highlight distinct areas of high ecological value, some of which continue to exist until 2090. They occur in areas where present and future polynyas occur. Examining area of importance across the ecosystem both now and in the future in this way is relatively novel and the results that indicate many areas of high ecological value don't substantially change in location are noteworthy and potentially surprising given the scale and magnitude of expected environmental changes. The results are clearly underpinned by a substantial set of earth system and biological models. My main criticism of this work is that there is no mention of any validation to determine how well the present-day biological models perform (and by extension future models will presumably perform), particularly in space, which is the basis upon which the AEV Index is built. Thus, it makes it difficult to determine how robust the AEV spatial patterns are. This particularly concerns me for the krill and demersal fish models- 2/5 of the aggregated index.

The underlying biological models would also have uncertainty associated with their predictions, yet this is not presented in the spatial layers of each of the component of the index (i.e. krill growth, demersal fish biomass) and not propagated through to the AEV Index. A sensitivity analysis is conducted on the calculation of the AEV index in terms of whether means or medians etc should be used, but I should imagine these effects would be small for each grid cell compared to propagating biological model/prediction uncertainties.

The penguin population models are well described in the appendix and supplementary material, however the krill growth and the demersal fish biomass models are not. The krill model is stated as an empirical relationship between chl_a and sea surface temperature. The demersal fish are a product of the FIESTY global size-structured model and readers are mostly referred to a reference. More detail is needed about these models to evaluate the utility, without having to read the cited paper. For example, what environmental forcings are used in the FIESTY model. It is a global model- does it have appropriate inputs in the Southern Ocean (many do not). Is this even an appropriate model to use if it doesn't capture the habitat preferences of the two most common and abundant (and in the case of toothfish high biomass) species (as mentioned in the discussion)? Would a species distribution model for the major species be more appropriate? Again, this highlights the importance of some type of validation to help the reader believe the underlying models are robust.

Some additional items for consideration include giving greater detail on some of the reasoning behind decisions made, and improving the discussion of results with regards to other studies and polynya dynamics. We also recommend the authors depict both regionally weighted and hemispheric weighted results to aid the reader in distinguishing regional from circumpolar importance

Having said the above, we do think that the manuscript addresses an important question in a novel way and is a significant body of work- IF we can be convinced that the spatial patterns and predictions of the models that make up the AEV present-day index, in particular are robust.

Specific Comments

Line 121 onwards: Might be worth being a bit more specific with regards to the dynamics, for your less oceanographically-inclined reader. Also are we looking at just coastal polynyas or open water ones also?

Lines 150-151: "Here, we present the Antarctic Ecosystem Value (AEV) Index that identifies regions of high ecological value around Antarctica." Maybe consider adding "based on several key trophic levels" to underscore the novelty of your work early on.

Line 158: Please specify the climate change scenario number, makes it easier for the reader not to have to jump to the methods.

Line 173: Consider adding a sentence to clarify some MPAs have passed and some have been proposed, otherwise might come as a bit of a surprise in Section 2.

Lines 180 onwards: Consider rephrasing to "Areas with the highest AEV Index occur along the coast in each region (Figure 1a). These high value areas are generally regions where there are high values for all trophic levels considered. Specifically, they have high net primary productivity (Figure 1b)..."

Figure 1 a): Why are there two black lines for the border between the Antarctic Peninsula and the Weddell Sea? This applies throughout the paper, does this have to do with the shift of the peninsula region? Could one of them be a dashed black line?

Figure 1 a): Seeing as you used the free-running model going forward, it might be worth depicting this here instead of the historic indices. Also please include text in the legend to indicate the indices have been regionally weighted.

Line 216: What is the free-running AEV version exactly? How is it model-based exactly? Needs explaining in main body of ms.

Lines 218 onwards: Are you describing the hemispheric (not regionally weighted) areas here? When looking at the regionally weighted ones, the Weddell Sea and East Antarctica offshore regions stand out also. There might be value to you discussing both the hemispheric and the regionally weighted approach – both show slightly different things and are useful for different aspects. In which case you could include the hemispheric map in Figure 1.

Table 1: If you are making the comparison of "AEV Index (unitless) inside polynyas", you probably also would need to look at the average AEV Index outside of polynyas.

Table 1: For the sake of statistical interpretation, we suggest adding a boxplot (at least in the supplementary figures) that shows the potential ranges for the values shown in the table.

Table 1: At what projection did you carry out the area % changes? (Was it an equal area projection?) Please specify this either in the legend text or in the methods.

Line 234: If you reference longitudes, please add some reference points to your maps.

Line 237: "The AEV index was calculated without the direct inclusion of environmental variables". This is a little confusing/distracting as environmental data is a key input into the models.

Figure 2: It would be useful to have circumpolar maps oriented in the same way in all figures. This one is different to Figure 1.

Figure 3: If we are comparing how AEVs have changed over time, we probably might also want to adjust the ecological layers to reflect change over time (i.e. 2090-2000).

Figure 4 and associated results: Why were averages only taken over a subset of the area of each domain? How were boundaries decided? What is the effect of the choice of boundaries on the results presented?

Lines 260-262: Could you provide a percentage?

Lines 251-253: Perhaps qualify this a bit more, after all some areas like the Peninsula decrease significantly.

Lines 268-270: Presumably things like iron play a role – do you have any insight how these are related to the polynya areas? There is some literature on individual polynyas (like the Amundsen sea one - <https://doi.org/10.1016/j.gca.2023.10.029>) you could draw on.

Lines 357-360: doesn't make sense that demersal fish biomass increases over time but fish habitat lost for present day populations??????

Lines 401-409: Might want to consider adding a line to say you targeted species across trophic levels that are core ecosystem components in the Southern Ocean and might be good indicators.

Lines 409 onwards: What are the differences of your study to the AES study – some of the Ross Sea Area only becomes highly valuable in the future in the AES study, while yours finds that it stays the same / increases? Why are those? This is briefly touched upon in the methods, please elaborate on this in the Discussion. Also the hemispheric results differ more from the AES study. This could highlight bias in the way tracking data was selected for the AES study, or limitations in hemispheric applicability of your index. Please discuss.

Lines 416-418: Perhaps rephrase to make this recommendation even stronger, something along the lines of: "This MPA is set to expire in 2052. This work provides evidence the MPA would contain valuable regions through to 2090 and should be considered for extension."

Lines 421 onwards: Might be worth citing <https://doi.org/10.1016/j.marpol.2024.106232> that addresses value of Domain 9

Lines 428 onwards: A bit confusing, suddenly coastal and "large-scale" polynyas are described rather than just polynyas. Line 427: Please consider citing some of the sources that proposed "ecological oases" as a concept (I believe it originates from <https://doi.org/10.1371/journal.pone.0023047?>)

Line 433: Was there any patterns or explanations for which polynyas did not have high AEVs?

Line 437-8: What is the hypothesized link between increased carbon flux and demersal fish??? Many demersal species in this region are carnivores.

Lines 444-448: Gives rise to the question of weighting. Do we care more about some species than others? Are there any areas projected to be suitable for Emperors- should they be targeted more for protection? I'm not necessarily saying that the layers should be weighted differently, this depends on the intended use in terms of conservation outcomes.

Discussion starting line 475: While this is an honest discussion around some of the potential issues with demersal fish model, it highlights a significant question around whether this model is actually in any way realistic. Icefish and toothfish are common species and would generally account for a large portion of the biomass. Toothfish are well known to exhibit an ontogenetic shift in habitat with smaller fish in shallow coastal waters and larger/ higher biomass fish in deeper water from around 800m depth. This seems in contrast to the maps presented. Again, it underscores the importance of presenting validation information for the demersal fish model to enable the reader to believe it is robust.

Line 504: Consider adding a sentence or two about where current anthropogenic activities occur (fishing, but maybe also tourism?) Is there overlap with any of the high AEV areas? Any overlap with high AEV areas that will have decreased by 2090 (i.e. where fishing might contribute to an even faster decline)?

Lines 521 onwards: You are not considering reductions in the ice shelves – your models only extend to the shelf break. Is there a way you could extend your analyses to the areas freeing up in the future?

Lines 663-4: States that the AEV was validated against Areas of ecological significance- which comprise areas of sea bird and mammal use. This is not a validation. At best it is a qualitative comparison. The logic that they are similar therefore the AEV is working well if flawed. Firstly, if they are so similar then why do we need the new AEV index? Secondly, a better comparison would be to quantitatively compare your layers for Emperor penguins and Adelie penguins to the corresponding AES layers which are generated from actual data using tracked animals.

Lines 668 onwards: This is true for the regionally scaled AEV, but more significant differences occur when looking at the hemispheric solution.

Appendix B- 1214-1217: The penguin population models seem to already include sea ice variability in population size estimates. Is there circularity in using sea-ice again for the projection of important areas?

Appendix B- 1308-1312: Indicates some validation of the emperor penguin model was conducted, but nothing is reported. Is this a 'good' or fit for purpose model?

(Remarks on code availability)

Coding was completed in Python and I am not proficient in this language.

Reviewer #2

(Remarks to the Author)

(Remarks on code availability)

Reviewer #3

(Remarks to the Author)

This manuscript presents an interesting and timely approach to identifying ecologically valuable areas in the Southern Ocean by creating the Antarctic Ecosystem Value (AEV) Index. While the study provides valuable insights into current valuable areas and potential changes in them under future climate scenarios, I have some methodological concerns that should be addressed before publication. These include mainly a clarification of the regional standardisation on the text to avoid potential misinterpretation, and further explanation on the choices made for the weighting of ecosystem components. The weighting as is overrepresents penguins, and a justification or adjustment of the model is necessary. I think the authors do a good job on the discussion addressing some of their model limitations, but don't mention the lack of other components of the food webs of the Southern Ocean, or why penguins are so important. Addressing these issues would strengthen the ecological foundation of the study and improve its utility for conservation planning in the Antarctic region.

General comments

I think this study will be of relevance to Southern Ocean studies, and it's novel enough compared to previous attempts at creating similar indices. The code provided with the article is well documented. I think the work supports most of the conclusions, except for what I raise below.

Specific comments

Lines 164-166: "The AEV Index is useful for three reasons. First, it seeks to comprise a holistic ecosystem value by focusing on Antarctic-specific species and integrates across trophic levels (see 28)."

See my comments below, but calling it a "holistic" ecosystem value may be too generous when you're missing some key taxonomic groups (and by consequence overestimating the importance of the ones included) that are very important in regional and global carbon and nutrient cycles.

Lines 156-159: "We use data from an Earth system model (ESM) that includes marine biogeochemistry [19, 20] for both a historical reconstruction [21] and coupled future projection with upper-middle CO2 emissions [22]."

Reading this, the methods, and looking at Table 1, you have what it looks like two historical periods; one that you can historical in table 1, and another you name as "2000s". I understand one is based on historical reconstructions and the other is model-based and used for future projections, but it's not very clear on the methods which one is which and why you need both. Please make it clearer on the text thinking of the average reader that may not be used to modelling and future projections. This also applies to Table A1: understand the Earth System Model includes the future projections, but I think both the table and the text in general would benefit from clarifying or identifying that the ESM includes future projections. Lines 268-270: "In all regions total net primary productivity increases over time (Figure 4b), likely because ice loss reduces phytoplankton light limitation and extends the growing season." If NPP is coming from the ESM, does the ESM specifically model the relationship between sea ice loss and phytoplankton productivity? While this mechanism is plausible, it would be helpful to confirm whether the model actually simulates these specific processes (light limitation changes due to ice loss) or

if this is an interpretation of the results. If the model doesn't explicitly track light limitation, there could be alternative explanations for increased NPP.

Fig. 2 "a) Model based 2000s locations of high (pink) and exceptional (red) Antarctic Ecosystem Value (AEV) Index indicating ecologically valuable regions"

The regional normalization approach used for the AEV Index creates a potential misinterpretation risk that I think would benefit from being addressed, at least in this figure. By normalizing values within each planning region rather than across the entire Southern Ocean, areas labelled as "exceptional value" in different regions are not directly comparable. For example, an "exceptional value" area in East Antarctica may have significantly lower absolute ecological importance than an "exceptional value" area in the Ross Sea or Antarctic Peninsula. So I would recommend clearly stating in the figure legend and maybe more clearly in the main text that the AEV Index values represent relative importance within each planning region only, not across the entire Southern Ocean. I do understand the reasoning behind this standardization, but making it clear for the interpretation of your results doesn't hurt.

Lines 588-590: This line mentions how CESM2 does not represent specific Antarctic phytoplankton species. It would be nice to know which phytoplankton communities are not included, and how much they are expected to contribute to NPP. I'm not overly familiar with CESM2 and I think a lot of people will not know the specific details, so possibly just a sentence here clarifying how this could impact estimations of NPP would be nice. For instance, does the model capture ice algae communities? Understanding these limitations would help readers better interpret the NPP projections.

Lines 645-655: You gave equal weighting to all five metrics (NPP, KGP, DFB, EPP, APP), and I want to understand the reasoning as this may overrepresent certain trophic levels while undervaluing others. There's two penguin-specific metrics (EPP and APP) but only one primary productivity metric (NPP), which means that the index assigns 40% of the weighting to penguins while allocating just 20% to the trophic level that supports the entire ecosystem, from my understanding of it. This imbalance may distort the identification of high-value areas by overemphasizing regions important to penguins relative to their ecological significance in the broader Antarctic food web. I would like to at least see a justification in the text on why penguins as top predators would justify being 40% of the index. Additionally, the model excludes other key top predators such as whales and seals, which play critical roles in Southern Ocean ecosystem functioning. Their absence further complicates the potential bias created by the penguin-heavy weighting. If it's not possible to justify this weighting, then I would recommend implementing a more balanced weighting of the different trophic components.

(Remarks on code availability)

I haven't tried running the code as it's Python and I'm not super familiar with Python coding, but there's a README file and the code is in a Jupyter notebook and well commented.

Reviewer #4

(Remarks to the Author)

This manuscript introduces a new index to evaluate the value of Antarctic marine ecosystems called the Antarctic Ecosystem Value (AEV) that merges ecosystem information across food web trophic levels. With this index, high value regions in Antarctica were identified for current conditions and for future projections to 2090. They found areas of reduced sea ice (polynyas) to be particularly valuable. They deemed areas that continue to be valuable into the future especially important to preserve. These areas were largely within MPAs, but also in proposed MPA areas or areas not in MPAs, which led to the recommendation to develop protection plans for valuable but currently unprotected regions. The manuscript was well-written and uses the best available models and data to inform the index. I agree that the metrics included in the index together form an important and informative collection to evaluate the value of regions in Antarctica.

Since this is an introduction of a new index, I would like to see presented as part of the results that the index works. I suggest moving the model validation section into the result section with figures and statistical output, and provide a compelling conclusion from this that the index does indeed correspond with quality habitat/high value ecosystems. The same with the AEV ranges, allow the reader to conclude that the ranges and cut-offs from good to excellent are logically placed. Now that readers have the information to decide if the index can be trusted, results from the index, based on and future conditions, can be presented.

Minor note on page 5 and on: The hyperlink to S1 doesn't take you to the S1 figure, but to page 5.

(Remarks on code availability)

Reviewer #5

(Remarks to the Author)

This manuscript provides an aggregated index for considering productivity and accessibility for two predator species at a circumpolar scale in the Southern Ocean. The authors calculate this index for historical, present, and future climate conditions, using output from Earth System Model simulations. The index shows a strong relationship with the location of Antarctic polynyas, which generally persists under future climate conditions.

This work is ambitious and is a useful demonstration of an approach to aggregate information about net primary productivity, krill growth potential, demersal fish biomass potential, emperor and Adelie penguin populations, and access to areas for foraging. The authors should be congratulated on the volume of work underpinning the manuscript and the clear

presentation of results in figures and tables.

We have several concerns regarding the use of this index to consider Antarctic ecosystem value more generally, as well as the methods for calculating krill growth potential and demersal fish biomass potential. Large-scale ecosystem processes and connectivity, together with the key food webs roles of zooplankton groups and mesopelagic fish, are all critical components of ecosystem value in the Southern Ocean that underpin decisions about spatial protection, but are not discussed by the authors or considered as part of the index. There is little recognition that groups other than Antarctic krill drive energy transfer, with their relative influence differing across regions – for example, East Antarctic food webs show a higher number of alternative energy pathways between primary production and large predators (e.g., McCormack et al., 2020, 2021, doi: 10.1002/ece3.7017 & doi: 10.3389/fevo.2021.624763). This has broader implications with respect to the potential use of this index to inform marine spatial protection in the Southern Ocean, because (as we are sure the authors will be aware) Antarctic marine protected areas need to be of sufficient scale to protect ecosystem processes – not just small-scale high productivity areas. Equally, marine spatial protection is important for areas that are at risk of degrading in value due to climate change, because enhanced resilience (through reduction in other threats) provides the best chance of ecological processes and values persisting in these areas.

With respect to the methods used by the authors to calculate krill growth potential, we are concerned about the use of the 2006 Atkinson model, which is not designed to address the effects of climate change on physiology, energetics, and fecundity of krill, which are the primary factors governing productivity. The authors do discuss other krill models briefly, but indicate that “they are hindered by substantial uncertainties regarding krill ecology and their interactions with environmental drivers”. It is unclear to us which specific uncertainties prevent the use of more recent and improved krill models in this study. As it stands, the use of a simple empirical relationship between temperature, surface chlorophyll, and krill growth needs to be much more strongly caveated – with implications for interpretation of results explained; the equation for calculating krill growth should also be included for clarity. Consideration, or a caveat, should be given to the variety of krill diets and their dependence on food items other than diatoms, even during the productive summer months. The aggregation of all phytoplankton groups from the NPP model to simulate diatoms is potentially an oversimplification and might lead to an overestimation of krill growth, as krill have been shown to be quite selective feeders (Pauli et al., 2021 <https://doi.org/10.1038/s42003-021-02581-5>).

We are also concerned about the use of projections for demersal fish biomass based on a global model that does not account for the unique physiology of Antarctic fish. In lines 475-485 the authors clearly state that the projected changes for demersal fish are not consistent with expected impacts of temperature increases on key Antarctic species. Future increases in demersal fish biomass due to southward expansion of subpolar and temperate species (presumably not subtropical as the authors state in line 483?) are highly uncertain. As such, we think that the fish projections used in the index are misleading.

Finally, the usage of the climate model CESM2-LE should be justified in light of its inclusion in an ecosystem index. For example, the sea ice-ocean model ACCESS-OM2 has been specifically reviewed for its application in ecological studies (Fierro-Arcos et al., 2023 <https://doi.org/10.1016/j.pocean.2023.103049>). Were other ESMs considered?

Specific comments:

1. Line 146 – please see Cavanagh et al. (2021, doi: 10.3389/fmars.2020.615214), who consider more than two trophic levels and have a Southern Ocean focus.
2. Lines 216-217 – it is unclear to us how Figure S1 compares the historical reconstruction with the model-based index. Could the authors please explain how this comparison is being made?
3. Methods – the inclusion of equations within the bullet points in the AEV index methods section, on top of the written explanation, would make it clearer for the readers to follow. It would also be useful to see some form of sensitivity analysis regarding which of the inputs for the index is most important in driving the overall value of the index over time.
4. Figures: Fig. 1 side panels are very hard to read – making them larger would be very helpful. Fig. S1 refers to Fig. 1, but that is not immediately clear; including the historical reconstruction in S1 would make the comparison much easier. Fig. 2: Interestingly, a large swath of East Antarctic waters is considered to be of Medium value. However, no explanation is given in the text; what could be driving this?

--

This manuscript was co-reviewed by and early career researcher and an established researcher.

(Remarks on code availability)

Code not provided - only processes data are available at the link provided.

Reviewer #6

(Remarks to the Author)

(Remarks on code availability)

Code was not included in manuscript.

Version 1:

Reviewer comments:

Reviewer #1

(Remarks to the Author)

The authors have modified analyses, produced new figures, and substantially re-written large parts of the manuscript to take into account reviewers' comments. This has involved a large amount of work. I believe it is a much-improved manuscript, and the authors should be congratulated.

A few remaining comments to note. Any line numbers refer to the file with tracked changes.

Fish biomass model: Outputs from the global FIESTY model have now been constrained and taken to represent Antarctic Toothfish by truncating potential suitable habitat at <1deg C. Validation is qualitative based on Aquamaps which itself is based on the environmental envelope of occurrence data. We still do have some concerns over how well this model actually represents Antarctic demersal fish. To determine this would require validation with CCAMLR fisheries data, which is probably not in scope of this study.

In a related point, Line 511-513 states that there are no SDMs for Antarctic Toothfish. Actually, a model linking Antarctic Toothfish occurrence to environmental data has been published:

<https://pmc.ncbi.nlm.nih.gov/articles/PMC10984185/>

It shows some similarities and differences to the one presented here.

Line 915-916: Actually, Antarctic Toothfish are caught commercially not only on the shelf, but in high abundances on the slope 1000-1700m. So hopefully the temperature limits used in this ms don't constrain fish biomass to just the shelf. <https://doi.org/10.1016/j.fishres.2019.105338>

Comparison to Areas of Ecologically Significance, Lines 1031-1040: This comparison is interesting in it's own right and some of the species in this index and the AEV overlap, but I still don't think it is a validation/justification for a regionally weighted index. Where the indices are consistent and different are interesting points for discussion (potentially in the main ms rather than justifying the index in the supplemental material).

Minor points:

Line 267: The 30-70% higher should include 'average'. Looking at figure 3a, while there is a consistent trend for the average AEV index to be higher inside polynyas than outside, the variability inside polynyas is also much higher than outside. This should be acknowledged.

Line 314: The decline in AEV in the Ross Sea seems very minimal in Figure 4g.

Line 378-9: Prydz Bay does not appear to have a dramatic decline in projected toothfish biomass in Figure 6i), however the West Antarctic Peninsula does.

Line 452-453: This statement needs referencing "More mesozooplankton biomass and increased POC flux to the benthos would provide more prey resources for demersal fish".

Line 529-532: Regarding the criticism of krill dynamic models, this is at least equally applicable to the parameterisation of the *global* FIESTY model being used to represent Antarctic demersal fish and yet it is part of the AEV index!

(Remarks on code availability)

Code is extensive, but I can't evaluate as I am not a python user.

Reviewer #2

(Remarks to the Author)

(Remarks on code availability)

Reviewer #3

(Remarks to the Author)

I appreciate the revised version of the manuscript and the substantial work the authors have undertaken to address reviewer concerns and improve the manuscript. The main issues I raised have been appropriately addressed, and I am pleased with the improved manuscript.

This work represents a valuable contribution to our understanding of climate change impacts on Antarctic ecosystems, a research area that, as the authors correctly note, remains underrepresented in the literature (due to a lack of data that the authors have also faced). The study holds significant conservation value, and the revisions effectively highlight existing knowledge gaps while addressing limitations in the proposed index. I particularly appreciate that the authors now clearly identify where their index may miss information, as this transparency not only demonstrates the robustness of their approach in certain contexts but also highlights important gaps that can inform future research priorities.

I have only two minor editorial suggestions:

Line 168 (clean version): The abbreviation "KGP" appears here without prior definition. Please introduce and define this term when it first appears in the text.

Lines 173-175: Please specify which planning regions are included in "other planning regions." While readers could reference the figure, much of this paragraph discusses supplementary materials that many readers will not examine in detail. For clarity and to avoid relying on readers' memory of figure captions, particularly given the introduction of an "others" category in your index, please name these regions explicitly in the text.

(Remarks on code availability)

Again, I am not proficient in Python code but the code is well annotated with a README file.

Reviewer #4

(Remarks to the Author)

The authors have made significant changes to the manuscript and have addressed my suggestions and concerns. I now recommend publication of this manuscript.

(Remarks on code availability)

I have checked the link provided for the code and the code is made available. I am not proficient enough in python to provide an evaluation of the code itself. It looks well-organized and provides a README file.

Reviewer #5

(Remarks to the Author)

We commend the authors for thoroughly responding to a large set of review comments and note that the revisions have improved the rigour of the manuscript. Parts of the methodology have been modified following the initial review, including improvements in the treatment of fish habitat to reflect polar fish requirements, validation of the sea ice environmental variable and a sensitivity analysis.

Major points:

a) While the authors have much more clearly caveated the scope of their Antarctic Ecosystem Value Index in the revised manuscript – including an explanation of their use of the term “valuable” at the end of the Introduction and a new subsection of the Discussion on “Gaps in Ecosystem Representation” – we feel that this scope is still overstated in some parts of the manuscript. Notably, in the abstract line 059 “biologically valuable hot spots for the Antarctic ecosystem” would be more accurately stated as “biologically valuable hot spots for the groups considered, and potentially for the ecosystem more broadly” (as this has not been assessed in the study), and at the beginning of the Discussion where the authors claim they have answered the question “Where are the most ecologically valuable marine areas around Antarctica and will these locations change in the future” we do not think is accurate for the reasons outlined in our previous review (i.e. key trophic groups are excluded from the index).

b) Both in the revised manuscript and in the response to review comments the authors state that they “are unaware of any other species for which that models linking environmental conditions to the occurrence of a species is available and which we could include in these calculations at this time” (note there seems to be a slight issue with the formulation of this sentence). We think it is important that the authors provide an explanation as to why they are unable to use other available models, for example SEAPODYM (micronekton) (Green 2022), KRILLPODYM (Green et al. 2023), or MICE models available for baleen whales (Tulloch et al. 2017, 2019). If a set of criteria have been used to determine model suitability for the index could the authors please specify these? There is of course no problem with making particular choices about model suitability – such as using habitat suitability models rather than mechanistic models for krill and fish (but metapopulation models for penguins), but we think this needs to be explained in the Methods.

c) If policy recommendations are being included in the Discussion (Section 3.2) then we think it is important that the authors make it clear that this is an initial/exploratory study in the application of the AEV Index. While much effort has clearly been put into this manuscript, there are multiple key ecosystem processes that are not captured by the approach and choice of models (e.g. krill growth model not capturing sea ice conditions and the key trophic roles of groups such as mesopelagic fish, (McCormack et al. 2020)).

Minor points:

a) Lines 605-606: “we were able to investigate two important environmental variables known to impact Antarctic krill: SST and sea ice availability”. The sentence suggests that there is a modelled linkage between krill and sea ice, which is not the case as lines 597-598 state that “(krill growth potential) reflects adult krill growth and uses only surface chlorophyll and sea surface temperature (SST) as inputs”. It should be clarified that the modelled krill metric is not affected by sea ice, with appropriate limitations/caveats about result interpretation and potential future effects of sea ice decline (also see Meyer et al. 2017 for more details about krill habitat usage that might be relevant to the effects of sea ice loss on krill populations).

b) Line 857 (KGP methods): Initial krill length (L) is specified as 40cm – is this correct?

References

Green, DB (2022). The missing link : pelagic prey field prediction for Southern Ocean predators. University of Tasmania. Thesis. <https://doi.org/10.25959/23973123.v1>

Green et al. (2023) KRILLPODYM: a mechanistic, spatially resolved model of Antarctic krill distribution and abundance <https://www.frontiersin.org/journals/marine-science/articles/10.3389/fmars.2023.1218003/full#B52>

McCormack SA, Melbourne-Thomas J, Trebilco R, Blanchard JL, Raymond B, Constable A. Decades of dietary data demonstrate regional food web structures in the Southern Ocean. *Ecol Evol.* 2021; 11: 227–241. <https://doi.org/10.1002/ece3.7017>

Meyer, B., U. Freier, V. Grimm, J. Groeneveld, B. P. V. Hunt, S. Kerwath, R. King, C. Klaas, E. Pakhomov, K. M. Meiners, J. Melbourne-Thomas, E. J. Murphy, S. E. Thorpe, S. Stammerjohn, D. Wolf-Gladrow, L. Auerwald, A. Gamp, L. Halbach, S. Jarman, S. Kawaguchi, T. Krumpfen, G. Nehrke, R. Ricker, M. Sumner, M. Teschke, R. Trebilco, and N. I. Yilmaz. 2017. The winter pack-ice zone provides a sheltered but food-poor habitat for larval Antarctic krill. *Nature Ecology & Evolution* 1:1853–1861.

Tulloch VJD, Plagányi ÉE, Matear R, Brown CJ, Richardson AJ. Ecosystem modelling to quantify the impact of historical whaling on Southern Hemisphere baleen whales. *Fish Fish.* 2018; 19: 117–137. <https://doi.org/10.1111/faf.12241>

Tulloch VJD, Plagányi ÉE, Brown C, Richardson AJ, Matear R. Future recovery of baleen whales is imperiled by climate change. *Glob Change Biol.* 2019; 25: 1263–1281. <https://doi.org/10.1111/gcb.14573>

(Remarks on code availability)

Apologies we had previously missed the link provided for the Python code. We have not reviewed the code as we are not experts in Python coding.

Reviewer #6

(Remarks to the Author)

(Remarks on code availability)

Thanks to all six reviewers for your helpful comments and suggestions. We have addressed all concerns and responded to each reviewer in detail. Our response text is in blue and all line numbers in our responses refer to the revised manuscript. We include the page references, below, to help reviewers navigate our response since it is long.

Reviewer #1 (Remarks to the Author):	1
Reviewer #2 (Remarks to the Author):	21
Reviewer #3 (Remarks to the Author):	22
Reviewer #4 (Remarks to the Author):	30
Reviewer #5 (Remarks to the Author):	32
Reviewer #6 (Remarks to the Author):	46

Reviewer #1 (Remarks to the Author):

General Comments

The manuscript proposes a novel way to identify ecologically valuable areas in the Southern Ocean, applies this to current and future climate scenarios (up to 2090), and links their geospatial pattern to the occurrence of polynyas. It is novel in that it is the first index to consider multiple trophic levels, and the first to explicitly link them to polynya occurrence. Results highlight distinct areas of high ecological value, some of which continue to exist until 2090. They occur in areas where present and future polynyas occur. Examining area of importance across the ecosystem both now and in the future in this way is relatively novel and the results that indicate many areas of high ecological value don't substantially change in location are noteworthy and potentially surprising given the scale and magnitude of expected environmental changes.

Response:

Thanks for the thoughtful review and comments. We appreciate your recognition of the novelty of the work and some of the surprising results.

The results are clearly underpinned by a substantial set of earth system and biological models. My main criticism of this work is that there is no mention of any validation to determine how well the present-day biological models perform (and by extension future models will presumably perform), particularly in space, which is the basis upon which the AEV Index is built. Thus, it makes it difficult to determine how robust the AEV spatial patterns are. This particularly concerns me for the krill and demersal fish models- 2/5 of the aggregated index.

Response:

Thanks for pointing out this gap in the methods. We have substantially expanded the methods section to explain and validate the Antarctic sea ice as well as different AEV Index input layers

in detail. Please see Online Methods Section 4.3 in the revised manuscript (we have not pasted the whole section in this response since the revised text is several pages long).

The underlying biological models would also have uncertainty associated with their predictions, yet this is not presented in the spatial layers of each of the component of the index (i.e. krill growth, demersal fish biomass) and not propagated through to the AEV Index. A sensitivity analysis is conducted on the calculation of the AEV index in terms of whether means or medians etc should be used, but I should imagine these effects would be small for each grid cell compared to propagating biological model/prediction uncertainties.

Response:

Thanks for pointing out this gap in our discussion. Please see Discussion Section 3.4 in the revised manuscript that focuses on model uncertainties (we have not pasted the whole section in this response since the revised text is several pages long). Additionally, we have also included standard deviation in addition to averages (e.g. Revised Figure 3a, Figure 7, Figure S5, Table 1, Table 2) in the manuscript to quantify uncertainties where it is possible to do so.

The penguin population models are well described in the appendix and supplementary material, however the krill growth and the demersal fish biomass models are not. The krill model is stated as an empirical relationship between chl_a and sea surface temperature. The demersal fish are a product of the FIESTY global size-structured model and readers are mostly referred to a reference. More detail is needed about these models to evaluate the utility, without having to read the cited paper. For example, what environmental forcings are used in the FIESTY model. It is a global model- does it have appropriate inputs in the Southern Ocean (many do not). Is this even an appropriate model to use if it doesn't capture the habitat preferences of the two most common and abundant (and in the case of toothfish high biomass) species (as mentioned in the discussion)? Would a species distribution model for the major species be more appropriate? Again, this highlights the importance of some type of validation to help the reader believe the underlying models are robust.

Response:

As mentioned above, we have substantially expanded the methods section to explain and validate the Antarctic sea ice as well as different AEV Index input layers in detail. Please see Online Methods Section 4.3 in the revised manuscript, where we now include equations for the calculation of NPP, KGP, and explain what inputs the FESITY model needs.

In the expanded methods section we explain that FEISTY is a global model and the input variables. FEISTY provides estimates of biomass for three types (demersal, forage, and large pelagic fish) over three size classes, as illustrated in the diagram, below, from Krumhardt et al. 2024. The authors are unaware of any species distribution models specific to Antarctic fish. Thus, while FEISTY is not designed for Antarctic fish species, we believe it is the best current estimate of possible fish biomass for the present day and future. To account for habitat suitability

of Antarctic toothfish (an example of an Antarctic demersal fish species), we have now masked the Demersal Fish Biomass Potential input layer by ocean bottom temperatures and only use values where the ocean temperature is $<1^{\circ}\text{C}$, which is an upper limit on habitat suitability for Antarctic toothfish (Cheung et al. 2008). In Revised Figure S10 (copied below), we qualitatively compare the locations where FEISTY has highest possible biomass against the Aquamaps estimate of Antarctic toothfish habitat (Kaschner et al. 2019) and it compares well. While adequate observations of Antarctic toothfish biomass are missing to quantitatively evaluate the simulated biomass fields in FEISTY, this assessment gives us confidence in the simulated spatial distribution and thus in using these fields as an input layer for the AEV index.

Figure from Krumhardt et al. 2024

PUBLISHED DATA FIGURE REDACTED

Revised Figure S10

a) Aquamaps Toothfish Habitat

b) Demersal Fish from CE5M2-FOSI

c) Demersal Fish from CE5M2-LE

Some additional items for consideration include giving greater detail on some of the reasoning behind decisions made, and improving the discussion of results with regards to other studies and polynya dynamics. We also recommend the authors depict both regionally weighted and hemispheric weighted results to aid the reader in distinguishing regional from circumpolar importance

Response:

As the reviewer suggests, we now include both the hemispheric and regionally weighted results in the paper. Please see Results Section 2.1 in the revised manuscript that explains in detail why we focus on the regionally weighted AEV Index version for further analysis (we have not pasted the whole section in this response for brevity).

Revised Figure 1

Having said the above, we do think that the manuscript addresses an important question in a novel way and is a significant body of work- IF we can be convinced that the spatial patterns and predictions of the models that make up the AEV present-day index, in particular are robust.

Response:

Thank you to the reviewer for your feedback and support of the manuscript. You provided very good suggestions that have improved the manuscript substantially.

Specific Comments

Line 121 onwards: Might be worth being a bit more specific with regards to the dynamics, for your less oceanographically-inclined reader. Also are we looking at just coastal polynyas or open water ones also?

Response:

We have specified at this line that we are looking only at coastal polynyas. While the model can identify open ocean polynyas, they are generally less common and not the focus of this analysis. Line 79: “Coastal polynyas are regions along the Antarctic coast with reduced sea-ice.”

Lines 150-151: “Here, we present the Antarctic Ecosystem Value (AEV) Index that identifies regions of high ecological value around Antarctica.” Maybe consider adding “based on several key trophic levels” to underscore the novelty of your work early on.

Response:

We have modified this sentence as you suggest:

Line 111: “Here, we present the Antarctic Ecosystem Value (AEV) Index based on several key trophic levels.”

Line 158: Please specify the climate change scenario number, makes it easier for the reader not to have to jump to the methods.

Response:

We have included the scenario number here.

Lines 117-121: “We use data from an Earth system model (ESM) that includes marine biogeochemistry [19, 20] for both a historical ice-ocean reconstruction [21] and fully-coupled, free-running model that has a historical period and future projection with upper-middle CO2 emissions (scenario SSP3-7.0) [22].”

Line 173: Consider adding a sentence to clarify some MPAs have passed and some have been proposed, otherwise might come as a bit of a surprise in Section 2.

Response:

We have modified the text as you suggested here and in the abstract.

Line 64: “This study also shows that while many high-value AEV Index areas are within existing or proposed Marine Protected Areas (MPAs),...”

Lines 140-141: "...suggesting that they may be optimal for inclusion within existing and proposed MPAs."

Lines 180 onwards: Consider rephrasing to "Areas with the highest AEV Index occur along the coast in each region (Figure 1a). These high value areas are generally regions where there are high values for all trophic levels considered. Specifically, they have high net primary productivity (Figure 1b)..."

Response:

We have modified the text as you suggest and believe that the new wording is clearer.

Lines 179-182: "The most valuable areas occur along the coast (Figure 1d) and are generally places where there are high values across trophic level inputs to the AEV Index (Figure S2). Specifically, they have high net primary productivity (Figure 2a) and krill growth potential (Figure 2b)."

Figure 1 a): Why are there two black lines for the border between the Antarctic Peninsula and the Weddell Sea? This applies throughout the paper, does this have to do with the shift of the peninsula region? Could one of them be a dashed black line?

Response:

Thanks for pointing this out. There was an issue in the plotting scripts that caused this artifact. We've fixed all figures with the regional divisions so that this is corrected. See Revised Figure 1b (pasted above) as an example.

Figure 1 a): Seeing as you used the free-running model going forward, it might be worth depicting this here instead of the historic indices. Also please include text in the legend to indicate the indices have been regionally weighted.

Response:

We have modified the results section significantly so that the "Present-day" section focuses on the historical reconstruction AEV Index, including comparison of the hemispheric and regional weighted indices, but does not present results directly from the free-running model. Instead, the free-running model derived AEV Index is presented only in the "Future AEV Index" section of results. We have added that we are showing regionally weighted AEV Index results to figure legends.

Line 216: What is the free-running AEV version exactly? How is it model-based exactly? Needs explaining in main body of ms.

Response:

We now include a paragraph explaining how the free-running AEV Index differs from the historical reconstruction.

Lines 720-729: "The historical reconstruction AEV Index uses data from an ice-ocean model forced by atmospheric observations to reconstruct observed ocean and sea ice conditions

[23]. On the other hand, an individual fully-coupled, free-running model simulation should not exactly replicate observations, since observations represent one possible response of the climate system to given external forcing in the presence of internal climate variability. Instead, to ensure the AEV Index derived from the fully-coupled, free- running model data [24] is robust, we compare the historical reconstruction (average for 2000-2020 conditions) with the fully-coupled simulation data for the 2000s (average over 50 ensemble members from 1998-2002) to ensure they produce similar results in a similar climate state.”

Lines 218 onwards: Are you describing the hemispheric (not regionally weighted) areas here? When looking at the regionally weighted ones, the Weddell Sea and East Antarctica offshore regions stand out also. There might be value to you discussing both the hemispheric and the regionally weighted approach – both show slightly different things and are useful for different aspects. In which case you could include the hemispheric map in Figure 1.

Response:

We now include results comparing the hemispheric and regionally weighted AEV Index using the historical reconstruction data - Please see Results Section 2.1 in the revised manuscript (we have not pasted the full text in this response for brevity). We also include the hemispheric index in Revised Figure 1 (pasted above, in this response).

Table 1: If you are making the comparison of “AEV Index (unitless) inside polynyas”, you probably also would need to look at the average AEV Index outside of polynyas.

Table 1: For the sake of statistical interpretation, we suggest adding a boxplot (at least in the supplementary figures) that shows the potential ranges for the values shown in the table.

Response (to previous two comments):

We now include average AEV Index values inside and outside polynyas in Revised Table 1 (pasted below) and Table 2, as well as in Revised Figure 3a (pasted below). We also include standard deviation in these to provide information about the variability and therefore uncertainty in the values.

Revised Table 1 (example):

Table 1 Historical Reconstruction regional Antarctic Ecosystem Value (AEV) Index information related to polynyas (poly.) and marine protected areas (MPAs)

Region ¹	WS	EA	RS	AS	AP	Hemi.
AEV Index						
avg. inside poly.	0.52	0.61	0.73	0.52	0.52	0.43
avg. outside poly.	0.38	0.43	0.34	0.36	0.31	0.28
std. inside poly.	0.24	0.24	0.42	0.24	0.29	0.23
std. outside poly.	0.06	0.08	0.13	0.08	0.11	0.08
% difference ²	+31	+35	+72	+36	+50	+42
% Total Inside Poly.						
Regional area	3.0	6.7	3.6	4.8	2.4	4.3
Exceptional ³ area	38.2	58.7	45.5	52.1	14.8	1.3

Revised Figure 3:

Table 1: At what projection did you carry out the area % changes? (Was it an equal area projection?) Please specify this either in the legend text or in the methods.

Response:

We are using the native model grid to calculate the percent of the areas within given regions. On the native model grid we know the area of each grid cell (this is essential in an ESM so we can ensure accurate coupling where mass and energy are conserved). Thus, we're able to exactly calculate the area percentages within a given region.

Line 234: If you reference longitudes, please add some reference points to your maps.

Response:

We have added longitudes to all spatial plots for reference. See Revised Figure 1 (pasted above) as an example.

Line 237: “The AEV index was calculated without the direct inclusion of environmental variables”. This is a little confusing/distracting as environmental data is a key input into the models.

Response:

We meant that the AEV Index uses only input from biological models, which themselves include environmental data. We have removed this line in the re-written results section as it was unclear and the section has changed substantially.

Figure 2: It would be useful to have circumpolar maps oriented in the same way in all figures. This one is different to Figure 1.

Response:

We have ensured all polar stereographic maps are oriented the same way with the top center longitude of 0°, which is labeled on all maps (Please see Revised Figure 3, pasted above). In this figure we start the plots with a longitude of 0° and proceed from L to R as one would go clockwise around the polar stereographic projections.

Figure 3: If we are comparing how AEVs have changed over time, we probably might also want to adjust the ecological layers to reflect change over time (i.e. 2090-2000).

Response:

We have included the layer comparisons in the revised paper. Now we compare the AEV Indices over time in Revised Figure 4 and we show the ecological layer differences in Revised Figure 6. In both figures we show different plots in addition to the raw fields.

Revised Figure 4:

Revised Figure 6

Figure 4 and associated results: Why were averages only taken over a subset of the area of each domain? How were boundaries decided? What is the effect of the choice of boundaries on the results presented?

Response:

We focused on the most valuable areas and areas that generally coincide with existing or proposed MPA boundaries as examples of changes over time. We now show the Ross Sea in Revised Figure 7 and other regions in Revised Figure S5 as examples of the change in time.

Revised Figure 7

Revised Figure S5

Lines 260-262: Could you provide a percentage?

Response:

We include Revised Table 2 that compares how the percent of area in the different value bins has changed from 2000 to 2090.

Revised Table 2

Table 2 Earth System Model regional Antarctic Ecosystem Value (AEV) Index information related to polynyas (poly.) and marine protected areas (MPAs)

Region ¹	WS	EA	RS	AS	AP
Decade ²	2000/2090	2000/2090	2000/2090	2000/2090	2000/2090
AEV Index					
avg. inside poly.	0.55/0.52	0.57/0.65	0.82/0.77	0.66/0.44	0.54/0.80
avg. outside poly.	0.38/0.41	0.46/0.47	0.36/0.43	0.28/0.38	0.26/0.26
std. inside poly.	0.26/0.06	0.19/0.22	0.43/0.34	0.44/0.34	0.39/0.68
std. outside poly.	0.06/0.13	0.09/0.11	0.14/0.15	0.04/0.06	0.10/0.08
% difference ³	+35/+24	+21/+32	+78/+57	+81/+55	+70/+102
% Total Inside Poly.					
Regional area	1.9/1.9	6.9/11.1	9.1/8.4	1.4/1.9	1.0/0.6
Exceptional ⁴ area	25.1/23.0	53.0/93.9	79.9/45.1	35.3/40.7	9.7/13.0
Very High ⁵ area	5.0/7.7	17.7/27.6	66.5/61.0	0/2.8	3.9/0
High ⁶ area	2.0/1.8	5.1/14.9	29.7/30.7	0/0	1.4/0
% Total Inside MPAs					
Exceptional ⁴ area	100/100	54.6/51.7	86.8/85.0	0/0	91.6/86.7
Very High ⁵ area	100/86.6	48.9/64.7	87.2/85.1	0/0	82.7/80.3
High ⁶ area	67.6/44.8	42.0/66.2	66.3/72.6	0/0	46.6/83.0

¹Average (avg.) and standard deviation (std.) are area weighted over the relevant regions. Region labels correspond to labeled regions on Figure 1a and b.

²Decades are calculated by averaging data from all 50 ensembles in the five years surrounding a given decade (e.g. 2000 averages include data from years 1998-2002). Thus, each statistical calculation includes 250 total samples.

³Positive values indicate a larger AEV Index for points inside polynyas. Percent Difference = $100 * \frac{IN_{poly.} - OUT_{poly.}}{0.5 * (IN_{poly.} + OUT_{poly.})}$

⁴Exceptional indicates an AEV Index in the top 5th percentile. See labels on Figure 4.

⁵Very High indicates an AEV Index in the top 6-10th percentile. See labels on Figure 4.

⁶High indicates an AEV Index in the top 11-25th percentile. See labels on Figure 4.

Lines 251-253: Perhaps qualify this a bit more, after all some areas like the Peninsula decrease significantly.

Response:

We have clarified in the text that we mean that areas that are still within the sea ice zone, which the Peninsula is not, so we can't calculate an AEV Index there.

Lines 219-221: "Future projections show that the locations with the highest AEV Index, and in areas that are still in the seasonal sea ice zone, generally remain consistent in the same coastal locations from 2000 to 2090 (Figure S4 and S3)."

Lines 268-270: Presumably things like iron play a role – do you have any insight how these are related to the polynya areas? There is some literature on individual polynyas (like the Amundsen sea one - <https://doi.org/10.1016/j.gca.2023.10.029>) you could draw on.

Response:

As we now clarify within the methods section, in the MARBL biogeochemistry model, the NPP directly depends on nutrient availability for Nitrogen (N), Phosphorus (P), Silica (Si), and Iron (Fe). Delving into the limitation terms of the NPP is beyond the scope of this study, but because our model does not have nutrient input from ice shelf runoff, increased nutrients in polynyas areas would most likely be due to increased mixing as turbulent fluxes from ocean-atmosphere-ice interactions enhance mixing. In the main text we have added that the increase in productivity in the polynyas could also be due to more nutrient availability.

Lines 312-313: “There may also be increased nutrient availability in polynya regions, likely due to increased ocean mixing in these areas, which could lead to increases in productivity as well.”

Lines 357-360: doesn't make sense that demersal fish biomass increases over time but fish habitat lost for present day populations??????

Response:

You are correct that this was confusing. As we explain now in the methods, we have masked the demersal fish biomass potential by habitat temperature thresholds for Antarctic toothfish and the demersal fish biomass potential shows a decrease in time as bottom temperatures increase.

Lines 361-367: “Demersal fish biomass potential also remains elevated on the continental shelves, in the vicinity of polynyas and would be projected to increase given the likely increases in POC flux to the benthos (Figure 6g-i). However, when we take into account the habitat thresholds for present day Antarctic toothfish, we see that as projected bottom temperatures increase from the 2000s to 2090s the projected Antarctic toothfish biomass potential decreases as warming bottom waters start to limit habitat suitability for this species (Figure 7 e and f and S5).”

*Lines 917-931: “FEISTY is a global model, and the demersal fish functional type in FEISTY represents all types of demersal fish and not a particular species. In the present-day, Antarctic demersal fish include species like Antarctic toothfish (*Dissostichus mawsoni*) and Antarctic icefish (*Chaenocephalus aceratus*). We focus on demersal fish in the AEV index because Antarctic toothfish are important for fisheries, but FEISTY does not have polar specific modifications to represent this Antarctic species directly. Temperatures above 1 °C become increasingly uninhabitable for toothfish [44] and icefish cannot live in temperatures above 2 °C [45]. For the DFP used in the AEV Index, we have summed over each size class of the global FEISTY data. Additionally, we only use values where the bottom temperature is ≤ 1 °C to account for Antarctic toothfish habitat suitability. When we compare the locations of highest DFP from FEISTY with the best estimate of Antarctic toothfish habitat from AquaMaps [66], we see*

that the spatial patterns of maxima in DFP occur on the continental shelves where Antarctic toothfish are most likely to be found (Figure S10).

Lines 401-409: Might want to consider adding a line to say you targeted species across trophic levels that are core ecosystem components in the Southern Ocean and might be good indicators. **Response:**

We have included the following text in the first paragraph of the Discussion section.

Lines 434-435: “The ecological input layers we use include some representing species that are core ecosystem components in the Southern Ocean.”

Lines 409 onwards: What are the differences of your study to the AES study – some of the Ross Sea Area only becomes highly valuable in the future in the AES study, while yours finds that it stays the same / increases? Why are those? This is briefly touched upon in the methods, please elaborate on this in the Discussion. Also the hemispheric results differ more from the AES study. This could highlight bias in the way tracking data was selected for the AES study, or limitations in hemispheric applicability of your index. Please discuss.

Response:

There are some differences and some similarities in the results between the two studies for current and future scenarios. It is important to note that the studies use completely different data sets (as we note in the methods section). Hindell et al. (AES) was limited to using only tracking data (and thus had strong bias towards areas that had been well sampled), whereas we used both data and models to get at circumpolar information more completely and we worked across trophic levels (rather than just birds and mammals). That said, the results align well in some coastal areas, including the southwest Ross Sea, parts of the East Antarctic and the northern Antarctic peninsula. The results in our AEV and Hindell’s AES for future change also align well (see, e.g., losses in the East Antarctic and Weddell Sea regions). For the areas that don’t align as well, it is likely due to differences in the data. For example, areas not picked up by Hindell et al. - e.g., other parts of the East Antarctic and Weddell Sea, are driven by all ecosystem components, including NPP (net primary productivity) and KGP (krill growth potential), rather than penguins. Similarly, some areas emphasized by Hindell et al. (e.g., the Balleny Islands) did not emerge as AEVs in our study because none of our variables emphasized this region. This is likely because we did not have the array of birds and mammals that utilize the Balleny Islands in our study. To clarify why we provide this comparison to AES in the methods section, we now have included the following text.

Lines 1027-1035: “To demonstrate that the historical reconstruction AEV Index - both the hemispheric and regionally scaled versions - provides sensible information about valuable biological areas, we qualitatively compare it to the Areas of Ecological Significance (AES) identified by 12 using in-situ tracking data from 17 marine mammal and bird species to identify regions preferred by these predators. Both the AES and AEV Index metrics identify ecologically

valuable regions though the methodology is entirely different and the metrics do not share any overlapping data. This comparison is done primarily to verify that a model based metric to identify valuable biological areas is providing sensible results to a similar metric that uses observational data. We find...

Lines 416-418: Perhaps rephrase to make this recommendation even stronger, something along the lines of: “This MPA is set to expire in 2052. This work provides evidence the MPA would contain valuable regions through to 2090 and should be considered for extension.”

Response:

We feel our wording as it currently is makes this point clearly and we have not changed this sentence. Additionally, we are unable to imply endorsement of policy positions.

Lines 421 onwards: Might be worth citing <https://doi.org/10.1016/j.marpol.2024.106232> that addresses value of Domain 9

Response:

Thank you for this great suggestion. We have added this citation and text about there being priority areas in the region.

Lines 553-555: “In management regions with proposed MPAs, CCAMLR’s recent lack of progress in adopting these proposed MPAs [32] leaves the species within these areas vulnerable to the combination of both climate change and fishing pressure.”

Lines 428 onwards: A bit confusing, suddenly coastal and “large-scale” polynyas are described rather than just polynyas.

Response:

We intended this to mean that the largest polynyas are visible from space, but we understand it is confusing. We have changed the text as follows.

Lines 451-454: “Within each management region, coastal polynyas have an AEV Index 1.4-2 times larger for areas outside polynyas (Table 1), and our analysis indicates that coastal polynyas are particularly valuable for Antarctic ecosystems across trophic levels. Given the importance of coastal polynyas, many could be prioritized for protection.”

Line 427: Please consider citing some of the sources that proposed “ecological oases” as a concept (I believe it originates from <https://doi.org/10.1371/journal.pone.0023047?>)

Response:

Thanks for the suggestion, but we are not clear how this applies in our study (and we have limited space for elaborating).

Line 433: Was there any patterns or explanations for which polynyas did not have high AEVs?

Response:

No, we did not see a pattern for why the less valuable polynyas were so. It's beyond the scope of this paper to examine this in detail for each polynya as we are trying to evaluate at a higher level and suggest in the discussion that further studies at each polynya may be warranted.

Line 437-8: What is the hypothesized link between increased carbon flux and demersal fish???. Many demersal species in this region are carnivores.

Response:

We believe that the increase in particulate organic carbon (POC) flux to the ocean floor would lead to more prey for Demersal fish. We clarify this connection in the text.

Lines 460 and 527-529 (sequential lines are broken by figures): "More mesozooplankton biomass and increased POC flux to the benthos would provide more prey resources for demersal fish. Polynyas are thus potentially advantageous for predators as they provide access to pelagic and benthic resources and also allow an easy ability to return to the surface for breathing."

Lines 444-448: Gives rise to the question of weighting. Do we care more about some species than others? Are there any areas projected to be suitable for Emperors- should they be targeted more for protection? I'm not necessarily saying that the layers should be weighted differently, this depends on the intended use in terms of conservation outcomes.

Response:

This is a good question and something we discussed in detail while formulating the AEV Index. We believe there is no "right" answer to whether some species "should" be targeted more or less for protection as that is more of a values/political question rather than a scientific question which could be answered with analysis. For example, if you're concerned about the krill fishing industry, maybe you want krill weighted heavily. If you're more concerned about Emperor Penguins then perhaps they should be targeted more. It depends on one's aims/objectives and it isn't something we believe would be appropriate to speak to in this paper, which has a scientific focus, but rather to leave to policy-makers and others.

Discussion starting line 475: While this is an honest discussion around some of the potential issues with demersal fish model, it highlights a significant question around whether this model is actually in any way realistic. Icefish and toothfish are common species and would generally account for a large portion of the biomass. Toothfish are well known to exhibit an ontogenetic shift in habitat with smaller fish in shallow coastal waters and larger/ higher biomass fish in deeper water from around 800m depth. This seems in contrast to the maps presented. Again, it underscores the importance of presenting validation information for the demersal fish model to enable the reader to believe it is robust.

Response:

This is a good point. As stated in our response to one of the reviewer's general comments, we are unaware of any comprehensive species distribution models or biomass estimates for Antarctic toothfish and icefish. We acknowledge that toothfish life history complicates the assessment of

any simulated future change presented in our manuscript. Yet, we think that accounting for the specific life history of Antarctic toothfish is beyond the scope of this work. That is why, in our work, we focus on a larger scale and total potential DFB. We have significantly expanded the methods section where we describe the FEISTY fish model, including a spatial comparison of the FEISTY biomass distribution with the AquaMaps estimate of Antarctic Toothfish Habitat, which demonstrates the suitability of using FEISTY output for our purpose (see Revised Figure S10, and Online Methods Section 4.3 in the revised manuscript (we have not pasted the whole section in this response for brevity).

Line 504: Consider adding a sentence or two about where current anthropogenic activities occur (fishing, but maybe also tourism?) Is there overlap with any of the high AEV areas? Any overlap with high AEV areas that will have decreased by 2090 (i.e. where fishing might contribute to an even faster decline)?

Response:

While we agree that human activities are a major consideration for spatial management in the Southern Ocean, this would be beyond the scope of our study to consider overlap between AEV areas and human activities. Nor can we accurately predict if fishing would contribute to a faster decline given the current uncertainty around impacts of fishing on the ecosystem and the lack of knowledge about cumulative impacts. Ultimately, as noted above, we feel the discussion of human activities impacts and responses are discussions that policy-makers will need to have and indeed, we hope that our work is considered alongside other information as policy-makers consider management of fishing, tourism and other activities. We are actively working to ensure this work is part of the information considered in policy level discussions.

Lines 521 onwards: You are not considering reductions in the ice shelves – your models only extend to the shelf break. Is there a way you could extend your analyses to the areas freeing up in the future?

Response:

This is a great point, but unfortunately CESM, and no other current CMIP models of which we are aware, have evolving ice shelves to look at how those may change and therefore impact habitat in the future. Thus, extending the analysis as you suggest is beyond the scope of our models at this time.

Lines 663-4: States that the AEV was validated against Areas of ecological significance- which comprise areas of sea bird and mammal use. This is not a validation. At best it is a qualitative comparison. The logic that they are similar therefore the AEV is working well if flawed. Firstly, if they are so similar then why do we need the new AEV index?

Secondly, a better comparison would be to quantitatively compare your layers for Emperor penguins and Adelie penguins to the corresponding AES layers which are generated from actual data using tracked animals.

Response:

We have re-worded the methods section to clarify the purpose in comparing to the AES.

Primarily we want to verify that the model-based metric is providing sensible results as an index that uses observations. The main benefit, as we state in the introduction, is that a model based metric provides the ability to provide information at all grid points and into the future. A full quantitative comparison of the AEV Index and AES (and other types of indices and metrics) is beyond the scope of this paper and is currently being completed by co-author Brooks.

Lines 1027-1035: “To demonstrate that the historical reconstruction AEV Index - both the hemispheric and regionally scaled versions - provides sensible information about valuable biological areas, we qualitatively compare it to the Areas of Ecological Significance (AES) identified by 12 using in-situ tracking data from 17 marine mammal and bird species to identify regions preferred by these predators. Both the AES and AEV Index metrics identify ecologically valuable regions though the methodology is entirely different and the metrics do not share any overlapping data. This comparison is done primarily to verify that a model based metric to identify valuable biological areas is providing sensible results to a similar metric that uses observational data. We find...”

Lines 668 onwards: This is true for the regionally scaled AEV, but more significant differences occur when looking at the hemispheric solution.

Response:

This is a good point. In the methods section where we compare the hemispheric and regionally weighted AEV Index with the AES regions, we have included the following text.

Lines 1039-1041: “However, the regionally weighted AEV Index has more overlap with the AES in sectors beyond the Ross Sea and Antarctic Peninsula, thus bolstering our conclusion that the regional weighting is a better way to identify regionally important high value areas.”

Appendix B- 1214-1217: The penguin population models seem to already include sea ice variability in population size estimates. Is there circularity in using sea-ice again for the projection of important areas?

Response:

We do not believe that there is circularity here. The AEV Index inputs are only biological and do NOT directly include sea ice concentration, polynya area, or any other physical variable as a direct layer. The biological inputs do include environmental information, so environmental impacts on the AEV Index are only considered by their impact on the biological inputs in terms of their specific relationship with the five input biological layers. The analysis we do on polynyas uses a separate dataset and the polynyas are not included in the AEV Index calculation.

Appendix B- 1308-1312: Indicates some validation of the emperor penguin model was conducted, but nothing is reported. Is this a ‘good’ or fit for purpose model?

Response:

The validation of the emperor penguin model is documented in Jenouvrier et al. (2025 <https://doi.org/10.1016/j.biocon.2025.111037>) and detailed in that paper's Supporting Information. Specifically, Figure B.3 presents a long hindcast (1950–2018) and projection (2019–2100) of population percent change relative to 2009–2018. These results demonstrate that the models reproduce historical dynamics before being used for future projections. In addition, Figure E.1 reports an out-of-sample validation at Pointe Géologie successfully captures the recent increase in breeding pairs, showing skill under novel conditions despite calibration on a prior, non-increasing phase. Validation methods are summarized in Appendix E (and Section 2.7 in the main text). For reference, the Supporting Information is available here: <https://ars.els-cdn.com/content/image/1-s2.0-S0006320725000746-mmc1.pdf>

Figure B.3 from Jenouvrier et al. 2025

Figure E.1 from Jenouvrier et al. 2025

PUBLISHED DATA FIGURE REDACTED

Reviewer #1 (Remarks on code availability):

Coding was completed in Python and I am not proficient in this language.

Response:

Thanks for confirming our code was accessible and in a widely used language (Python). We have updated all the analysis scripts as part of this revision and the code remains available at: <https://github.com/duvivier/HotSpotsinIce>

Reviewer #2 (Remarks to the Author):

Response:

Thanks for the thoughtful review and comments. We appreciate your suggestions that have improved this manuscript substantially.

Reviewer #3 (Remarks to the Author):

This manuscript presents an interesting and timely approach to identifying ecologically valuable areas in the Southern Ocean by creating the Antarctic Ecosystem Value (AEV) Index. While the study provides valuable insights into current valuable areas and potential changes in them under future climate scenarios, I have some methodological concerns that should be addressed before publication. These include mainly a clarification of the regional standardisation on the text to avoid potential misinterpretation, and further explanation on the choices made for the weighting of ecosystem components. The weighting as is overrepresents penguins, and a justification or adjustment of the model is necessary. I think the authors do a good job on the discussion addressing some of their model limitations, but don't mention the lack of other components of the food webs of the Southern Ocean, or why penguins are so important. Addressing these issues would strengthen the ecological foundation of the study and improve its utility for conservation planning in the Antarctic region.

General comments

I think this study will be of relevance to Southern Ocean studies, and it's novel enough compared to previous attempts at creating similar indices. The code provided with the article is well documented. I think the work supports most of the conclusions, except for what I raise below.

Response:

Thanks for taking the time to review the manuscript. We appreciate your finding that it is timely, novel, and provides valuable insights. Below we describe how we have addressed the three methodological concerns you raise.

- 1) You raised a concern about regional standardization of the AEV Index. Based on other reviewer comments, we agree that our previous method of comparing between regions was not appropriate. We have changed how we are comparing the valuable areas between regions by using percentiles from the AEV Index distributions in each region to determine Exceptional, Very High, and High value points, and then comparing these points between regions. We describe this in the main text and methods sections, Revised Figures 1, 3, 5, and include Revised Figures S12 and S13.*

Lines 154-164: "It is important to note that the AEV Index values are not directly comparable between the hemispheric and regional indices, nor between separate regions, because each grid point has been scaled differently. To directly compare the most valuable locations, we identify use three bins - Exceptional (top 5th percentile), Very High (top 10th percentile), and High (top 25th percentile) value - to describe points based on the distribution of AEV Index values within a particular region. All other points are classified as Other (lowest 75th percentile). Grid points classified within the Exceptional, Very High, or High bins are the most valuable within a particular region (and we often refer to the combination of all three bins as highly valuable areas), and since they are determined from consistent percentiles from regional distributions of the AEV Index they are directly comparable between regions."

Lines 1021-1026: “To highlight the particularly valuable locations within each region, we have binned the AEV Index into four categories: Exceptional (AEV Index in the top 5th percentile for that region), very high (AEV Index in the top 10th percentile for that region), high (AEV Index in the top 25th percentile for that region), and other (all other points). Note that the percentile thresholds for these bins are determined separately for each of the regions and decades (see Figures S12 and S13).”

Revised Figure 1

Revised Figure 3

Revised Figure 4

Revised Figure S12 (and Revised Figure S13, not copied here)

2) You were concerned that the penguin inputs to the AEV index are overweighted. This is a good concern and something we discussed in detail while formulating the AEV Index. To be sure this choice did not affect our results we did some testing that we show in Figure R1 (below) for both the hemispheric and regionally weighted AEV Index calculations. The “AEV default” maps (left column) show the results when both penguin species have a weight of 1, equal to the other input layers. We also tested assigning both penguin species a weight of 0.5 (right column) so that the combined penguin input layer would have the same weight as the other individual layers (e.g. KGP). We found that the AEV Index did not substantially change in terms of which locations were identified as most valuable within a region. Thus, weighting each species by half does not substantively change the results of the paper in terms of identifying valuable biological locations within the Antarctic. We believe there is no “right” answer to whether some species “should” be targeted more or less for protection as that is more of a values/political question, and as such we don’t feel that we should weight any species more or less than others in this Index. We have added text (below) to the methods explaining how the penguin weighting in the AEV Index is done and why.

Figure R1

*Lines 934-941: “We use two input layers representing different species of penguins and their projected population evolution to represent secondary/tertiary consumers. We include the Emperor Penguin (*Aptenodytes forsteri*) and Adelie Penguin (*Pygoscelis adeliae*), both upper trophic level predators, because these two species experience different future population projection trajectories. In the AEV Index we assign each penguin input layer equal weight as the other layers, though testing shows that using a single penguin layer where each species has a weight of 0.5 results in a similar AEV Index as the results shown here.”*

- 3) *You had concerns about the biological model limitation and lack of representation about other species in the AEV Index. We agree that this lack of other species is a concern in our method. However, we used all the species of which the authors were aware that had models for future*

projections of their population, biomass, or productivity. We would love to include more species in the future, including seals or whales, but we did not have these models available at this time. We think future further expansion of the AEV Index would be valuable as such data becomes available. We have included the following text in the Discussion section to address these concerns.

Lines 570-590: “When possible, the AEV Index uses species-specific future projections, but more detailed species-specific modeling is needed, especially for ice-obligate species. We include input layers for primary producers (NPP), primary/secondary consumers (KGP), secondary consumers (DFP), and secondary/tertiary consumers (EPP/APP) (see Section 4 for details). However, other key ecosystem species like whales, seals, other penguins and fish, and flighted birds are not included. While it would have been ideal to include a larger range of species, we included only those species for which we had readily available future projection information around the entire Antarctic continent. Unfortunately, the authors are unaware of any other species for which that level of data is available and which we could include in these calculations. The results presented here show that the AEV Index framework is valuable as a tool to demonstrate the utility of such studies across trophic levels, and future work to include more species would help fill gaps about the Antarctic ecosystem that this study is unable to address.”

Specific comments

Lines 164-166: “The AEV Index is useful for three reasons. First, it seeks to comprise a holistic ecosystem value by focusing on Antarctic-specific species and integrates across trophic levels (see 28).”

See my comments below, but calling it a “holistic” ecosystem value may be too generous when you’re missing some key taxonomic groups (and by consequence overestimating the importance of the ones included) that are very important in regional and global carbon and nutrient cycles.

Response:

This is a good point and we agree “holistic” is incorrect since we do not include all important species. We have changed the text to use the wording “broader ecosystem”.

Lines 127-128: “First, it seeks to comprise the broader ecosystem value by focusing on Antarctic-specific species...”

Lines 156-159: "We use data from an Earth system model (ESM) that includes marine biogeochemistry [19, 20] for both a historical reconstruction [21] and coupled future projection with upper-middle CO2 emissions [22]."

Reading this, the methods, and looking at Table 1, you have what it looks like two historical periods; one that you can historical in table 1, and another you name as "2000s". I understand one is based on historical reconstructions and the other is model-based and used for future projections, but it's not very clear on the methods which one is which and why you need both. Please make it clearer on the text thinking of the average reader that may not be used to modelling and future projections. This also applies to Table A1: understand the Earth System

Model includes the future projections, but I think both the table and the text in general would benefit from clarifying or identifying that the ESM includes future projections.

Response:

We realize how confusing this was. The first section of the Results focuses on the historical reconstruction and the second section focuses on the free-running model. We also added a paragraph explaining how the free-running model derived AEV Index differs from the historical reconstruction AEV Index, and why that means there are two “historical” periods.

Lines 720-739: “The historical reconstruction AEV Index uses data from an ice-ocean model forced by atmospheric observations to reconstruct observed ocean and sea ice conditions [23]. On the other hand, an individual fully-coupled, free-running model simulation should not exactly replicate observations, since observations represent one possible response of the climate system to given external forcing in the presence of internal climate variability. Instead, to ensure the AEV Index derived from the fully-coupled, free-running model data [24] is robust, we compare the historical reconstruction (average for 2000-2020 conditions) with the fully-coupled simulation data for the 2000s (average over 50 ensemble members from 1998-2002) to ensure they produce similar results in a similar climate state. Indeed, we find that the valuable areas identified by the regional AEV Index historical reconstruction (Figure 1d) align well with valuable areas identified by the AEV Index calculation using a fully-coupled, free-running model for the 2000s (Figure 4a). Both methods identify the same large, valuable areas in all regions, though the free-running model has more valuable grid points further north in the Amundsen Sea and the East Antarctic. A major benefit of the AEV Index derived from the fully-coupled model is that it can be used to identify valuable areas into the future. Because the two AEV Index formulations provide similar information in the present day, it is reasonable to use the free-running, model-based AEV Index in the next section of this manuscript to provide information about potential future changes.”

Lines 268-270: "In all regions total net primary productivity increases over time (Figure 4b), likely because ice loss reduces phytoplankton light limitation and extends the growing season." If NPP is coming from the ESM, does the ESM specifically model the relationship between sea ice loss and phytoplankton productivity? While this mechanism is plausible, it would be helpful to confirm whether the model actually simulates these specific processes (light limitation changes due to ice loss) or if this is an interpretation of the results. If the model doesn't explicitly track light limitation, there could be alternative explanations for increased NPP.

Response:

Yes, the ESM models the relationship between light availability and phytoplankton productivity. While we feel that delving into the limitation terms of the NPP is beyond the scope of this study, we have substantially expanded the methods section 4.3.1 of the paper to describe how NPP is calculated.

Lines 789-794: “This input represents primary producers, the base of the food web. As documented in detail in 20, the Marine Biogeochemistry Library (MARBL) component of

CESM2 directly calculates time evolving phytoplankton production for several functional types. The phytoplankton growth rate (μ_i) is a function of the resource-unlimited growth rate (μ_{ref}) at a reference temperature (30°C), and the temperature ($LimT$), nutrient ($LimV$), and light availability ($LimL$) functions...

Fig. 2 "a) Model based 2000s locations of high (pink) and exceptional (red) Antarctic Ecosystem Value (AEV) Index indicating ecologically valuable regions"

The regional normalization approach used for the AEV Index creates a potential misinterpretation risk that I think would benefit from being addressed, at least in this figure. By normalizing values within each planning region rather than across the entire Southern Ocean, areas labelled as "exceptional value" in different regions are not directly comparable. For example, an "exceptional value" area in East Antarctica may have significantly lower absolute ecological importance than an "exceptional value" area in the Ross Sea or Antarctic Peninsula. So I would recommend clearly stating in the figure legend and maybe more clearly in the main text that the AEV Index values represent relative importance within each planning region only, not across the entire Southern Ocean. I do understand the reasoning behind this standardization, but making it clear for the interpretation of your results doesn't hurt.

Response:

This is a very good point and one that we had not fully considered in the previous version of the paper. As we described above, we now use percentiles within the distribution of each region to determine the "Exceptional", "Very High", and "High" value regions so that direct comparison between regions is possible. Thank you very much for bringing this up - we feel that this change has substantially improved the robustness of the results even though the results themselves did not change substantially.

Lines 588-590: This line mentions how CESM2 does not represent specific Antarctic phytoplankton species. It would be nice to know which phytoplankton communities are not included, and how much they are expected to contribute to NPP. I'm not overly familiar with CESM2 and I think a lot of people will not know the specific details, so possibly just a sentence here clarifying how this could impact estimations of NPP would be nice. For instance, does the model capture ice algae communities? Understanding these limitations would help readers better interpret the NPP projections.

Response:

We have expanded section 4.3.1 in the methods substantially to describe the calculation of NPP. While the CESM model does not include phaeocystis or in-ice phytoplankton, efforts are underway to include these and to use additional, polar-specific types of phytoplankton and zooplankton. However, to maintain the broader applicability of this model based metric, we use NPP alone within the AEV Index calculation.

Lines 823-831: "While we are aware of the complex dynamics of plankton functional types, we chose to use NPP alone to maintain the metrics broader applicability across different

models without requiring detailed assumptions about species-specific responses. NPP is a direct output from CESM2. All configurations of CESM2 calculate phytoplankton growth rates separately for three phytoplankton functional types: diatoms, diazotrophs, and small phytoplankton. The CESM2-FOSI configuration also includes coccolithophores as a functional type. None of these functional types are Antarctic specific, but instead broadly applicable to a global model. Sea-ice algae are not included.”

Lines 645-655: You gave equal weighting to all five metrics (NPP, KGP, DFB, EPP, APP), and I want to understand the reasoning as this may overrepresent certain trophic levels while undervaluing others. There's two penguin-specific metrics (EPP and APP) but only one primary productivity metric (NPP), which means that the index assigns 40% of the weighting to penguins while allocating just 20% to the trophic level that supports the entire ecosystem, from my understanding of it. This imbalance may distort the identification of high-value areas by overemphasizing regions important to penguins relative to their ecological significance in the broader Antarctic food web. I would like to at least see a justification in the text on why penguins as top predators would justify being 40% of the index. Additionally, the model excludes other key top predators such as whales and seals, which play critical roles in Southern Ocean ecosystem functioning. Their absence further complicates the potential bias created by the penguin-heavy weighting. If it's not possible to justify this weighting, then I would recommend implementing a more balanced weighting of the different trophic components.

Response:

We have clarified above and in the text why we weight each species the same in the AEV Index and the testing we did to ensure this was okay with two penguin species. We also addressed why other species like whales and seals were not included (See above response to methodology concern #2 and #3).

Reviewer #3 (Remarks on code availability):

I haven't tried running the code as it's Python and I'm not super familiar with Python coding, but there's a README file and the code is in a Jupyter notebook and well commented.

Response:

Thanks for confirming our code was accessible, in a widely used language (Python), and well commented in the code and with a README. We have updated all the analysis scripts as part of this revision and the code remains available at: <https://github.com/duvivier/HotSpotsinIce>

Reviewer #4 (Remarks to the Author):

This manuscript introduces a new index to evaluate the value of Antarctic marine ecosystems called the Antarctic Ecosystem Value (AEV) that merges ecosystem information across food web trophic levels. With this index, high value regions in Antarctica were identified for current

conditions and for future projections to 2090. They found areas of reduced sea ice (polynyas) to be particularly valuable. They deemed areas that continue to be valuable into the future especially important to preserve. These areas were largely within MPAs, but also in proposed MPA areas or areas not in MPAs, which led to the recommendation to develop protection plans for valuable but currently unprotected regions. The manuscript was well-written and uses the best available models and data to inform the index. I agree that the metrics included in the index together form an important and informative collection to evaluate the value of regions in Antarctica.

Response:

Thanks for taking the time to review the manuscript. We are happy you found it informative, and we appreciate your comments that have substantially improved the manuscript and analysis.

Since this is an introduction of a new index, I would like to see presented as part of the results that the index works. I suggest moving the model validation section into the result section with figures and statistical output, and provide a compelling conclusion from this that the index does indeed correspond with quality habitat/high value ecosystems. The same with the AEV ranges, allow the reader to conclude that the ranges and cut-offs from good to excellent are logically placed. Now that readers have the information to decide if the index can be trusted, results from the index, based on and future conditions, can be presented.

Response:

We did not include the comparison to the AES presented by Hindell et al. in the main text due to word limit constraints. Unfortunately, this remains an issue in the revised manuscript. However, we have expanded the methods section discussing the qualitative comparison. In the methods section we clarify the purpose in comparing the AES with the AEV Index. Primarily, we want to verify that the model-based metric is providing sensible results as an index that uses observations. A full quantitative comparison of the AEV Index and AES (and other types of indices and metrics) is beyond the scope of this paper and is currently being completed by co-author Brooks. Below, is the paragraph in the methods where we validate the AEV Index against the AES.

Lines 1027-1052: “To demonstrate that the historical reconstruction AEV Index - both the hemispheric and regionally scaled versions - provides sensible information about valuable biological areas, we qualitatively compare it to the Areas of Ecological Significance (AES) identified by 12 using in-situ tracking data from 17 marine mammal and bird species to identify regions preferred by these predators. Both the AES and AEV Index metrics identify ecologically valuable regions though the methodology is entirely different and the metrics do not share any overlapping data. This comparison is done primarily to verify that a model based

metric to identify valuable biological areas is providing sensible results to a similar metric that uses observational data. We find that both metrics identify similar grid cells as valuable (grid cells in the Exceptional, Very High, and High value bins) as the AES locations, particularly in the coastal areas of the southwest Ross Sea, parts of the East Antarctic, and in the northern Antarctic Peninsula. However, the regionally weighted AEV Index has more overlap with the AES in sectors beyond the Ross Sea and Antarctic Peninsula, thus bolstering our conclusion that the regional weighting is a better way to identify regionally important high value areas. In other areas, the results don't align as well, which may be due to differences in data availability. In the Amundsen Sea there is less animal tracking data and it is also an area the AES does not identify as a valuable area while the AEV Index does. In other areas, like the Balleny Islands northeast of the D'Urville station in East Antarctica, the AEV Index does not show these as valuable possibly because we didn't include species inputs that emphasized this area. Thus, we believe that the AEV Index, which is calculated using data on a coarse (~1°) Earth system model grid, is capable of providing high level information about valuable areas around Antarctica, particularly into the future. It is important to note that additional research and monitoring, including with existing observational datasets such as those used in the AES study, could be conducted to further refine locally valuable regions.”

Minor note on page 5 and on: The hyperlink to S1 doesn't take you to the S1 figure, but to page 5.

Response:

Thanks for finding this error. We believe we have fixed the links to particular figures, tables, and sections in the revised paper.

Reviewer #5 (Remarks to the Author):

This manuscript provides an aggregated index for considering productivity and accessibility for two predator species at a circumpolar scale in the Southern Ocean. The authors calculate this index for historical, present, and future climate conditions, using output from Earth System

Model simulations. The index shows a strong relationship with the location of Antarctic polynyas, which generally persists under future climate conditions.

This work is ambitious and is a useful demonstration of an approach to aggregate information about net primary productivity, krill growth potential, demersal fish biomass potential, emperor and Adelie penguin populations, and access to areas for foraging. The authors should be congratulated on the volume of work underpinning the manuscript and the clear presentation of results in figures and tables.

Response:

Thanks for taking the time to review the manuscript! We appreciate that you found it clear and ambitious. Thank you for your suggestions, addressed in detail below, which we think have helped improve the manuscript.

We have several concerns regarding the use of this index to consider Antarctic ecosystem value more generally, as well as the methods for calculating krill growth potential and demersal fish biomass potential. Large-scale ecosystem processes and connectivity, together with the key food webs roles of zooplankton groups and mesopelagic fish, are all critical components of ecosystem value in the Southern Ocean that underpin decisions about spatial protection, but are not discussed by the authors or considered as part of the index. There is little recognition that groups other than Antarctic krill drive energy transfer, with their relative influence differing across regions – for example, East Antarctic food webs show a higher number of alternative energy pathways between primary production and large predators (e.g., McCormack et al., 2020, 2021, doi: 10.1002/ece3.7017 & doi: 10.3389/fevo.2021.624763). This has broader implications with respect to the potential use of this index to inform marine spatial protection in the Southern Ocean, because (as we are sure the authors will be aware) Antarctic marine protected areas need to be of sufficient scale to protect ecosystem processes – not just small-scale high productivity areas. Equally, marine spatial protection is important for areas that are at risk of degrading in value due to climate change, because enhanced resilience (through reduction in other threats) provides the best chance of ecological processes and values persisting in these areas.

Response:

Thank you for bringing up this concern about gaps in which species and components of the Antarctic ecosystem are included in this work. We agree that this lack of other species is a concern in our method. However, we used all the species of which the authors were aware that had models for future projections of their population, biomass, or productivity. We would love to include more species in the future, including other zooplankton or mesopelagic fish, but we did not have these models available at this time. We think future further expansion of the AEV Index would be valuable as such data becomes available. We have included the following text in the Discussion section to address these concerns and cite the papers you suggest.

Lines 570-592: “When possible, the AEV Index uses species-specific future projections, but more detailed species-specific modeling is needed, especially for ice-obligate species. We

include input layers for primary producers (NPP), primary/secondary consumers (KGP), secondary consumers (DFP), and secondary/tertiary consumers (EPP/APP) (see Section 4 for details). However, other key ecosystem species like whales, seals, other penguins and fish, and flighted birds are not included. Additionally, this study focuses on the species for which future projection data are available from biological models. However, this means that we do not include other mesozooplankton or pelagic fish that are important for energy pathways from the primary producers to predators [e.g. 33, 34]. Additionally, excluding all the species means that we may not be capturing the scale needed for protection to protect all ecosystem processes, not just the species represented here. While it would have been ideal to include a larger range of species, we included only those species for which we had readily available future projection information around the entire Antarctic continent. Unfortunately, the authors are unaware of any other species for which that level of data is available and which we could include in these calculations. The results presented here show that the AEV Index framework is valuable as a tool to demonstrate the utility of such studies across trophic levels, and future work to include more species would help fill gaps about the Antarctic ecosystem that this study is unable to address. Until then, the AEV Index is associated with larger uncertainty in regions or at times when species not included in our assessment play a vital role in the ecosystem. ”

With respect to the methods used by the authors to calculate krill growth potential, we are concerned about the use of the 2006 Atkinson model, which is not designed to address the effects of climate change on physiology, energetics, and fecundity of krill, which are the primary factors governing productivity. The authors do discuss other krill models briefly, but indicate that “they are hindered by substantial uncertainties regarding krill ecology and their interactions with environmental drivers”. It is unclear to us which specific uncertainties prevent the use of more recent and improved krill models in this study. As it stands, the use of a simple empirical relationship between temperature, surface chlorophyll, and krill growth needs to be much more strongly caveated – with implications for interpretation of results explained; the equation for calculating krill growth should also be included for clarity. Consideration, or a caveat, should be given to the variety of krill diets and their dependence on food items other than diatoms, even during the productive summer months. The aggregation of all phytoplankton groups from the NPP model to simulate diatoms is potentially an oversimplification and might lead to an overestimation of krill growth, as krill have been shown to be quite selective feeders (Pauli et al., 2021 <https://doi.org/10.1038/s42003-021-02581-5>).

Response:

We understand Atkinson 2006 is a simple model, however it’s important to note that in this study, KGP is not used as a mechanistic predictor of krill productivity, but as an indicator of habitat suitability based on temperature and chlorophyll, consistent with prior broad scale applications (e.g., Hill et al. 2013; Murphy et al. 2017; Veytia et al 2020; Sylvester et al. 2021). While the concern that aggregating phytoplankton groups may overestimate potential growth, this bias is consistent across our study and does not affect the relative comparisons central to the AEV

Index. We now include a comparison of the Atkinson 2006 derived KGP term in our model to krill habitat estimates from AquaMaps (Kaschner et al. 2019) and it compares reasonably well (Revised Figure S9, below). Since the authors are unaware of any other more complex model that is suitable for addressing the impacts of climate change on krill growth or secondary productivity and we find the KGP term compares reasonably well to observed habitat we continue to use the Atkinson 2006 model with NPP from all phytoplankton in this work.

Please see Online Methods Section 4.3.2 in the revised manuscript where we have substantially expanded the description of the krill growth potential model (we have not pasted the whole section in this response for brevity) with equations and caveats, and references, as you suggest.

Revised Figure S9

We are also concerned about the use of projections for demersal fish biomass based on a global model that does not account for the unique physiology of Antarctic fish. In lines 475-485 the authors clearly state that the projected changes for demersal fish are not consistent with expected impacts of temperature increases on key Antarctic species. Future increases in demersal fish biomass due to southward expansion of subpolar and temperate species (presumably not subtropical as the authors state in line 483?) are highly uncertain. As such, we think that the fish projections used in the index are misleading.

Response:

As mentioned above, we have substantially expanded the methods section to explain and validate the Antarctic sea ice as well as different AEV Index input layers in detail. Please see Online Methods Section 4.3 in the revised manuscript, where we now include equations for the calculation of NPP, KGP, and explain what inputs the FESITY model needs.

In the expanded methods section we explain that FEISTY is a global model and the input variables. FEISTY provides estimates of biomass for three types (demersal, forage, and large pelagic fish) over three size classes, as illustrated in the diagram, below, from Krumhardt et al. 2024. The authors are unaware of any species distribution models specific to Antarctic fish. Thus, while FEISTY is not designed for Antarctic fish species, we believe it is the best current estimate of possible fish biomass for the present day and future. To account for habitat suitability

of Antarctic toothfish (an example of an Antarctic demersal fish species), we have now masked the Demersal Fish Biomass Potential input layer by ocean bottom temperatures and only use values where the ocean temperature is $<1^{\circ}\text{C}$, which is an upper limit on habitat suitability for Antarctic toothfish (Cheung et al. 2008). In Revised Figure S10 (copied below), we qualitatively compare the locations where FEISTY has highest possible biomass against the Aquamaps estimate of Antarctic toothfish habitat (Kaschner et al. 2019) and it compares well. While adequate observations of Antarctic toothfish biomass are missing to quantitatively evaluate the simulated biomass fields in FEISTY, this assessment gives us confidence in the simulated spatial distribution and thus in using these fields as an input layer for the AEV index.

Figure from Krumhardt et al. 2024

PUBLISHED DATA FIGURE REDACTED

Revised Figure S10

With masking of Demersal fish biomass based on Antarctic toothfish habitat limits, we find that the timeseries of changing average biomass over each region makes sense with increasing bottom temperatures over time decreasing habitat for this species. See Revised Figure 7 (below) - panel e and the Revised Figure S5, DFP panels (not copied in for space).

Revised Figure 7 - see panel e

Finally, the usage of the climate model CESM2-LE should be justified in light of its inclusion in an ecosystem index. For example, the sea ice-ocean model ACCESS-OM2 has been specifically reviewed for its application in ecological studies (Fierro-Arcos et al., 2023 <https://doi.org/10.1016/j.pocan.2023.103049>). Were other ESMs considered?

Response:

The ice-ocean ACCESS-OM2 model you reference is forced using an atmospheric reanalysis, so it does not provide information for the future and it is not fully-coupled so it cannot capture the full range of change in the presence of climate variability. We have clarified in the text why we use CESM2 for this study and that we use both a forced ice-ocean configuration (the same type of model as ACCESS-OM2) as well as a fully-coupled, free-running configuration. In particular, CESM2-LE is an ideal tool for this study because the large number of ensemble members provide a robust signal of forced change due to emissions in the presence of internal climate variability. Importantly, the referenced ACCESS-OM2 simulation does not include any representation of ocean biogeochemistry and lower trophic level ecosystem dynamics. While Fierro-Arcos et al. state that updated ACCESS-OM2 simulations do include these dynamics, they have not been, to the best of our knowledge, thoroughly evaluated for Southern Ocean applications yet. As such, ACCESS-OM2 is currently not suitable for use in our work, which requires, e.g., estimates of chlorophyll concentrations or NPP as input layers. CESM2 not only includes an established, well-evaluated biogeochemistry component (MARBL) but has been successfully coupled to FEISTY in the past (Krumhardt et al., 2024). We clarify in the Results section why we use both the forced ice-ocean CESM2 output as well as CESM2-LE for the future. We include in the Discussion section on model uncertainty that other models such as

ACCESS-OM2 could be used for similar work. And finally, we have expanded the validation of CESM2 in the methods to show that it is a reasonable tool to use for this work.

Lines 720-729: “The historical reconstruction AEV Index uses data from an ice-ocean model forced by atmospheric observations to reconstruct observed ocean and sea ice conditions [23]. On the other hand, an individual fully-coupled, free-running model simulation should not exactly replicate observations, since observations represent one possible response of the climate system to given external forcing in the presence of internal climate variability. Instead, to ensure the AEV Index derived from the fully-coupled, free- running model data [24] is robust, we compare the historical reconstruction (average for 2000-2020 conditions) with the fully-coupled simulation data for the 2000s (average over 50 ensemble members from 1998-2002) to ensure they produce similar results in a similar climate state.”

Lines 655-660: “Moreover, the Earth system model we use has a coarse ($\sim 1^\circ$), but common, grid size for this type of model. Other forced and fully-coupled models or high resolution ice-ocean modeling (e.g. ACCESS-OM2 [54] or FESOM-REcoM [18]) could elucidate the uncertainty in small-scale ice features, marine heat waves, bathymetric impacts on optimal marine habitat, or intensity and locations of warm circumpolar deep water intrusions onto the shelf that could impact habitat suitability [18, 55–57].”

Revised Figure S6

Specific comments:

1. Line 146 – please see Cavanagh et al. (2021, doi: 10.3389/fmars.2020.615214), who consider more than two trophic levels and have a Southern Ocean focus.

Response:

Thank you for making us aware of this paper. We have modified the wording in the text and cited this paper.

Lines 103-105: “..though these often focus on one species (e.g., Emperor penguins, Aptenodytes forsteri) [15, 16], fewer trophic levels [e.g. 17, 18], or are global in scope...”

2. Lines 216-217 – it is unclear to us how Figure S1 compares the historical reconstruction with the model-based index. Could the authors please explain how this comparison is being made?

Response:

This figure has been removed. The text has been updated to clarify how to compare the historical reconstruction with the fully-coupled model for the 2000s.

Lines 720-729: “The historical reconstruction AEV Index uses data from an ice-ocean model forced by atmospheric observations to reconstruct observed ocean and sea ice conditions [23]. On the other hand, an individual fully-coupled, free-running model simulation should not exactly replicate observations, since observations represent one possible response of the climate system to given external forcing in the presence of internal climate variability. Instead, to ensure the AEV Index derived from the fully-coupled, free-running model data [24] is robust, we compare the historical reconstruction (average for 2000-2020 conditions) with the fully-coupled simulation data for the 2000s (average over 50 ensemble members from 1998-2002) to ensure they produce similar results in a similar climate state.”

3. Methods – the inclusion of equations within the bullet points in the AEV index methods section, on top of the written explanation, would make it clearer for the readers to follow. It would also be useful to see some form of sensitivity analysis regarding which of the inputs for the index is most important in driving the overall value of the index over time.

Response:

We have substantially expanded the methods section to include more information about each input layer and validation. Additionally, we thought your suggestion to analyze quantitatively which input is the largest driver of the AEV Index was a great one. We have now included this in the model and it had interesting results about shifts in the ecosystem. We have added text (see below) in the results section, Revised Figure 5, and Revised Table 3 to show which species dominates the AEV Index in the 2000s and 2090s in the free running model. We also include Revised Figures S2 and S4 to show the raw contributions by species.

Lines 269-275 & 306-308 (separated by a Figure): “Investigation of individual inputs to the AEV Index reveals that the importance of penguins on the index decreases while lower trophic levels becomes more pronounced (Figure 5 and SF4; Table 3). In the 2000s the Emperor penguin layer is particularly prominent in the Weddell Sea and Ross Sea and polynyas, while the Adelie penguin layer is prominent in the Antarctic Peninsula. However, by the 2090s the penguin dominant regions have shrunk dramatically. The shift in which input layer is the primary contribution to the AEV Index reflects large changes occurring throughout the food web and highlights ecosystem change from climate change pressures. Below, we describe changes in each AEV Index input layer both hemispherically and in regional averages.”

Revised Figure 5

Revised Figure S2

Revised Figure S4

Revised Table 3

Table 3 Regional average percent (%) contribution of each input layer to the Antarctic Ecosystem Value (AEV) Index

Region ¹	WS	EA	RS	AS	AP
Decade ²	2000 → 2090	2000 → 2090	2000 → 2090	2000 → 2090	2000 → 2090
Inside polynyas					
NPP	28.5→36.8	33.7→36.0	30.6→30.1	39.8→37.3	25.8→33.4
KGP	20.1→37.2	28.8→48.1	27.9→27.6	35.9→49.1	22.2→30.2
DFP	3.1→0.0	6.2→5.1	8.1→8.3	6.7→0.0	23.3→23.3
EPP	48.4→25.9	23.4→3.3	31.2→30.9	15.1→7.5	25.4→10.8
APP	0.0→0.0	7.9→7.6	2.3→3.1	2.5→6.2	3.4→2.4
Outside polynyas					
NPP	26.7→22.8	28.2→23.6	22.2→16.8	26.9→15.6	22.4→19.7
KGP	27.1→24.2	31.7→23.6	26.2→30.6	29.6→21.3	24.1→19.7
DFP	0.2→0.1	0.3→0.2	0.4→0.4	0.1→0.1	0.6→0.4
EPP	43.0→9.1	16.7→2.1	26.1→16.4	15.8→4.5	31.5→11.0
APP	0.0→0.0	1.7→1.4	1.2→1.8	0.3→0.6	3.3→1.1
Valuable³ areas					
NPP	12.5→34.0	33.1→33.2	26.1→23.6	39.9→34.9	26.7→35.2
KGP	9.3→42.3	41.1→55.5	27.4→30.1	41.8→50.0	31.8→32.1
DF	0.8→0.6	1.6→2.6	5.4→5.0	0.5→0.3	2.8→3.8
EPP	77.4→23.1	20.6→4.9	36.1→32.5	17.4→11.9	26.4→24.6
APP	0.0→0.1	3.2→3.9	5.0→8.7	0.4→2.8	12.2→4.3

¹Average (avg.) and standard deviation (std.) are area weighted over the relevant regions. Region labels correspond to labeled regions on Figure 1a and b. Note that at each grid point we have verified the sum of the contribution from each input layer is 100%, but the average percent contribution over a region does not necessarily add up to 100%.

²Decades are calculated by averaging data from all 50 ensembles in the five years surrounding a given decade (e.g. 2000 averages include data from years 1998-2002). Thus, each statistical calculation includes 250 total samples.

³Valuable areas correspond to areas classified as Exceptional, Very High, and High, based on having an AEV Index in the top 25th percentile. See labels on Figure 4.

4. Figures: Fig. 1 side panels are very hard to read – making them larger would be very helpful.

Response:

This figure has changed substantially and the input layer panels are now shown larger in Revised Figure 2.

Revised Figure 2

Fig. S1 refers to Fig. 1, but that is not immediately clear; including the historical reconstruction in S1 would make the comparison much easier.

Response:

The figures have changed and this concern is no longer relevant to the updated figures.

Fig. 2: Interestingly, a large swath of East Antarctic waters is considered to be of Medium value. However, no explanation is given in the text; what could be driving this?

Response:

We have modified how we quantify the value of grid points based on comments by R3. Instead of using an AEV Index threshold, we now use percentiles within the AEV Index distribution for each region. This makes the “exceptional”, “very high”, and “high” value points between regions directly comparable. We note this change in the text and as a result of the change in categorizing the value of the grid points we do not believe the swath of waters you mention is still notable.

Lines 154-164: “It is important to note that the AEV Index values are not directly comparable between the hemispheric and regional indices, nor between separate regions,

because each grid point has been scaled differently. To directly compare the most valuable locations, we identify use three bins - Exceptional (top 5th percentile), Very High (top 10th percentile), and High (top 25th percentile) value - to describe points based on the distribution of AEV Index values within a particular region. All other points are classified as Other (lowest 75th percentile). Grid points classified within the Exceptional, Very High, or High bins are the most valuable within a particular region (and we often refer to the combination of all three bins as highly valuable areas), and since they are determined from consistent percentiles from regional distributions of the AEV Index they are directly comparable between regions.”

Lines 1021-1026: “To highlight the particularly valuable locations within each region, we have binned the AEV Index into four categories: Exceptional (AEV Index in the top 5th percentile for that region), very high (AEV Index in the top 10th percentile for that region), high (AEV Index in the top 25th percentile for that region), and other (all other points). Note that the percentile thresholds for these bins are determined separately for each of the regions and decades (see Figures S12 and S13).”

--

This manuscript was co-reviewed by and early career researcher and an established researcher.

Reviewer #5 (Remarks on code availability):

Code not provided - only processes data are available at the link provided.

Response:

We believe the reviewer missed the URL under the “Declarations” section that provides the link to the code availability. Since R1 and R3 commented that the code was available and well commented, if not in their preferred analysis language, we believe the repository link is universally accessible. We have updated all the analysis scripts as part of this revision and the data and code remain available at the links provided in the manuscript.

Declarations

- Funding: NASA Award 80NSSC21K1132 (AKD, LLL, ZS, CB, SJ); NASA Award 80NSSC20K1289 (KMK, LLL, SJ, CCC, BS, MNL); NSF Award 2037531 (MMH, LLL); NSF Award 2037561 (SJ, AE); US Department of Energy, Grant no. DE-SC0025505 (CN); Royal Society Rutherford Discovery Fellowship, Royal Society CSG-UOC2302 (MLR); Centre national de la recherche scientifique (SL);
- Competing interests: The authors declare no competing interests.
- Ethics approval and consent to participate: Not Applicable
- Consent for publication: Not Applicable
- Data availability:
Processed data used in this analysis is available at: <https://doi.org/10.5281/zenodo.14827913>
The raw CESM2 Large Ensemble data are freely available at: <https://www.earthsystemgrid.org/dataset/ucar.cgd.cesm2le.output.html>
The raw CESM2 FOSI data are freely available at: https://app.globus.org/file-manager?origin_id=6f5e56da-0353-4bd4-bac0-04a104e05d58&origin_path=%2F%2F%2F&two-pane=false
- Code availability:
Data processing and analysis code is available at: <https://github.com/duvivier/HotSpotsinIce>
- Materials availability: Not Applicable
- Author contribution: AKD, KMK, LLL, ZS, MMH, SJ, and CB conceptualized the science plan and methodology. SJ performed the penguin population projections with inputs from BS, CCC, and AE. MNL and KMK performed the FEISTY simulations and provided guidance on using the data. Original species illustrations provided by KMK. AKD performed the analysis and visualizations and wrote the original manuscript draft. All co-authors provided feedback throughout the analysis and reviewed and edited the manuscript.

Reviewer #6 (Remarks to the Author):

Response:

Thanks for taking the time to co-review the manuscript!

Reviewer #6 (Remarks on code availability):

Code was not included in manuscript.

Response:

We believe the reviewer missed the URL under the “Declarations” section that provides the link to the code availability. Since R1 and R3 commented that the code was available and well commented, if not in their preferred analysis language, we believe the repository link is universally accessible. We have updated all the analysis scripts as part of this revision and the data and code remain available at the links provided in the manuscript. (see above for screenshot)

Thank you to all the reviewers for providing additional comments on our revised manuscript. We have considered the comments and we believe that we have addressed all the concerns in detail below. All line numbers in responses refer to the clean revised manuscript.

Reviewer #1 (Remarks to the Author):

The authors have modified analyses, produced new figures, and substantially re-written large parts of the manuscript to take into account reviewers' comments. This has involved a large amount of work. I believe it is a much-improved manuscript, and the authors should be congratulated.

A few remaining comments to note. Any line numbers refer to the file with tracked changes. *Thanks for the thoughtful review and comments. We believe we have addressed your additional concerns, as detailed below.*

Fish biomass model: Outputs from the global FIESTY model have now been constrained and taken to represent Antarctic Toothfish by truncating potential suitable habitat at <1 deg C. Validation is qualitative based on Aquamaps which itself is based on the environmental envelope of occurrence data. We still do have some concerns over how well this model actually represents Antarctic demersal fish. To determine this would require validation with CCAMLR fisheries data, which is probably not in scope of this study.

Thanks for this comment. We agree that validation of the model would be helpful to test the degree to which it represents demersal fish. However, as the reviewer notes, this is well outside the scope of this study. Getting access to CCAMLR fishing data could help, but this data is unfortunately not publicly available. This would be a great project for a future study!

In a related point, Line 511-513 states that there are no SDMs for Antarctic Toothfish. Actually, a model linking Antarctic Toothfish occurrence to environmental data has been published:

<https://pmc.ncbi.nlm.nih.gov/articles/PMC10984185/>

It shows some similarities and differences to the one presented here.

Thank you for bringing this paper to our attention. It is interesting that this paper has many qualitatively similar findings to our results, including that Antarctic Toothfish habitat is predicted well by sea ice thickness, ocean temperature, and depth. We are continuing to use the FEISTY fish data since FEISTY is a mechanistic model and this SDM is statistical, which we think is important since it is unclear that the statistical relationships that exist in the present day may not be robust into future conditions. We have added text to the paper in the Discussion and Methods sections as detailed below to cite this study and how our results compare.

*Lines 580-588: Some detailed species-specific models already exist describing today's environmental conditions [e.g. 37–39], but these would require substantial work to couple with a global ESM and to assess potential future changes and if statistical relationships are robust in different climate states. For other species (e.g. Antarctic silverfish, *Pleuragramma antarctica*)*

there are no existing models of which we are aware, so species-specific model development would be necessary. Our study focuses on the Antarctic-specific and ice-dependent species for which future projection data are available from existing biological models and that use the types of data available from existing ESM experiments.

Lines 923-925: The Fisheries Size and Functional Type (FEISTY) model is a mechanistic global size- and trait- based model that resolves the structure of fish communities from environmental forcing [26].

Lines 958-960: The map of FEISTY biomass is also qualitatively similar to statistical predictions of Antarctic toothfish habitat, which shows that Antarctic toothfish can be particularly well predicted by sea ice thickness, ocean temperature, and depth [39].

Line 915-916: Actually, Antarctic Toothfish are caught commercially not only on the shelf, but in high abundances on the slope 1000-1700m. So hopefully the temperature limits used in this ms don't constrain fish biomass to just the shelf.

<https://doi.org/10.1016/j.fishres.2019.105338>

We did not mask the Antarctic Toothfish by the ocean depth, so this statement is a qualitative way to describe the locations where FEISTY shows highest demersal fish biomass. We have changed the wording in the manuscript to clarify this and thank you for bringing the paper to our attention, which we have now cited.

Lines 953-957: When we compare the locations of highest DFP from FEISTY with the best estimate of Antarctic toothfish habitat from AquaMaps [81], we see that the spatial patterns of maxima in DFP occur primarily on the continental shelves and continental slopes (Figure S10). These locations are consistent with catch data from East Antarctica of where Antarctic toothfish are likely to be found [83].

Comparison to Areas of Ecologically Significance, Lines 1031-1040: This comparison is interesting in it's own right and some of the species in this index and the AEV overlap, but I still don't think it is a validation/justification for a regionally weighted index. Where the indices are consistent and different are interesting points for discussion (potentially in the main ms rather than justifying the index in the supplemental material).

We appreciate the reviewers' perspective that regional weighting is not justified. However, because CCAMLR management for MPAs is by regional planning domains (see <https://gis.ccamlr.org/> or map, below), we feel that regional analysis is justified and valuable as it is consistent with CCAMLR management practices. Thus to be clear, we chose regional weighting based on CCAMLRs' approach, but not on the Areas of Ecological Significance approach or overlap.

We also agree that the AES comparison in itself is interesting, and some of this manuscripts' co-authors are currently working on analysis specifically comparing different Antarctic metrics of biodiversity and ecological importance since that is beyond the scope of this study. Primarily for space and clarity reasons, we have decided to keep this AEV/AES comparison in the methods, as we do for each of the input layers' validations. We have modified the text to make clear why this comparison is done and the scope.

Lines 1059-1071: A full comparison of biodiversity metrics around Antarctica is beyond the scope of this study. However, to demonstrate that the historical reconstruction AEV Index - both the hemispheric and regionally scaled versions - provides sensible information about valuable biological areas, we qualitatively compare it to the Areas of Ecological Significance (AES) identified by 12 using in-situ tracking data from 17 marine mammal and bird species to identify regions preferred by these predators. Both the AES and AEV Index metrics identify ecologically valuable regions using multiple species, yet it is important to note that while species overlap between these two metrics the methodology is entirely different and the metrics do not share any overlapping data. This comparison is done primarily to show that a model based metric to identify valuable biological areas is comparable to a similar metric that uses observational data, but a full analysis comparing multiple Antarctic metrics of biodiversity and biological importance is beyond the scope of this work.

Minor points:

Line 267: The 30-70% higher should include 'average'. Looking at figure 3a, while there is a consistent trend for the average AEV index to be higher inside polynyas than outside, the variability inside polynyas is also much higher than outside. This should be acknowledged.

Thanks for pointing this out, we have modified the text as follows.

Lines 193-196: The AEV Index also shows there is co-location of coastal polynyas and valuable areas across the ecosystem and that the AEV Index values within polynyas are on average 30-70% higher than outside polynyas, though there is higher variability within polynyas as well (Figure 3a; Table 1).

Line 314: The decline in AEV in the Ross Sea seems very minimal in Figure 4g. *We have modified the text to make clear the decline is minimal.*

Line 227: In the Ross Sea, the AEV Index decreases very slightly though still remains highly valuable in most areas.

Line 378-9: Prydz Bay does not appear to have a dramatic decline in projected toothfish biomass in Figure 6i), however the West Antarctic Peninsula does.

*Thanks for pointing out that this sentence was confusing. In Figure 6 we show the general demersal fish biomass potential, but it does not include the 1C temperature threshold masking the data. This is consistent with other studies of multiple fish model results. On Figure 6 we overlay the ocean bottom temperatures to show the environmental change that would affect habitat, but the fish contour field is not masked using the temperatures. In contrast, Figure 7 and Figure S5 do show the integrated biomass of demersal fish that **includes** masking with the 1C temperature threshold for Antarctic Toothfish. We have changed the text to make this clear.*

Lines 361-396: Demersal fish biomass potential also remains elevated on the continental shelves, in the vicinity of polynyas and is projected to increase given the likely increases in POC flux to the benthos with warming (Figure 6g-i). This increase in fish biomass is consistent with other studies across multiple models that show increasing fish biomass around Antarctica [32]. However, present day Antarctic toothfish habitat temperature thresholds (see Section 4) imply that this species would lose habitat since projected bottom temperatures increase from the 2000s to 2090s (Figure 6g-h, blue contours). As a result, regional projections of Antarctic toothfish potential biomass decline over the 21st century in most regions (Figure 7 e and f and S5) as the ocean bottom temperatures are projected to become increasingly uninhabitable for Antarctic toothfish as they exceed the upper bound of their habitat threshold range of 1 °C [33].

Line 452-453: This statement needs referencing “More mesozooplankton biomass and increased POC flux to the benthos would provide more prey resources for demersal fish”.

This statement is related to the FEISTY mechanistic forcing and we have modified the text as follows to make this clear.

Line 460: More mesozooplankton biomass and increased POC flux to the benthos would provide more prey resources for demersal fish through the FEISTY mechanistic forcing [26].

Line 529-532: Regarding the criticism of krill dynamic models, this is at least equally applicable to the parameterisation of the *global* FEISTY model being used to represent Antarctic demersal fish and yet it is part of the AEV index!

We understand your concern about the global FEISTY model not being validated for Antarctic fish specifically. Unfortunately, since CCAMLR Antarctic toothfish fishing records are not publicly available around Antarctica, this data is just not available. However, we still feel there is utility in including results from the global fish model because it is based on fish metabolism and trophic connections that have been validated in a wide variety of ocean environments where fishing records allow and extrapolating this model to novel environments is an interesting and insightful exercise. While there is the potential for sensitivity to our choice of fish model, all fish models presented in Heneghan et al. (2021) showed an increase in fish biomass in the Antarctic sea ice zone under climate warming (see their Figure 3). Therefore based on our current understanding of fish physiology, the potential for fish biomass to increase (as is shown from FEISTY) with warming appears to be a possibility, albeit it would likely be fish that are adapted to warmer conditions, not Antarctic species.*

**<https://doi.org/10.1016/j.pocean.2021.102659>*

Reviewer #1 (Remarks on code availability):

Code is extensive, but I can't evaluate as I am not a python user.

Thanks for confirming our code was accessible and annotated.

Reviewer #2 (Remarks to the Author):

Response:

Thanks for re-reading our revised manuscript and contributing to your co-reviewers' comments.

Reviewer #3 (Remarks to the Author):

I appreciate the revised version of the manuscript and the substantial work the authors have undertaken to address reviewer concerns and improve the manuscript. The main issues I raised have been appropriately addressed, and I am pleased with the improved manuscript.

This work represents a valuable contribution to our understanding of climate change impacts on Antarctic ecosystems, a research area that, as the authors correctly note, remains underrepresented in the literature (due to a lack of data that the authors have also faced). The study holds significant conservation value, and the revisions effectively highlight existing

knowledge gaps while addressing limitations in the proposed index. I particularly appreciate that the authors now clearly identify where their index may miss information, as this transparency not only demonstrates the robustness of their approach in certain contexts but also highlights important gaps that can inform future research priorities.

Thanks for the thoughtful review and comments. We appreciate your suggestions that have improved this manuscript substantially.

I have only two minor editorial suggestions:

Line 168 (clean version): The abbreviation "KGP" appears here without prior definition. Please introduce and define this term when it first appears in the text.

Thanks for finding this - we have now defined KGP at this first instance (Line 168).

Lines 173-175: Please specify which planning regions are included in "other planning regions." While readers could reference the figure, much of this paragraph discusses supplementary materials that many readers will not examine in detail. For clarity and to avoid relying on readers' memory of figure captions, particularly given the introduction of an "others" category in your index, please name these regions explicitly in the text.

We are interested in the CCAMLR MPA planning domains (see <https://gis.ccamlr.org/> and turn on MPA planning domains to see map, below).

We have changed the text here to be more precise about these regions:

Lines 174-175: "...within the CCAMLR MPA planning domains that may be locally critical for the ecosystem"

Reviewer #3 (Remarks on code availability):

Again, I am not proficient in Python code but the code is well annotated with a README file.

Response:

Thanks for confirming our code was accessible and annotated.

Reviewer #4 (Remarks to the Author):

The authors have made significant changes to the manuscript and have addressed my suggestions and concerns. I now recommend publication of this manuscript.

Reviewer #4 (Remarks on code availability):

I have checked the link provided for the code and the code is made available. I am not proficient enough in python to provide an evaluation of the code itself. It looks well-organized and provides a README file.

Thanks for confirming our code was accessible and annotated.

Reviewer #5 (Remarks to the Author):

We commend the authors for thoroughly responding to a large set of review comments and note that the revisions have improved the rigour of the manuscript. Parts of the methodology have been modified following the initial review, including improvements in the treatment of fish habitat to reflect polar fish requirements, validation of the sea ice environmental variable and a sensitivity analysis.

Thanks for the thoughtful review and comments. We believe we have addressed your additional concerns, as detailed below.

Major points:

a) While the authors have much more clearly caveated the scope of their Antarctic Ecosystem Value Index in the revised manuscript – including an explanation of their use of the term “valuable” at the end of the Introduction and a new subsection of the Discussion on “Gaps in Ecosystem Representation” – we feel that this scope is still overstated in some parts of the manuscript. Notably, in the abstract line 059 “biologically valuable hot spots for the Antarctic ecosystem” would be more accurately stated as “biologically valuable hot spots for the groups considered, and potentially for the ecosystem more broadly” (as this has not been assessed in the study), and at the beginning of the Discussion where the authors claim they have answered the question “Where are the most ecologically valuable marine areas around Antarctica and will these locations change in the future” we do not think is accurate for the reasons outlined in our previous review (i.e. key trophic groups are excluded from the index).

You are correct that we have not included all possible species that are part of the Antarctic ecosystem, though we did choose to focus on Antarctic-specific, ice-dependent species for which we had the ability to use standard coupled ESM data with biological models to evaluate present and future conditions. To address your concern about the scope, we have modified the text as follows.

Lines 58-59: ...these areas are biologically valuable hot spots for a number of ice-dependent Antarctic Species.

Lines 409-435: Where are the most ecologically valuable marine areas around Antarctica and will these locations change in the future? To answer this question, our study presents the AEV Index, a model-based integrative tool that synthesizes information across trophic levels for several species within the seasonal sea ice zone to investigate the present and future ecological value of broader regions. The ecological input layers we use include some representing ice-dependent species that are core ecosystem components in the Southern Ocean.

b)Both in the revised manuscript and in the response to review comments the authors state that they “are unaware of any other species for which that models linking environmental conditions to the occurrence of a species is available and which we could include in these calculations at this time” (note there seems to be a slight issue with the formulation of this sentence). We think it is important that the authors provide an explanation as to why they are unable to use other available models, for example SEAPODYM (micronekton) (Green 2022), KRILLPODYM (Green et al. 2023), or MICE models available for baleen whales (Tulloch et al. 2017, 2019). If a set of criteria have been used to determine model suitability for the index could the authors please specify these? There is of course no problem with making particular choices about model suitability – such as using habitat suitability models rather than mechanistic models for krill and fish (but metapopulation models for penguins), but we think this needs to be explained in the Methods.

Thanks for bringing up this concern. Our model selection was driven by a pragmatic set of criteria focused on data availability (e.g., does the ESM output the necessary variables at the right frequency?), model complexity, and the specific objectives of our AEV Index. We have clarified this in the text as follows.

Line 795-804: The AEV index uses five inputs from different trophic levels in the Antarctic ecosystem to provide a broader metric across trophic levels (illustrated graphically in Figure 2). This research prioritizes the use of models with reliance on readily available environmental variables produced by standard ESM data, which is available at monthly temporal resolution. While this limits the biological models we can use, it highlights the models that are capable of linking with an ESM and thus providing a framework for integrating large-scale environmental change to ecological/biological processes. This is a tradeoff that allows our research to

simultaneously make broad projections and highlight pathways for additional research to address.

We are familiar with the SEAPODYM and KRILLPODYM models. While KRILLPODYM offers a mechanistic, spatially resolved model of Antarctic krill distribution and abundance, its detailed life history requirements, such as 291 age classes each lasting one week, would necessitate a level of temporal and spatial resolution in environmental data that was not compatible with our broader framework of using existing ESM model data (usually at monthly resolution) and computational availabilities since we could not re-run large ensemble experiments to increase temporal frequency. While ESMs are suitable for our broader regional analysis, their output do not adequately support the fine-grained advection and life-stage tracking required by models like KRILLPODYM, which has data requirements and inherent complexities, particularly concerning fine-scale spatial and temporal dynamics, that were not compatible with the scope and available data from our ESM simulations for a comprehensive, circumpolar assessment across multiple trophic levels and future projections. We have modified the text as follows to clarify why these models were not used in this analysis.

Line 609-614: While more complex Antarctic krill models that incorporate biomass dynamics and full life cycle processes could provide a more detailed perspective [e.g. 41], such models were not used in this study because of the necessity to use existing coupled ESM data which does not have temporal or spatial resolution needed. A more sophisticated regional modeling study of krill in some of the large scale regions identified by this study would be valuable.

We were unfamiliar with the MICE baleen whale model and we very much appreciate the reviewer bringing this to our attention! We have read the paper you reference and while we think that including this model would be interesting, we feel that including it at this stage is beyond the scope of the present study. Some major questions arose, including how to account for the sea ice importance of a species that migrates far from the ice edge when the AEV Index is calculated only in the seasonal ice zone. Additionally, we felt that using MARBL mesozooplankton output for the krill and copepod components of MICE would require careful consideration and substantial testing. Thus, we are not including the whales in the present study, though we would be interested in integrating MICE with the CESM and MARBL in the future, so if you are the MICE developer and interested in collaborating please feel free to reach out. Since we feel that integrating MICE output at this time is beyond the scope of this present study, we have clarified in the text that we are focusing on ice-dependent species with existing models in text.

Line 574-600: When possible, the AEV Index uses species-specific future projections, but more detailed species-specific modeling is needed, especially for ice-obligate species. We include input layers for primary producers (NPP), primary/secondary consumers (KGP), secondary consumers (DFP), and secondary/tertiary consumers (EPP/APP) (see Section 4 for details).

However, other key ecosystem species like whales, seals, other penguins and fish, and flighted birds are not included. Some detailed species models already exist [e.g. 37–39] though may require substantial work to couple with a global ESM, but in other cases more detailed species specific model development may be required. This study focuses on the Antarctic-specific and ice-dependent species for which future projection data are available from biological models and that use the types of data available from existing ESM experiments. However, this means that we do not include other mesozooplankton or pelagic fish that are important for energy pathways from the primary producers to predators [e.g. 40, 41]. Additionally, excluding species means that we may not be capturing the scale needed for protection of all critical ecosystem processes, and we can only assess processes relevant to the species represented as AEV Index inputs. While it would have been ideal to include a larger range of species, we included only those species for which we had readily available future projection information around the entire Antarctic continent. The results presented here show that the AEV Index framework is valuable as a tool to demonstrate the utility of such studies across trophic levels, and future work to include more species would help fill gaps about the Antarctic ecosystem that this study is unable to address. Until then, the AEV Index is associated with larger uncertainty in regions or at times when species not included in our assessment play a vital role in the ecosystem.

c) If policy recommendations are being included in the Discussion (Section 3.2) then we think it is important that the authors make it clear that this is an initial/exploratory study in the application of the AEV Index. While much effort has clearly been put into this manuscript, there are multiple key ecosystem processes that are not captured by the approach and choice of models (e.g. krill growth model not capturing sea ice conditions and the key trophic roles of groups such as mesopelagic fish, (McCormack et al. 2020)).

We agree that our study alone should not be the basis of policy recommendations. Additionally, a number of the authors are prohibited from giving policy recommendations or advocating for particular policies by our institutions. Thus, we intentionally do not include explicit policy recommendations. We have changed the text as follows to make clear the exploratory nature of this study as well as the need for additional work and to ensure we are not recommending a policy decision.

Lines 540-554: The primary outcomes of this work are the identification of highly valuable regions across the marine ecosystem in the present-day and the assessment to what extent they remain valuable into the future. It is important to note that this is an exploratory scale that uses standard ESM spatial and temporal resolution data to identify locations valuable to the marine ecosystem at a large scale, but that more detailed studies for these regions is necessary for specific policy decisions. Still, this initial study identifies some important patterns. In most management regions. In most management regions, the majority (55-100%) of exceptional value areas are within current proposed or existing MPA boundaries; the exception is the Amundsen Sea (discussed below). The existing Ross Sea region MPA protects the majority of exceptional,

very high, and high value areas in this region in the present day (Table 1), and is projected to keep protecting the majority of these high value areas (Table 2). While the MPA is set to expire in 2052, this study suggests that keeping the MPA in effect would likely continue to protect most of these valuable regions through 2090.

Minor points:

a) Lines 605-606: “we were able to investigate two important environmental variables known to impact Antarctic krill: SST and sea ice availability”. The sentence suggests that there is a modelled linkage between krill and sea ice, which is not the case as lines 597-598 state that “(krill growth potential) reflects adult krill growth and uses only surface chlorophyll and sea surface temperature (SST) as inputs”. It should be clarified that the modelled krill metric is not affected by sea ice, with appropriate limitations/caveats about result interpretation and potential future effects of sea ice decline (also see Meyer et al. 2017 for more details about krill habitat usage that might be relevant to the effects of sea ice loss on krill populations).

Thanks for bringing up that this is a confusing sentence. We have modified the sentence as follows to clarify how we consider sea ice availability.

Lines 615-619: Nonetheless, we were able to investigate two important environmental variables known to impact Antarctic krill: SST and sea ice availability. By assessing KGP only within the sea ice zone we implicitly consider only locations where sea ice is available during the year, though not the seasonality or direct impact of sea ice on the krill lifecycle. Additionally, krill are stenothermic...

b) Line 857 (KGP methods): Initial krill length (L) is specified as 40cm – is this correct?
This is a typo, thank you for catching it - it should read 40 mm and we have fixed this (Line 884).

References

- Green, DB (2022). The missing link : pelagic prey field prediction for Southern Ocean predators. University of Tasmania. Thesis. <https://doi.org/10.25959/23973123.v1>
- Green et al. (2023) KRILLPODYM: a mechanistic, spatially resolved model of Antarctic krill distribution and abundance [https://www.frontiersin.org/journals/marine-science/articles/10.3389/fmars.2023.1218003 /full#B52](https://www.frontiersin.org/journals/marine-science/articles/10.3389/fmars.2023.1218003/full#B52)
- McCormack SA, Melbourne-Thomas J, Trebilco R, Blanchard JL, Raymond B, Constable A. Decades of dietary data demonstrate regional food web structures in the Southern Ocean. *Ecol Evol.* 2021; 11: 227–241. <https://doi.org/10.1002/ece3.7017>
- Meyer, B., U. Freier, V. Grimm, J. Groeneveld, B. P. V. Hunt, S. Kerwath, R. King, C. Klaas, E. Pakhomov, K. M. Meiners, J. Melbourne-Thomas, E. J. Murphy, S. E. Thorpe, S. Stammerjohn, D. Wolf-Gladrow, L. Auerwald, A. Gamp, L. Halbach, S. Jarman, S. Kawaguchi, T. Krumpfen, G. Nehrke, R. Ricker, M. Sumner, M. Teschke, R. Trebilco, and

N. I. Yilmaz. 2017. The winter pack-ice zone provides a sheltered but food-poor habitat for larval Antarctic krill. *Nature Ecology & Evolution* 1:1853–1861.

- Tulloch VJD, Plagányi ÉE, Matear R, Brown CJ, Richardson AJ. Ecosystem modelling to quantify the impact of historical whaling on Southern Hemisphere baleen whales. *Fish Fish.* 2018; 19: 117–137. <https://doi.org/10.1111/faf.12241>
- Tulloch VJD, Plagányi ÉE, Brown C, Richardson AJ, Matear R. Future recovery of baleen whales is imperiled by climate change. *Glob Change Biol.* 2019; 25: 1263–1281. <https://doi.org/10.1111/gcb.14573>

Reviewer #5 (Remarks on code availability):

Apologies we had previously missed the link provided for the Python code. We have not reviewed the code as we are not experts in Python coding.

Thanks for confirming our code was accessible.

Reviewer #6 (Remarks to the Author):

Thanks for re-reading our revised manuscript and contributing to your co-reviewers' comments!